# Minor First, Major Last: A Depth-Induced Implicit Bias of Sharpness-Aware Minimization

**Chaewon Moon**
Kim Jaechul Graduate School of AI, KAIST
`chaewon.moon@kaist.ac.kr`

**Dongkuk Si**
Mobilint, Inc.
`dongkuk@mobilint.com`

**Chulhee Yun**
Kim Jaechul Graduate School of AI, KAIST
`chulhee.yun@kaist.ac.kr`

## Abstract

We study the implicit bias of Sharpness-Aware Minimization (SAM) when training $L$-layer linear diagonal networks on linearly separable binary classification. For linear models ($L = 1$), both $\ell_\infty$- and $\ell_2$-SAM recover the $\ell_2$ max-margin classifier, matching gradient descent (GD). However, for depth $L = 2$, the behavior changes drastically—even on a single-example dataset. For $\ell_\infty$-SAM, the limit direction depends critically on initialization and can converge to $\mathbf{0}$ or to any standard basis vector, in stark contrast to GD, whose limit aligns with the basis vector of the dominant data coordinate. For $\ell_2$-SAM, we show that although its limit direction matches the $\ell_1$ max-margin solution as in the case of GD, its finite-time dynamics exhibit a phenomenon we call *sequential feature amplification*, in which the predictor initially relies on minor coordinates and gradually shifts to larger ones as training proceeds or initialization increases. Our theoretical analysis attributes this phenomenon to $\ell_2$-SAM's gradient normalization factor applied in its perturbation, which amplifies minor coordinates early and allows major ones to dominate later, giving a concrete example where infinite-time implicit-bias analyses are insufficient. Synthetic and real-data experiments corroborate our findings.

## 1 Introduction

Modern deep networks often generalize well despite extreme over-parameterization. One explanation emphasizes the geometry of the objective: models perform better when optimization settles in flatter regions of the landscape (Hochreiter & Schmidhuber, 1994; Keskar et al., 2016; Neyshabur et al., 2017; Jiang et al., 2019). Motivated by this view, Foret et al. (2020) introduce Sharpness-Aware Minimization (SAM), which seeks parameters that minimize the worst-case loss within a small neighborhood. Following its empirical success (Chen et al., 2021; Bahri et al., 2021; Kaddour et al., 2022a), various theoretical works have analyzed SAM's implicit bias to understand its effectiveness (Andriushchenko & Flammarion, 2022; Behdin & Mazumder, 2023a; Zhou et al., 2025). However, these analyses primarily apply to scenarios with attainable finite minimizers (e.g., squared loss), leaving open the case of losses whose infimum lies at infinity (e.g., logistic loss).

We consider the implicit bias of SAM when training $L$-layer linear diagonal networks on linearly separable classification datasets with logistic loss. We study two variants of SAM, $\ell_\infty$-SAM and $\ell_2$-SAM, named after the norm defining their local perturbation (See Section 2). For $L = 1$ (linear models), gradient descent (GD) is known to converge in direction to the $\ell_2$ max-margin classifier (Soudry et al., 2018). For both $\ell_\infty$-SAM and $\ell_2$-SAM, we show that they also align with the same limit direction. Thus, SAM does not change the implicit bias here, as shown in Figure 1a.

However, for 2-layer diagonal linear networks, we find that the trajectory of the linear coefficient vector $\boldsymbol{\beta}(t)$ under both $\ell_\infty$- and $\ell_2$-SAM can differ substantially from the maximum $\ell_1$-margin implicit bias of GD (Gunasekar et al., 2018b). In Figure 1b, we consider a toy separable dataset $\{(\boldsymbol{\mu}, +1)\}$ with $\boldsymbol{\mu} = (1, 2)$. In this case, the $\ell_1$ max-margin direction is $\boldsymbol{e}_2 = (0, 1)$, the standard basis vector for the major component of $\boldsymbol{\mu}$. As predicted, all GD trajectories and some SAM tra-

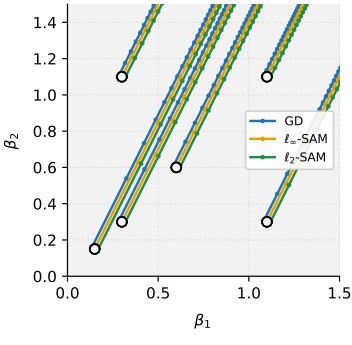
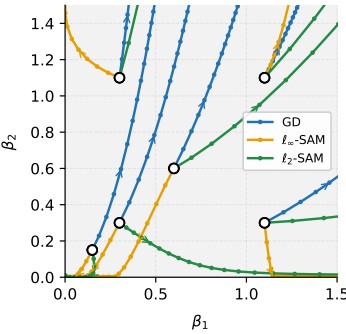

(a) Depth 1 (linear network)    (b) Depth 2 (linear diagonal network)

Figure 1: Trajectories of the predictor $\boldsymbol{\beta}(t) \in \mathbb{R}^2$ from identical initial conditions under discrete GD, $\ell_\infty$-SAM and $\ell_2$-SAM on $\{(\boldsymbol{\mu}, +1)\}$ with $\boldsymbol{\mu} = (1, 2)$. We used $\eta = 0.3$ and $\rho = 1$ for SAM.

jectories show increasing alignment of $\boldsymbol{\beta}(t)$ with $\boldsymbol{e}_2$. However, for some initializations, we observe that some trajectories of $\boldsymbol{\beta}(t)$ under $\ell_\infty$-SAM and $\ell_2$-SAM instead converge to zero, or even align with $\boldsymbol{e}_1 = (1, 0)$—a seemingly paradoxical implicit bias favoring the *minor* feature rather than the major one. It is interesting that the addition of a single layer—from $L = 1$ to $L = 2$—introduces this peculiar behavior of SAM different from GD, even for the simple setting of linear diagonal networks trained with a single example.

## 1.1 Summary of Our Contributions

We analyze the optimization trajectory and implicit bias of $\ell_\infty$-SAM and $\ell_2$-SAM in $L$-layer linear diagonal networks trained on linearly separable data with logistic loss. For theoretical analysis, we analyze the evolution of the linear coefficient $\boldsymbol{\beta}(t)$ of the linear diagonal network under *continuous-time* versions of SAM, $\ell_\infty$-**SAM flow** and $\ell_2$-**SAM flow**. We characterize their limit directions, obtained when training on general linearly separable data, and their pre-asymptotic behavior before aligning with the limit directions, analyzed on a single-example dataset $\{(\boldsymbol{\mu}, +1)\}$.

- **Depth 1 (linear).** For linear models ($L = 1$), both $\ell_\infty$-SAM flow and $\ell_2$-SAM flow have the same $\ell_2$ max-margin implicit bias as GD on linearly separable data; in the single-example setting, we further show that the $\ell_\infty$-SAM coincides exactly with the GD trajectory.

- **Depth $L$, $\ell_\infty$-SAM.** For $L \geq 2$ and $\ell_\infty$-SAM flow, we characterize the coordinate-wise trajectory of $\boldsymbol{\beta}(t)$ determined by the relative scale of each coordinate at initialization and the perturbation radius of $\ell_\infty$-SAM (Theorem 3.2). For almost all initializations, $\boldsymbol{\beta}(t)$ diverges and its limit direction is one of the standard basis vectors $\boldsymbol{e}_1, \ldots, \boldsymbol{e}_d$ or it converges to a finite point (Corollary 3.5). Compared to GD, the limit direction of $\ell_\infty$-SAM becomes more sensitive to initialization.

- **Depth 2, $\ell_2$-SAM.** For $L = 2$ and $\ell_2$-SAM flow, we first prove that the limit direction (if convergent to zero loss) is the $\ell_1$ max-margin solution (Theorem 4.2); however, this infinite-time characterization does not explain our observation from Figure 1b. We empirically investigate the finite-time trajectory of $\boldsymbol{\beta}(t)$ and identify the **sequential feature amplification** phenomenon, in which $\boldsymbol{\beta}(t)$ initially relies on minor coordinates and gradually shifts to larger ones as $t$ increases or initialization scale grows. We provide a theoretical explanation of both time-wise (Theorem 4.4) and initialization-wise (Theorem 4.5) aspects of the phenomenon. This example shows that focusing only on the $t \to \infty$ limit can overlook aspects of the training dynamics. SAM provides a clear instance where a *finite-time* view is essential to understanding how its implicit bias emerges. This finite-time bias offers a possible theoretical perspective on recent empirical observations that SAM captures atypical subpatterns more effectively than GD (Kim et al., 2026).

- We present synthetic and real-data experiments to corroborate our findings.

## 1.2 Related Work

**Implicit Bias of GD on Linear Diagonal Networks.** Soudry et al. (2018) show that, for linearly separable logistic regression, the weight of a linear model diverges while the direction converges to the $\ell_2$ max-margin classifier. For linear diagonal networks, GD biases toward sparse predictors (Gunasekar et al., 2018b), with 2-layer models converging to the $\ell_1$ max-margin direction under

directional convergence (Ji & Telgarsky, 2020). Subsequent papers have studied this implicit bias in sparse regression, in which initialization scale governs the bias: large initialization favors $\ell_2$-type bias, while small initialization favors $\ell_1$-type sparsity (Woodworth et al., 2020; Yun et al., 2020; Moroshko et al., 2020). Other works study stochastic effects, kernel-regime behavior, and implicit regularization via mirror flow or related dynamics (Pesme et al., 2021; Even et al., 2023; Nacson et al., 2022; Jacobs et al., 2025; Wang & Klabjan, 2024; Papazov et al., 2024; Jacobs & Burkholz, 2024); we provide a brief overview in Appendix A.2.1. Prior work on small-initialization GD under squared loss in the same diagonal network setting shows incremental *saddle-to-saddle* learning dynamics, where coordinates become active in discrete stages as the predictor moves between saddles (Berthier, 2023; Pesme & Flammarion, 2023). We provide a detailed comparison between our setting and these saddle-to-saddle dynamics in Appendix A.2.2.

**Sharpness-Aware Minimization.** Motivated by the relationship between sharpness and generalization (Hochreiter & Schmidhuber, 1994; Keskar et al., 2016; Jiang et al., 2019; Neyshabur et al., 2017), Foret et al. (2020) propose SAM. Extensive empirical work has shown the superior performance of SAM and its variants across tasks and architectures (Sun et al., 2024; Kwon et al., 2021; Li et al., 2024b; Liu et al., 2022; Yun & Yang, 2023; Bahri et al., 2021; Zhuang et al., 2022; Kaddour et al., 2022b). Complementing these empirical findings, theoretical work has analyzed SAM's optimization dynamics, generalization, and implicit bias (Li et al., 2024a; Behdin & Mazumder, 2023b; Zhang et al., 2024; Agarwala & Dauphin, 2023; Wen et al., 2023; Long & Bartlett, 2024; Zhou et al., 2024; Springer et al., 2024; Baek et al., 2024; Chen et al., 2023; Chang & Khanna, 2026), including results in simplified settings such as diagonal linear networks on MSE loss (Andriushchenko & Flammarion, 2022; Clara et al., 2025). Recent work highlights convergence instabilities of SAM near local minima (Si & Yun, 2023; Kim et al., 2023). See Appendix A.2.3 for details.

## 2 PRELIMINARIES

**Notation.** We write the $i$-th standard basis vector as $e_i$. For $n \in \mathbb{N}$, let $[n] = \{1, \cdots, n\}$. For a vector $v \in \mathbb{R}^d$, we denote its coordinates by $v = (v_1, \cdots, v_d)$. For any block vector $Z = (z^{(1)}, \ldots, z^{(L)}) \in (\mathbb{R}^d)^L$, we denote its $\ell$-th block by $Z^{(\ell)} := z^{(\ell)} \in \mathbb{R}^d$. For $a, b \in \mathbb{R}^d$, $a \odot b$ denotes the element-wise product; for a collection $\{a^{(\ell)}\}_{\ell=1}^L$, we write $\bigodot_{\ell=1}^L a_l := a^{(1)} \odot \cdots \odot a^{(L)}$.

**Model.** We consider $L$-layer linear diagonal networks, a simple family of homogeneous networks widely used for the study of implicit bias (See Section 1.2). Let $\theta = (w^{(1)}, \ldots, w^{(L)}) \in (\mathbb{R}^d)^L$ be the parameter vector. For $x \in \mathbb{R}^d$, let the linear coefficient vector $\beta(\theta)$ and output $f(x)$ be

$$\beta(\theta) := \bigodot_{\ell=1}^L w^{(\ell)} \in \mathbb{R}^d, \quad f(x) := \langle \beta(\theta), x \rangle.$$

**Data and Loss.** We consider the standard supervised learning setting where a binary classification dataset $\{(x_i, y_i)\}_{i=1}^N$ is given. Let the logistic loss be $\ell(u) = \log(1 + \exp(-u))$. Then the training loss function is defined as $\mathcal{L}(\theta) := \frac{1}{N} \sum_{i=1}^N \ell(y_i \langle \beta(\theta), x_i \rangle)$. We write the gradient of $\mathcal{L}$ with respect to $\theta$ in a block form, as $\nabla \mathcal{L}(\theta) = (\nabla_{w^{(1)}} \mathcal{L}(\theta), \ldots, \nabla_{w^{(L)}} \mathcal{L}(\theta))$.

**Optimization Algorithms.** In this paper, we mainly consider the implicit bias of **Sharpness-Aware Minimization** (**SAM**, Foret et al. (2020)) and how depth causes it to deviate from the baseline algorithm, **gradient descent** (**GD**). At iteration $t$, a GD update reads $\theta(t+1) := \theta(t) - \eta \nabla \mathcal{L}(\theta(t))$, where $\eta > 0$ is called the step size or learning rate.

On the other hand, SAM updates parameters by evaluating the gradient at a perturbed one:

$$\hat{\theta}(t) := \theta(t) + \varepsilon_p(\theta(t)), \quad \theta(t+1) := \theta(t) - \eta \nabla \mathcal{L}(\hat{\theta}(t)),$$

where the perturbation $\varepsilon_p(\theta(t))$ is the approximate worst-case direction inside the $\ell_p$-ball of perturbation radius $\rho > 0$: $\varepsilon_p(\theta) := \arg\max_{\|\varepsilon\|_p \le \rho} \varepsilon^\top \nabla \mathcal{L}(\theta)$. We refer to $\hat{\theta}$ as the ascent point. Since $\theta = (w^{(1)}, \ldots, w^{(L)})$ has a block structure, we also write $\hat{\theta} = (\hat{w}^{(1)}, \ldots, \hat{w}^{(L)})$ and $\varepsilon_p(\theta) = (\varepsilon_p^{(1)}(\theta), \ldots, \varepsilon_p^{(L)}(\theta))$ so that we can say $\hat{w}^{(i)} = w^{(i)} + \varepsilon_p^{(i)}(\theta)$. For $p = 2$ and $\infty$, the perturbation $\varepsilon_p(\theta)$ has clean closed-form solutions:

$$\varepsilon_2(\theta) := \rho \frac{\nabla \mathcal{L}(\theta)}{\|\nabla \mathcal{L}(\theta)\|_2}, \quad \varepsilon_\infty(\theta) := \rho \, \text{sign}(\nabla \mathcal{L}(\theta)),$$

and we consider the two variants, referred to as $\ell_2$-**SAM** when $p = 2$ and $\ell_\infty$-**SAM** when $p = \infty$. For $p = \infty$, the maximizer is not unique when a coordinate of the gradient is zero. To make sure that the update is uniquely determined, we adopt the convention $\text{sign}(0) := 0$, applied coordinate-wise.

**Continuous-time Flows.** In the study of optimization algorithms, it is often useful to reduce the original discrete-time updates of an optimizer to a corresponding continuous-time flow. Unless the step size is too large, continuous-time flows offer a good approximation of the discrete-time optimizers, while allowing for clean and simplified analyses.

For GD, a common continuous-time counterpart is **gradient flow (GF)**: $\dot{\boldsymbol{\theta}}(\tau) = -\nabla\mathcal{L}(\boldsymbol{\theta}(\tau))$. With gradient flow, the analysis of GD trajectory boils down to solving an ordinary differential equation (ODE). Likewise, we define and study the flow counterparts of SAM, governed by the ODE

$$\dot{\boldsymbol{\theta}}(\tau) = -\nabla\mathcal{L}(\hat{\boldsymbol{\theta}}(\tau)). \tag{1}$$

Depending on the choice of norm, we will use the terms $\ell_\infty$-**SAM flow** and $\ell_2$-**SAM flow** to refer to the continuous-time versions of SAM. Figure 6 in Appendix A.1 plots the trajectory of $\ell_\infty$-SAM flow and $\ell_2$-SAM flow under the same setup of Figure 1. We observe that the trajectories stay almost the same and the surprising implicit bias of SAM carries over to SAM flows. Hence, we aim to understand this unusual behavior of SAM by studying the corresponding SAM flows.

**Rescaled Flows.** As shown in Appendix A.3, for the special case of single-example dataset $\{(\boldsymbol{\mu}, +1)\}$, the $\ell_p$-SAM flow ($p = 2, \infty$) of the $i$-th layer weight follows the *same spatial trajectory* as the following **rescaled $\ell_p$-SAM flow**:

$$\dot{\boldsymbol{w}}^{(i)}(t) = \boldsymbol{\mu} \odot \left( \bigodot_{\ell \neq i} \left( \boldsymbol{w}^{(\ell)}(t) + \boldsymbol{\varepsilon}_p^{(\ell)}(\boldsymbol{\theta}(t)) \right) \right), \tag{2}$$

obtained by taking out the loss derivative $-\ell'(\langle \boldsymbol{\beta}(\hat{\boldsymbol{\theta}}(t)), \boldsymbol{\mu}\rangle) > 0$ from the original $\ell_p$-SAM flow. Note that the original $\ell_p$-SAM flow (1) and the rescaled flow in (2) differ only by a *reparameterization of time*. Let $\boldsymbol{\theta}_{\text{orig}}(t_{\text{orig}})$ denote the original SAM flow and $\boldsymbol{\theta}(t)$ the rescaled flow. Then there exists a strictly increasing map $t_{\text{orig}} = \tau(t)$ such that $\boldsymbol{\theta}_{\text{orig}}(\tau(t)) = \boldsymbol{\theta}(t)$. Applying the chain rule yields the relation

$$\frac{d\boldsymbol{\theta}}{dt} = \frac{d\boldsymbol{\theta}_{\text{orig}}}{d\tau}\frac{d\tau}{dt} = -\frac{\nabla\mathcal{L}(\hat{\boldsymbol{\theta}}_{\text{orig}}(\tau(t)))}{\ell'(\boldsymbol{\beta}(\hat{\boldsymbol{\theta}}(t))^\top\boldsymbol{\mu})}, \qquad \frac{d\tau}{dt} = -\frac{1}{\ell'(\boldsymbol{\beta}(\hat{\boldsymbol{\theta}}(t))^\top\boldsymbol{\mu})}.$$

Since $\ell'(u) \uparrow 0$ as $u \to \infty$, the rescaled flow accelerates time in the large-margin regime. Formally,

$$\tau(t) = \int_0^t -\frac{1}{\ell'(\boldsymbol{\beta}(\hat{\boldsymbol{\theta}}(s))^\top\boldsymbol{\mu})}ds.$$

The rescaled flow makes the analysis easier due to the omitted term. Since our goal is to gain a better understanding of the spatial trajectory, we study the rescaled SAM flows in our analysis.

**Directional Convergence.** Let $\boldsymbol{\beta} : [0, T_{\max}) \to \mathbb{R}^d$ be a trajectory with maximal existence time $T_{\max} \in (0, \infty]$. We say that $\boldsymbol{\beta}(t)$ **converges in direction** or **directionally converges** if the limit $\bar{\boldsymbol{\beta}}^\infty = \lim_{t \to T_{\max}} \frac{\boldsymbol{\beta}(t)}{\|\boldsymbol{\beta}(t)\|}$ exists. In this case, $\bar{\boldsymbol{\beta}}^\infty$ is called the **limit direction** of $\boldsymbol{\beta}$.

## 3 SAM WITH $\ell_\infty$-PERTURBATIONS

We begin with $\ell_\infty$-SAM. For single-example data, its counterpart—rescaled $\ell_\infty$-SAM flow—has the nice property that each coordinate evolves independently, enabling an exact characterization of the trajectory for any depth $L$.

### 3.1 DEPTH-1 NETWORKS

We start with the depth-1 case, in which the implicit bias of $\ell_\infty$-SAM coincides with that of GD.

**Theorem 3.1.** *For almost every dataset which is linearly separable, any perturbation radius $\rho$ and any initialization, consider the linear model $f(\boldsymbol{x}) = \langle \boldsymbol{w}, \boldsymbol{x}\rangle$ trained with logistic loss. Then, $\ell_\infty$-SAM flow directionally converges in the $\ell_2$ max-margin direction.*

The proof is deferred to Appendix C.1. Since Theorem 3.1 holds for any $\rho$, it also recovers the implicit bias of GF. While Theorem 3.1 characterizes the limit direction for almost all linearly separable datasets, Theorem C.1 shows that, for the single-example data, the $\ell_\infty$-SAM flow follows the

same trajectory as GF. The yellow lines in Figure 6a depict the flows. As $t \to \infty$, $\boldsymbol{w}(t)$ converges in direction to the $\ell_2$ max-margin direction $\boldsymbol{\mu}$. Hence, when $L = 1$, GD and $\ell_\infty$-SAM share the same bias toward the $\ell_2$ max-margin solution, independent of the initialization.

## 3.2  DEEPER NETWORKS ($L \geq 2$)

To isolate the depth-induced implicit bias of SAM from effects of data-point configuration and obtain a tractable characterization of the SAM dynamics, we analyze the minimalist separable dataset $\mathcal{D}_{\boldsymbol{\mu}} := \{(\boldsymbol{\mu}, +1)\}$ with feature vector $\boldsymbol{\mu} \in \mathbb{R}^d$ satisfying $0 < \mu_1 < \cdots < \mu_d$; without loss of generality, we assume this monotone ordering of $\mu_i$'s. The additional technical difficulties that arise in the multi-point setting are deferred to Appendix C.2. In Appendix C.6, we empirically verify that the same behaviors persist under multi-point datasets and discrete SAM updates, suggesting that our insights extend beyond the single-point setting.

In contrast to the depth-1 case, for deeper (linear diagonal) networks, the implicit bias of $\ell_\infty$-SAM differs from GD—even on this single-example dataset. For example, when $L = 2$, while GD always aligns with the major feature, $\ell_\infty$-SAM can favor minor features depending on the initial condition. For $L \geq 3$, we show that the implicit bias of $\ell_\infty$-SAM is more sensitive to initialization than GD, in the sense that a wider range of initialization leads to solutions focusing on minor features. The next theorem characterizes the trajectory selected by the flow for different choices of initialization.

**Theorem 3.2.** *For $i \in [L]$, suppose $\boldsymbol{w}^{(i)}(0) = \boldsymbol{\alpha} \in \mathbb{R}_+^d$. Let $\boldsymbol{w}^{(i)}(t)$ follow the rescaled $\ell_\infty$-SAM flow (2) with perturbation radius $\rho > 0$ on the dataset $\mathcal{D}_{\boldsymbol{\mu}}$. Then, for the $j$-th coordinate of $\boldsymbol{\beta}(t)$:*

- *If $\alpha_j < \rho$, then $\beta_j(t)$ converges to 0 if $L$ is even, or to $\rho^L$ if $L$ is odd.*
- *If $\alpha_j = \rho$, then $\beta_j(t) = \rho^L$ for all $t \geq 0$.*
- *If $\alpha_j > \rho$ and $L = 2$, then $\beta_j(t)$ grows exponentially: $\beta_j(t) = \Theta(\exp(2\mu_j t))$.*
- *If $\alpha_j > \rho$ and $L > 2$, let $J := \arg\max_{j:\alpha_j>\rho} \mu_j(\alpha_j - \rho)^{L-2}$, and also let $T := \min_{k \in J} 1/(L-2)\mu_k(\alpha_k - \rho)^{L-2}$. If $j \in J$, then $\beta_j(t) \to \infty$ as $t \to T$; otherwise, $\beta_j(t)$ stays bounded for all $t < T$.*

We provide the proof of Theorem 3.2 in Appendix C.3. The behavior of each coordinate $\beta_j(t)$ is completely determined by whether the initialization $\alpha_j$ lies below, at, or above the threshold $\rho$. In each of these three regimes, $\beta_j(t)$ is monotone in $t$. Recall that $\varepsilon_\infty(\boldsymbol{\theta}) := \rho \operatorname{sign}(\nabla \mathcal{L}(\boldsymbol{\theta}))$. For $\mathcal{D}_{\boldsymbol{\mu}}$, the sign of the gradient (5) is determined coordinate-wise. Thus, the rescaled $\ell_\infty$-SAM flow (2) decouples across coordinates, and each $\beta_j(t)$ evolves independently, allowing us to state Theorem 3.2 for each separate trajectory of $\beta_j(t)$.

*Remark* 3.3 (Interpretation of the Finite-time Blow-up). For $L > 2$, the rescaled $\ell_\infty$-SAM flow (2) exhibits finite-time blow-up: some coordinates satisfy $\beta_j(t) \to \infty$ as $t \to T$. Interpreting this phenomenon in the original SAM time scale, the blow-up corresponds to *infinite time* in the original SAM flow. Indeed, as $\hat{\boldsymbol{\beta}}(t)^\top \boldsymbol{\mu} \to \infty$, we have $\ell'(\hat{\boldsymbol{\beta}}(t)^\top \boldsymbol{\mu}) \to 0^-$, and therefore

$$\tau(t) = \int_0^t -\frac{1}{\ell'(\hat{\boldsymbol{\beta}}(s)^\top \boldsymbol{\mu})} ds \to \infty \quad \text{as} \quad t \to T.$$

Thus, in the original SAM flow, only the coordinates in $J$ diverge as the original time $\tau(t) \to \infty$, while all other coordinates remain bounded.

*Remark* 3.4 (Interpretation of Exponential Growth). For $L = 2$, each coordinate $\beta_j(t)$ with $\alpha_j > \rho$ grows exponentially as $t \to \infty$. Since $\tau(t) \to \infty$ as $t \to \infty$, divergence occurs on the same infinite-time limit in both the rescaled and original $\ell_\infty$-SAM flows. Nevertheless, because the dynamics are obtained after a time reparameterization, the exponential rate observed in the rescaled flow should not be directly interpreted as the actual divergence speed in the original SAM dynamics. Still, for fixed $L = 2$, all coordinates share the same rescaled time, so their relative growth can be compared. Among the coordinates with $\alpha_j > \rho$, the one with the largest feature weight $\mu_j$ dominates asymptotically and the $\ell_\infty$-SAM flow therefore converges in that coordinate direction. We formalize these conclusions for general $L$ in the following corollary, characterizing the dominant direction.

**Corollary 3.5.** *Under the assumptions of Theorem 3.2, let $S := \{j : \alpha_j > \rho\}$ and assume $S \neq \varnothing$. If there is a unique maximizing index $j^* := \arg\max_{j \in S} \mu_j(\alpha_j - \rho)^{L-2}$, then the $\ell_\infty$-SAM flow converges in the $\boldsymbol{e}_{j^*}$ direction. In particular, when $L = 2$, we have $j^* := \arg\max_{j \in S} \mu_j$.*

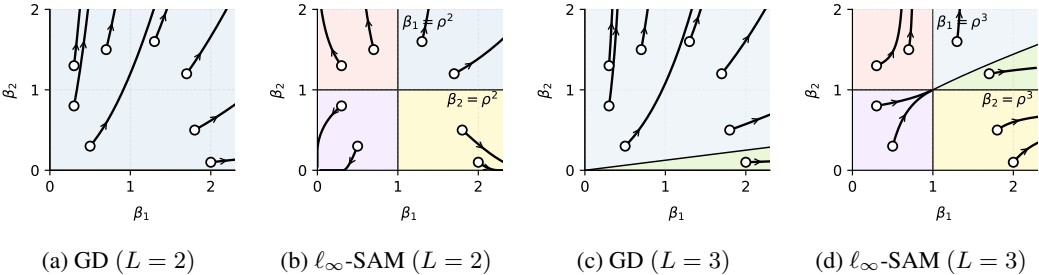

(a) GD ($L = 2$)   (b) $\ell_\infty$-SAM ($L = 2$)   (c) GD ($L = 3$)   (d) $\ell_\infty$-SAM ($L = 3$)

Figure 2: Trajectories $\beta(t)$ from identical initializations under GF and $\ell_\infty$-SAM flow with $d = 2$ and $\boldsymbol{\mu} = (1, 2)$. For SAM, $\rho = 1$.

The proof is deferred to Appendix C.4. When $L = 2$ and $\boldsymbol{\alpha} \in \mathbb{R}^d_{++}$, setting $\rho = 0$ in Corollary 3.5 yields $S = [d]$. Hence, Corollary 3.5 recovers that the GF always aligns in the $\boldsymbol{e}_d$ direction—the $\ell_1$ max-margin direction—regardless of the initialization.

**Illustrative Example.** Figure 2 shows the trajectories of $\boldsymbol{\beta}(t)$ under GF and $\ell_\infty$-SAM flow with $L = 2, 3$ and $\boldsymbol{\mu} = (1, 2)$. Figure 2a depicts the $L = 2$, GF case, where GF always aligns in the $\boldsymbol{e}_2$ direction. For $L = 2$ and $\ell_\infty$-SAM (Figure 2b), the plane $(\beta_1, \beta_2)$ is partitioned by the thresholds $\beta_j = \alpha_j^2 = \rho^2$. If $\alpha_2 > \rho$ (so $2 \in S$), the $\ell_\infty$-SAM flow shows directional convergence in $\boldsymbol{e}_2$ (red/blue regions). In the yellow region, $2 \notin S$ and $1 \in S$, so the limit direction is $\boldsymbol{e}_1$—the "minor" feature. If all coordinates satisfy $\alpha_j < \rho$, the flow converges to $\mathbf{0}$ (purple region), by Theorem 3.2.

For $L > 2$ (Figures 2c and 2d), the blue regions get partitioned once more because large $\alpha_1$ leads to $\mu_1(\alpha_1 - \rho)^{L-2} > \mu_2(\alpha_2 - \rho)^{L-2}$, leading to directional convergence toward $\boldsymbol{e}_1$. Comparing the green regions in Figures 2c and 2d shows that the slope of the boundary between blue and green regions is steeper in $\ell_\infty$-SAM flow than that of GF. Considering that initializations in the yellow region also result in the limit direction $\boldsymbol{e}_1$, these together indicate that $\ell_\infty$-SAM exhibits a greater sensitivity to initialization and stronger implicit bias toward minor features than GD.

## 4 SAM WITH $\ell_2$-PERTURBATIONS: SEQUENTIAL FEATURE AMPLIFICATION

We now turn to $\ell_2$-SAM, which is the form most commonly used in practice.

### 4.1 ASYMPTOTIC BEHAVIOR ON DEPTH-1 AND DEPTH-2 NETWORKS

For depth-1 models, $\ell_2$-SAM converges in the $\ell_2$ max-margin direction regardless of initialization, matching the implicit bias of GD and $\ell_\infty$-SAM. We prove the following theorem in Appendix D.1:

**Theorem 4.1.** *For almost every dataset which is linearly separable, any perturbation radius $\rho$ and any initialization, consider the linear model $f(\boldsymbol{x}) = \langle \boldsymbol{w}, \boldsymbol{x} \rangle$ trained with logistic loss. Then, $\ell_2$-SAM flow directionally converges in the $\ell_2$ max-margin direction.*

While Theorem 4.1 characterizes the limit direction for linearly separable datasets, Theorem D.1 shows that, for the single-example data, the $\ell_2$-SAM flow follows the same trajectory as GF.

For depth-2 models, $\ell_2$-SAM asymptotically converges in the $\ell_1$ max-margin direction as the loss converges to zero, independently of the initialization scale. This parallels the well-known behavior of GD (Gunasekar et al., 2018b). We formalize this below, with the proof in Appendix D.3.

**Theorem 4.2.** *For almost every dataset which is linearly separable, and any perturbation radius $\rho$, consider the linear diagonal network of depth 2, $f(\boldsymbol{x}) = \langle \boldsymbol{w}^{(1)} \odot \boldsymbol{w}^{(2)}, \boldsymbol{x} \rangle$ trained with logistic loss. Let $(\boldsymbol{w}^{(1)}(t), \boldsymbol{w}^{(2)}(t))$ follow the $\ell_2$-SAM flow with $\boldsymbol{w}^{(1)}(0) = \boldsymbol{w}^{(2)}(0)$. Assume (a) the loss vanishes $\mathcal{L}(\boldsymbol{w}^{(1)}(t), \boldsymbol{w}^{(2)}(t)) \to 0$, (b) the predictor $\boldsymbol{\beta}(t) := \boldsymbol{w}^{(1)}(t) \odot \boldsymbol{w}^{(2)}(t)$ converges in direction. Then the limit direction of $\boldsymbol{\beta}(t)$ is the $\ell_1$ max-margin direction.*

Since Theorems 4.1 and 4.2 holds for any $\rho$, it also recovers the implicit bias of GF. We now revisit Figure 6, which is the flow counterpart of Figure 1, and compare the trajectories with the asymptotic directional convergence results above. First, the green lines in Figure 6a visualize the trajectories of $\ell_2$-SAM flow for $L = 1$, and we can check that the trajectories coincide with GD's, as expected by theory. In the $L = 2$ case (Figure 6b), the green $\ell_2$-SAM flow curves include ones that (i) drift

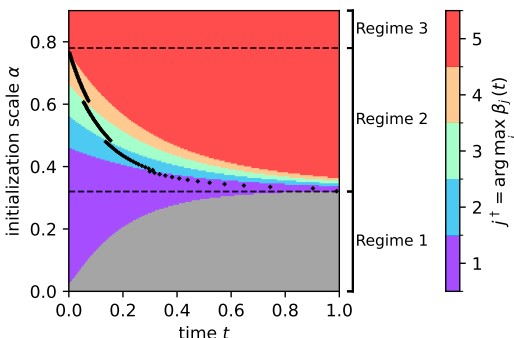 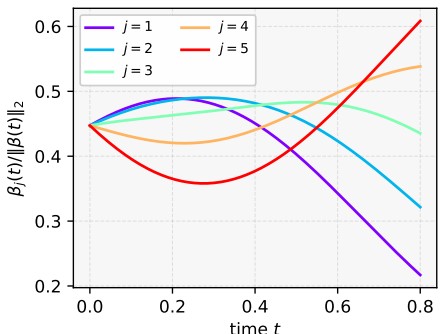

(a) Dominant index $j^\dagger := \arg\max_j \beta_j(t)$ over $\alpha$ and $t$.    (b) Normalized $\beta_j(t)/\|\boldsymbol{\beta}(t)\|_2$ for $\alpha = 0.4$.

Figure 3: Rescaled $\ell_2$-SAM flow on $\mathcal{D}_{\boldsymbol{\mu}}$ with $\boldsymbol{\mu} = (4, 5, 6, 7, 8) \in \mathbb{R}^5$ and $\rho = 1$.

toward the origin, and those that (ii) initially align with $\boldsymbol{e}_1$, a direction *orthogonal* to the $\ell_1$ max-margin direction $\boldsymbol{e}_2$. Such behaviors are not explained by Theorem 4.2. Hence, to account for what is observed in Figure 6b, we move on to analyze the dynamics of $\ell_2$-SAM in finite time.

## 4.2 PRE-ASYMPTOTIC BEHAVIOR ON DEPTH-2 NETWORKS

We investigate the pre-asymptotic dynamics of $\ell_2$-SAM on depth-2 linear diagonal networks and show that the trajectory exhibits a behavior markedly different from its asymptotic limit. This contrast highlights the need for a *finite-time* analysis to understand how the implicit bias of SAM actually emerges. In this section, we study the toy dataset $\mathcal{D}_{\boldsymbol{\mu}} := \{(\boldsymbol{\mu}, +1)\}$ with $\boldsymbol{\mu} \in \mathbb{R}^d$ satisfying $0 < \mu_1 < \cdots < \mu_d$. We further present experiments on multi-point datasets, discrete-time $\ell_2$-SAM, and deeper models ($L \geq 3$) in Appendix D.8, which confirm that the qualitative behaviors identified in the depth-2 single-point $\ell_2$-SAM flow persist in these more realistic settings. Moreover, to capture the effect of the initialization scale with a single parameter, we adopt a coordinate-wise and layer-wise uniform initialization $\boldsymbol{w}^{(1)}(0) = \boldsymbol{w}^{(2)}(0) = \alpha \boldsymbol{1}$ throughout this subsection. We additionally report similar empirical results under random Gaussian initialization in Appendix E.2.

### 4.2.1 SEQUENTIAL FEATURE AMPLIFICATION

We begin by describing a newly observed and surprising phenomenon of $\ell_2$-SAM—**sequential feature amplification**. For certain initialization scales $\alpha$ and times $t$, $\ell_2$-SAM first aligns with minor features; as $t$ increases or as $\alpha$ increases, the dominant coordinate transitions from minor, intermediate to major features. In contrast, GD selects the major feature regardless of $\alpha$ and $t$. We visualize this using rescaled $\ell_2$-SAM flow in Figure 3a and show the GF and $\ell_\infty$-SAM flow counterparts in Figure 7. To quantify the phenomenon along the two axes—time $t$ and initialization scale $\alpha$—at each $t$ and $\alpha$, we track the index $j^\dagger = \arg\max_j \beta_j(t)$ and color the grid $(t, \alpha)$ according to $j^\dagger$. Regions where $\boldsymbol{\beta}$ is negligibly small are shown in gray, indicating convergence to $\boldsymbol{0}$. Based on the observations from Figure 3a, we partition the initialization scale $\alpha$ into three regimes.

**(Regime 1)** Starting from any $\alpha$ in this range, the trajectory eventually collapses to the origin as training proceeds; effectively no feature is expressed and the loss does not vanish.

**(Regime 2)** **Time-wise sequential feature amplification** emerges. With a fixed $\alpha$ chosen from this regime and increasing $t$, there exists the period where the dominant coordinate index $j^\dagger$ increases over time, transitioning from minor to major features. As shown in Figure 3b, $j^\dagger$ sequentially changes from 1 to 5 over time for $\alpha = 0.4$.

**(Regime 3)** $\boldsymbol{\beta}$ aligns with the major feature from the outset and maintains this alignment throughout.

Beyond the time-wise phenomenon, Figure 3a also suggests that sequential feature amplification also happens in the $\alpha$-axis. To see this, consider a fixed slice of time $t$ and navigate through the $\alpha$-axis: for small $\alpha$, the predictor $\boldsymbol{\beta}$ remains near the origin with no feature discovered. As $\alpha$ grows, the dominant coordinate at $t$ shifts sequentially—$\beta_1$ becomes largest first, then $\beta_2$, and so on. However, this is *not* a fair comparison between trajectories, because Figure 3a is obtained from the rescaled flow; each trajectory (for each $\alpha$) has a different time scale.

Nevertheless, we can compare between trajectories if we base our comparison on trajectory-wise maxima. Specifically, in Regime 2 (the sequential feature amplification phase), we define the trajectory-wise most-amplified index, to understand how the initialization scale $\alpha$ affects the "amplification" of minor components. For each coordinate $j$, we track the ratio $\beta_j(t)/\beta_d(t)$ over the entire trajectory, and define $j^*(\alpha) := \arg\max_j \max_t \beta_j(t)/\beta_d(t)$ as the coordinate with the greatest maximum relative amplification. In Figure 3a, for each value of $\alpha$ in Regime 2, we plot the time step that attains the maximum value of $\beta_{j^*(\alpha)}(t)/\beta_d(t)$ in black dots; we can clearly observe that $j^*(\alpha)$ increases from the minor index 1 to second-most major index $d-1$ in Regime 2. We call this phenomenon **initialization-wise sequential feature amplification**.

### 4.2.2 UNDERSTANDING THE EFFECT OF $\ell_2$-SAM

Before analyzing sequential feature amplification, we describe the rescaled $\ell_2$-SAM flow for depth-2 linear diagonal networks and offer an intuitive explanation of the sequential feature amplification phenomenon. With initialization $\boldsymbol{w}^{(1)}(0) = \boldsymbol{w}^{(2)}(0) \in \mathbb{R}_+^d$, we have $\boldsymbol{w}^{(1)}(t) = \boldsymbol{w}^{(2)}(t) =: \boldsymbol{w}(t)$ for all $t \geq 0$. Using this, we derive in Appendix D.2 that the rescaled $\ell_2$-SAM flow for $\boldsymbol{w}(t)$ reads

$$\dot{\boldsymbol{w}}(t) = \boldsymbol{\mu} \odot \left( \boldsymbol{w}(t) - \rho \frac{\boldsymbol{\mu} \odot \boldsymbol{w}(t)}{n_{\boldsymbol{\theta}}(t)} \right), \text{ where } n_{\boldsymbol{\theta}}(t) := \sqrt{2 \|\boldsymbol{\mu} \odot \boldsymbol{w}(t)\|_2^2}. \tag{3}$$

Compared to the $\rho = 0$ case, the extra term scales $\boldsymbol{\mu} \odot \boldsymbol{w}(t)$ coordinate-wise by $1 - \rho \frac{\boldsymbol{\mu}}{n_{\boldsymbol{\theta}}(t)} < 1$. When $n_{\boldsymbol{\theta}}(t)$ is large (e.g., under large initialization or after sufficient training), this factor is close to one and the dynamics becomes close to GF. When $n_{\boldsymbol{\theta}}(t)$ is small (e.g., small initialization), the coordinate-wise scaling factor multiplies different scalars to different coordinates, some of which can even be negative and decrease the corresponding coordinates of $\boldsymbol{w}(t)$. Notice that larger $\mu_j$ leads to smaller $1 - \rho \frac{\mu_j}{n_{\boldsymbol{\theta}}(t)}$. Thus, in the early stage of training, major features are suppressed while minor features are comparatively amplified, yielding the observed emphasis on minor features.

### 4.2.3 ANALYSIS OF TIME-WISE SEQUENTIAL FEATURE AMPLIFICATION

We next provide a theoretical account of the time-wise sequential feature amplification. At each time $t$, we analyze the instantaneous growth rate of each coordinate $\beta_j(t)$, viewed as a function of both $t$ and the initialization scale $\alpha$. This reveals how the growth behavior of different coordinates evolves across the training trajectory. In particular, we derive a coordinate-wise growth rule of $\beta_j(t)$, in a form analogous to Equation (3). The proof is provided in Appendix D.4.3, and an extension to the $L$-layer setting—where an analogous growth rate can be derived—is given in Appendix D.5.

**Lemma 4.3.** *The rescaled $\ell_2$-SAM flow (2) is $\dot{\beta}_j(t) = r_j(t)\beta_j(t)$ with $r_j(t) := 2\mu_j \left(1 - \frac{\rho \mu_j}{n_{\boldsymbol{\theta}}(t)}\right)$.*

By Lemma 4.3, the rate $r_j(t)$ controls the instantaneous growth or decay of $\beta_j(t)$. For fixed $t$, $r_j(t)$ is concave quadratic in $\mu_j$, maximized at $\mu_j = m_c(t) := \frac{n_{\boldsymbol{\theta}}(t)}{2\rho}$. Hence, indices with $\mu_j$ closest to $m_c(t)$ attain the largest $r_j(t)$; **coordinates with feature strength $\mu_j$ nearest to $m_c(t)$ are amplified the most**, while those farther away may even decay. Consequently, the trajectory of $m_c(t)$ dictates the feature-amplification dynamics, and it exhibits three regimes depending on the initialization scale. Recall that $0 < \mu_1 < \cdots < \mu_d$.

**Theorem 4.4.** *There exists a unique $\alpha_1$ such that $\alpha_0 := \rho\frac{\mu_1}{\sqrt{2}\|\boldsymbol{\mu}\|_2} < \alpha_1 < \rho\frac{\|\boldsymbol{\mu}\|_4^4}{\sqrt{2}\|\boldsymbol{\mu}\|_2\|\boldsymbol{\mu}\|_3^3} < \alpha_2 := \rho\frac{\mu_{d-1}+\mu_d}{\sqrt{2}\|\boldsymbol{\mu}\|_2}$ and the trajectory of $m_c(t)$ falls into one of the following three regimes.*

*(Regime 1) If $\alpha < \alpha_1$, then $m_c(t)$ strictly decreases for all $t \geq 0$ and there exists $T_1$ such that for $j \in [d]$, $\beta_j(t)$ strictly decreases for all $t \geq T_1$.*

*(Regime 2) If $\alpha_1 < \alpha < \alpha_2$, there exists $T_2$ such that $m_c(T_2) < \frac{\mu_{d-1}+\mu_d}{2}$ and $m_c(t)$ strictly increases for all $t \geq T_2$.*

*(Regime 3) If $\alpha > \alpha_2$, then $m_c(t) > \frac{\mu_{d-1}+\mu_d}{2}$, and $\beta_d(t)$ has the largest growth rate for all $t \geq 0$.*

The proof of Theorem 4.4 is provided in Appendix D.4.5. Theorem 4.4 identifies three regimes of the $m_c(t)$ dynamics, each corresponding to a qualitatively different pattern of feature amplification.

**Regime 1**. $m_c(t)$ decreases for all $t \geq 0$, and reaches $\frac{\mu_1}{2}$ at time $T_1$. Once $m_c(t) \leq \frac{\mu_1}{2}$, every coordinate satisfies $r_j(t) \leq 0$ by the form of $r_j(t)$, and thus $\beta_j(t)$ strictly decreases for all $j \in [d]$.

**Regime 3.** When $m_c(t) > \frac{\mu_d + \mu_{d-1}}{2}$, the closest feature strength to $m_c(t)$ is $\mu_d$, so $\beta_d(t)$ attains the largest growth rate. This explains why the major feature remains dominant throughout this regime.

**Regime 2.** When $m_c(T_2) < \frac{\mu_d + \mu_{d-1}}{2}$, the closest index $j_c$ satisfies $j_c < d$. At this time, the largest growth rate is therefore achieved by the non-major coordinate $\beta_{j_c}(T_2)$. Since $m_c(t)$ strictly increases for all $t \geq T_2$, the coordinate with the largest growth rate increases, exhibiting the *time-wise sequential feature amplification* observed empirically in Section 4.2.1. In Regime 2, there also exist instances where $m_c(t)$ initially *decreases* and later increases, leading to a *non-monotonic* sequential feature amplification phenomenon. We discuss this in Appendix A.5.

Regime 2 also leaves a clear trace in the training loss. $\ell_2$-SAM exhibits an early plateau while it mainly amplifies minor coordinates, and the loss drops quickly only after it shifts to major coordinates, whereas GD shows a steadier decrease without this minor-to-major transition. The corresponding loss curves and further explanation are given in Figure 4 and Appendix E.1.

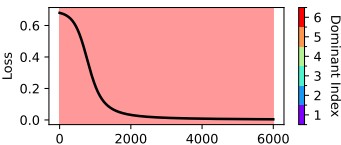 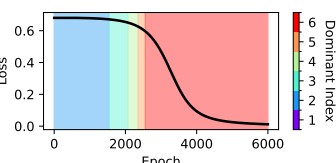

Figure 4: Loss curves of GD (left) and $\ell_2$-SAM (right) on a 2-layer diagonal network in Regime 2 ($\alpha = 0.35$, $\mu = (1, 2, 3, 4, 5, 6)$, $\rho = 0.1$). Colored regions mark the coordinate with highest growth.

### 4.2.4 ANALYSIS OF INITIALIZATION-WISE SEQUENTIAL FEATURE AMPLIFICATION

In the previous subsection, we examined which coordinate attains the maximal instantaneous growth rate. We now turn to the cumulative update over time and study initialization-wise sequential feature amplification. In Theorem 4.4, we characterize the range of $\alpha$ (Regime 2) in which sequential feature amplification can occur. Here, we quantify the strength of amplification within Regime 2 as a function of $\alpha$. Since a coordinate $\beta_j(t)$ can diverge, we assess which feature is amplified—and by how much—via the ratio of the $j$-th feature to the major feature, $\beta_j(t)/\beta_d(t)$. For a given initialization scale $\alpha$, we track and bound how large the amplification ratio $\beta_j(t)/\beta_d(t)$ can be along the trajectory.

Integrating the rescaled $\ell_2$-SAM flow (3) (derived in Appendix D.6.1) yields the coordinate ODE

$$\beta_j(t) = \beta_j(0) \exp\left(2\mu_j t - 2\rho\mu_j^2 I(t)\right) \quad \text{where } I(t) := \int_0^t \frac{1}{n_{\boldsymbol{\theta}}(s)} ds \qquad \text{for } j \in [d]. \quad (4)$$

The behavior of $\boldsymbol{\beta}$ in (4) is determined by $I(t)$. Recall that $n_{\boldsymbol{\theta}}(t)$ controls the behavior of $\ell_2$-SAM in Section 4.2.2 and is used to characterize the instantaneous growth rate in Section 4.2.3. Here, we focus on cumulative updates over time, where the time integral $I(t)$ of $1/n_{\boldsymbol{\theta}}$ becomes decisive. By bounding $I(t)$, we quantify how strongly each feature is amplified relative to the major feature.

**Theorem 4.5.** *Let $\alpha_0, \alpha_2$ be defined in Theorem 4.4 and $\alpha_1$ be the threshold from there. Suppose $\alpha_1 < \alpha \leq \rho\frac{\mu_1 + \mu_d}{\sqrt{2}\|\boldsymbol{\mu}\|_2} < \alpha_2$. Then, for $j \in [d]$, there exists $T_j$ such that*

$$\frac{\beta_j(T_j)}{\beta_d(T_j)} \geq \mathrm{LB}_j(\alpha) := \exp\left(2R'_j\left((R_j - 1)\log\left(\frac{1}{1-\alpha_0/\alpha}\right) + \log\left(\frac{1}{\alpha_0/\alpha}\right) - C(R_j)\right)\right)$$

*where $R_j := (\mu_j + \mu_d)/\mu_1 > 2$, $R'_j := (\mu_d - \mu_j)/\mu_1$ and $C(R) := R\log R - (R-1)\log(R-1)$.*

The proof follows from a lower bound on $I(t)$, and is deferred to Appendix D.6.2. A numerical illustration of $\mathrm{LB}_j(\alpha)$ for several choices of $\boldsymbol{\mu}$ is provided in Appendix D.7. Theorem 4.5 applies to the small-$\alpha$ portion of Regime 2. For each coordinate $j$, we select the time $T_j$ maximizing $\frac{\beta_j(t)}{\beta_d(t)}$ over the entire trajectory, and obtain a nontrivial lower bound $\mathrm{LB}_j(\alpha)$ for this maximal amplification.

The theorem goes beyond the qualitative picture in Figure 3a, which only identifies which coordinate becomes dominant (the index $j^\dagger$). Theorem 4.5 additionally quantifies *how large* this dominant coordinate must grow: as shown in Appendix D.7, $\mathrm{LB}_j(\alpha)$ often exceeds 10, indicating that the minor to intermediate coordinates can take values more than ten times larger than the major coordinate.

**Dependence on $\alpha$.** For all $\alpha$ in Regime 2, the ratio $\alpha_0/\alpha$ lies in $(0, 1)$, so both logarithmic terms in $\mathrm{LB}_j(\alpha)$ are positive. Since $R_j > 2$, the first logarithmic term dominates the exponent, making $\mathrm{LB}_j(\alpha)$ grow rapidly as $\alpha \to \alpha_1$. Thus smaller $\alpha$ in Regime 2 produces stronger amplification as

shown in Appendix D.7. This is substantiated by Figure 3a: smaller $\alpha$ in Regime 2 keeps the dynamics aligned with minor-intermediate features for a longer time $t$, leading to greater amplification.

**Dependence on Feature Geometry.** The coefficients $R_j$ and $R'_j$ increase with the spectral gap $\mu_d/\mu_1$, so datasets with larger feature contrast amplify more strongly as shown in Appendix D.7.

Since $\mathrm{LB}_j(\alpha)$ varies across $j$, it is natural to ask which coordinate experiences the strongest amplification. Proposition 4.6 identifies the maximizing index $j^*(\alpha)$, with the proof in Appendix D.6.3.

**Proposition 4.6.** *Under the conditions of Theorem 4.5, define $j^*(\alpha) := \arg\max_{j \in [d]} \mathrm{LB}_j(\alpha)$ and set $\alpha_0^* := \alpha_0$. Then, there exist thresholds $\alpha_0^* < \alpha_1^* < \cdots < \alpha_m^* \le \rho\frac{\mu_1 + \mu_d}{\sqrt{2}\|\boldsymbol{\mu}\|_2}$ for some $m \le d-1$ such that $j^*(\alpha) = j$ for $\alpha \in (\alpha_{j-1}^*, \alpha_j^*]$.*

Proposition 4.6 shows $j^*(\alpha)$ monotonically increases sequentially from 1 to $m$ on $\alpha \in (\alpha_0, \alpha_m^*]$. Namely, as the initialization scale $\alpha$ grows, the index that maximizes the lower bound $\mathrm{LB}_j(\alpha)$ shifts monotonically from minor to intermediate features. This matches the *initialization-wise sequential feature amplification* discussed in Section 4.2.1 (i.e., the black dots in Figure 3a). Within Regime 2, our theoretical bound predicts a progression of the most-amplified coordinate from 1 to $m$.

Lastly, through the cumulative update analysis, we characterize the asymptotic behavior of $\ell_2$-SAM flow for some extreme ranges of $\alpha$. We prove the following proposition in Appendix D.6.4.

**Proposition 4.7.** *Consider $\alpha_0$ defined in Theorem 4.4. (i) If $\alpha < \alpha_0$, then $\boldsymbol{\beta}(t)$ converges to zero. (ii) If $\alpha > \rho\frac{\|\boldsymbol{\mu}\|_2^2}{\sqrt{2d}(\prod_{i=1}^d \mu_i)^{1/d}\|\boldsymbol{\mu}\|_1}$, then $\boldsymbol{\beta}(t)$ converge in $\ell_1$ max-margin direction.*

Recall that Theorem 4.2 assumes that the loss vanishes and the limit direction exists. Proposition 4.7(i) shows that for small $\alpha$ in Regime 1, the loss never vanishes. Proposition 4.7(ii) shows that for some $\alpha$'s in Regimes 2 or 3, the limit direction exists and is the $\ell_1$ max-margin direction.

## 5 EXPERIMENTS

Our investigation shows how depth, perturbation geometry, and initialization jointly shape SAM's optimization trajectory. We substantiate these findings with controlled experiments: 2-layer CNNs and linear networks on synthetic banded data, where we systematically vary the dataset construction and metrics across architectures (Appendix E.3), as well as multi-point (Appendix D.8.2) and deeper diagonal models (Appendix D.8.3). We also present experiments with practical CNNs trained on MNIST, where we use Grad-CAM (Selvaraju et al., 2017) to visualize which image pixels are emphasized (Figure 5 and Appendix E.4). These experiments show that $\ell_2$-SAM places relatively greater emphasis on weaker/background pixels than GD, qualitatively matching our theory.

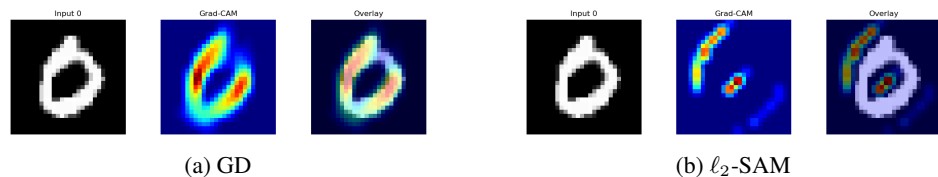

|  |  |
|:-:|:-:|
| (a) GD | (b) $\ell_2$-SAM |

Figure 5: Grad-CAM comparison of GD and $\ell_2$-SAM on a CNN trained on MNIST. GD focuses on dominant digit pixels, whereas $\ell_2$-SAM highlights minor background regions.

## 6 CONCLUSION

We characterized how network depth changes SAM's implicit bias on linear diagonal networks. For depth 1, SAM preserves GD's implicit bias. For deeper networks ($L \ge 2$) with $\ell_\infty$-SAM, we derived precise weight trajectories depending on initialization scale and perturbation radius, where each weight coordinate either diverges toward a standard basis vector or converges to a finite point. The most interesting regime arises for $L = 2$ with $\ell_2$-SAM: while the limit direction converges to the $\ell_1$ max-margin solution, the finite-time dynamics exhibit *sequential feature amplification*, where the predictor initially relies on minor coordinates and gradually shifts to larger ones. These observations suggest that implicit bias statements made only in the $t \to \infty$ limit can overlook how the bias emerges, motivating a *finite-time* perspective.

## ETHICS STATEMENT

This work is purely theoretical, analyzing the optimization dynamics and implicit bias of SAM in simplified models. It does not involve human subjects, personal data, or sensitive information, and introduces no new datasets. Broader impacts are indirect: the results may inform more reliable training and diagnosis of SAM-like methods, but they do not by themselves address safety, fairness, or deployment risks, which must be evaluated in application-specific settings.

## REPRODUCIBILITY STATEMENT

We support the reproducibility of our results by (1) fully specifying the SAM variants and rescaled-flow dynamics, together with complete theoretical statements and proofs; (2) reporting exact experimental setups, including the initialization scheme, models, datasets, and the values of step sizes, perturbation radius, and initialization scale; and (3) computing several quantities used in our theoretical simulations in closed form, making the corresponding plots exactly reproducible.

## ACKNOWLEDGMENT

We thank Junsoo Oh for the helpful discussions. This work was supported by a National Research Foundation of Korea (NRF) grant funded by the Korean government (MSIT) (No. RS-2023-00211352) and an Institute of Information & communications Technology Planning & Evaluation (IITP) grant (No. RS-2019-II190075, Artificial Intelligence Graduate School Program (KAIST)) funded by the Korean government (MSIT).

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

## CONTENTS

## DECLARATION OF LLM USAGE

We used Large Language Models (LLMs) solely to aid or polish writing. They did not generate ideas, analyses, or conclusions. All LLM-assisted text was reviewed and edited by the authors.

## A FIGURES AND DISCUSSIONS OMITTED FROM MAIN TEXT

### A.1 FLOW TRAJECTORIES OF GD AND SAM

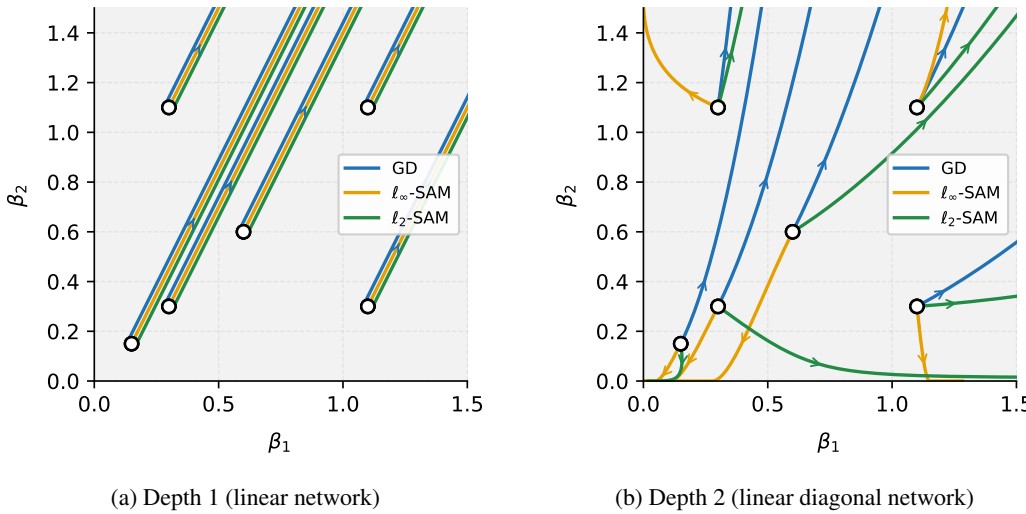

(a) Depth 1 (linear network)

(b) Depth 2 (linear diagonal network)

Figure 6: Trajectories of the predictor $\boldsymbol{\beta}(t) \in \mathbb{R}^2$ from identical initial conditions under GF, $\ell_\infty$-SAM flow and $\ell_2$-SAM flow on $\{(\boldsymbol{\mu}, +1)\}$ with $\boldsymbol{\mu} = (1, 2)$. For SAM, $\rho = 1$.

### A.2 MORE DISCUSSION ON RELATED WORK

#### A.2.1 RECENT WORK ON IMPLICIT BIAS IN DIAGONAL LINEAR NETWORKS

Jacobs & Burkholz (2024) study continuous sparsification with time-varying weight decay, formulating a time-dependent Bregman potential that causes the implicit bias to evolve from $\ell_2$- to $\ell_1$-type behavior over the course of training. Wang & Klabjan (2024) study smoothed sign descent on a quadratically parameterized regression problem, introducing a time-varying mirror map. and prove that the resulting limit point is an approximate KKT point of a Bregman-divergence–style objective, where the stability constant $\varepsilon$ quantifies the gap to KKT optimality. Papazov et al. (2024) analyze momentum gradient descent on diagonal linear network through a momentum gradient flow, showing that a newly defined intrinsic parameter determines the optimization trajectory and admits a second order, time-varying mirror-flow formulation. Within this framework, they characterize the induced implicit regularization and demonstrate that smaller values of this intrinsic parameter yield more balanced weights and sparser solutions compared to standard gradient flow. Jacobs et al. (2025) extend the mirror flow framework to account for explicit regularization and analyze the evolution of the corresponding Legendre function over time, thereby describing how the implicit bias changes in different reparameterizations, including diagonal linear networks. In particular, they track how the implicit bias evolves in terms of its positional bias, bias type, and range shrinking.

#### A.2.2 COMPARISON WITH SADDLE-TO-SADDLE DYNAMICS

In this section, we provide further details on the relation between our work and the saddle-to-saddle dynamics of gradient descent/flow. Pesme & Flammarion (2023) consider diagonal linear networks trained with squared loss in the infinitesimal-initialization limit. In this regime, gradient flow exhibits incremental, stage-wise learning: the flow undergoes long plateaus near a saddle whose predictor is supported on the first $k$ coordinates, then escapes along a low-dimensional "fast escape"

manifold to a saddle with support on $k+1$ coordinates, and so on. Sequentiality thus appears as *discrete* transitions between saddles with support size $k$ and $k+1$. In the diagonal setting, complexity is captured by the number of active coordinates, which is constant on each plateau and changes only at these transition times.

In contrast, our work on the sequential feature amplification focuses on a linear diagonal *classifier* trained with $\ell_2$-SAM and logistic loss, and on a different notion of complexity: individual coordinates (features) ordered by the strength of the teacher signal, from minor to major features. In our setting, all coordinates are present from the beginning. Instead of coordinate jumps, we track how the coordinate-wise alignments and margins evolve both over time and as a function of the initialization scale, where by "alignment" we mean the magnitude of the predictor at each coordinate, indicating how strongly the predictor attends to each feature. We show that $\ell_2$-SAM gives rise to two complementary forms of sequential feature amplification: (i) a *time-wise* ordering, where alignment with minor features is relatively amplified earlier in training and gradually shifts toward major features; and (ii) an *initialization-scale-wise* ordering, where the most-amplified feature over a finite training process changes systematically with the initialization scale. In both views, the ordering emerges through a *continuous* evolution of the alignment across coordinates, and sequentiality is captured by which feature is currently most amplified, rather than by discrete activation or deactivation of features.

The mechanisms underlying these two phenomena are conceptually distinct. First, saddle-to-saddle dynamics start from the zero vector and involve successive coordinate *activations*, where previously inactive coordinates become active over time. Our setting, by contrast, starts from $\alpha\mathbf{1}$ (without taking the limit $\alpha \to 0$), where all coordinates are already active, and the dynamics involve successive *amplification* of already-active coordinates. Activation and amplification are fundamentally different: even if saddle-to-saddle dynamics exhibit successive activation, the identity of the most dominant coordinate can remain unchanged, unlike in our setting where dominance itself shifts over time.

Second, the ordering principles differ. In our work, the ordering of amplified coordinates is driven directly by the data geometry, namely the ordering of the signal strengths $\mu_j$. In saddle-to-saddle dynamics, the progression is governed by a dual-thresholding mechanism, tied to when integrated gradients hit constraint boundaries, and does not correspond to a minor-to-major feature progression.

Third, the role of initialization is opposite. Saddle-to-saddle dynamics arise in the vanishing-initialization limit ($\alpha \to 0$). In contrast, we observe sequential feature amplification across a wide range of non-vanishing initialization scales, and in fact show that increasing $\alpha$ induces a clear and systematic amplification ordering. Our phenomenon is therefore not a small-initialization effect.

Fourth, saddle points play no constructive role in our mechanism. Aside from the trivial effect that extremely small initialization can prevent SAM trajectories from escaping the origin, saddle points do not drive the sequential feature amplification we characterize. The observed dynamics are not mediated by saddle escape.

Finally, the problem setups are fundamentally different. Prior saddle-to-saddle works analyze regression under squared loss, whereas our work studies classification under logistic loss, where the optimization landscape and asymptotic behavior are qualitatively different.

Taken together, these observations indicate that sequential feature amplification is a SAM-specific phenomenon, distinct from known saddle-to-saddle or incremental learning dynamics, and does not arise under conventional gradient descent.

### A.2.3 PROPERTIES OF SAM

SAM seeks flatter minima through a two-step update procedure that first perturbs parameters in the direction of steepest ascent before computing the gradient update. This unique optimization strategy has led to the discovery of several distinctive dynamical properties that differ fundamentally from standard gradient descent.

SAM exhibits distinctive valley-bouncing dynamics (Bartlett et al., 2022; Wen et al., 2022). Bartlett et al. (2022) demonstrated that SAM oscillates between ravines in quadratic loss landscapes, with update directions naturally aligning with the dominant eigenvector of the Hessian matrix. Building on this geometric intuition, Wen et al. (2022) proved that SAM implicitly follows trajectories that

minimize the maximum eigenvalue of the Hessian, providing a precise characterization of SAM's eigenvalue regularization effect.

However, SAM's convergence properties present both opportunities and challenges. Si & Yun (2023) showed that under practical settings, SAM can struggle to converge to local minima, while Kim et al. (2023) identified specific instabilities in SAM dynamics near saddle points. These findings highlight the delicate balance between SAM's beneficial regularization effects and potential optimization difficulties.

Andriushchenko et al. (2023) demonstrated that SAM dynamics drive networks toward low-rank feature representations by systematically pruning activations, providing mechanistic insights into how SAM shapes learned representations. Dai et al. (2023) showed that normalization terms in SAM updates play a crucial role in stabilizing training dynamics and preventing gradient vanishing, highlighting the importance of architectural components in SAM's effectiveness. Compagnoni et al. (2023) derived continuous-time stochastic differential equations for SAM, proving that the dynamics are equivalent to SGD on an implicitly regularized loss with Hessian-dependent noise, thereby connecting SAM to principled stochastic optimization theory. Vani et al. (2024) argued that SAM's ascent perturbation mechanism systematically discards output-exposed biases, offering a perspective on how SAM's perturbation strategy contributes to improved generalization.

Previous works (Andriushchenko & Flammarion, 2022; Clara et al., 2025) have studied SAM's implicit bias in diagonal linear networks. Andriushchenko & Flammarion (2022) analyze 2-layer linear diagonal networks under sparse regression with MSE loss, showing SAM induces better sparsity than gradient descent, but require the small-$\rho$ assumption. Clara et al. (2025) study SAM dynamics with noise, proving weight balancing across layers and sharpness minimization, also limited to MSE loss. Our analysis removes the small-$\rho$ assumption to capture the full perturbation effect and studies logistic loss, revealing distinct implicit bias properties compared to the squared loss setting.

Kim et al. (2026) empirically observe that SAM is more capable than SGD of capturing atypical subpatterns, while SGD relies more heavily on majority features. Our result provides a complementary mechanistic view: in diagonal linear networks, SAM can amplify minor coordinates differently from GD during finite-time training, even though the two methods share the same asymptotic implicit bias. This suggests that SAM's improved generalization may partly arise from its ability to discover non-majority but informative features.

### A.2.4 DEPTH-INDUCED IMPLICIT BIAS

A line of work shows that increasing depth in overparameterized linearized models can qualitatively change the implicit bias of gradient-based training dynamics (Vardi, 2023). In matrix problems, deep matrix factorization exhibits an enhanced tendency toward low-rank solutions as depth increases (Arora et al., 2019; Chou et al., 2024), and recent work further connects depth to low-rank bias and loss of plasticity in matrix completion (Shin & Yun, 2026). Relatedly, depth can also induce incremental learning dynamics and sparsity-promoting behavior in simplified models (Gissin et al., 2019). Our results complement these findings by isolating a depth-dependent implicit bias mechanism specific to SAM in diagonal linear networks.

### A.3 DERIVATION OF RESCALED $\ell_p$-SAM FLOW

For the dataset $\{(\boldsymbol{\mu}, +1)\}$, the loss function is given as:

$$\mathcal{L}(\boldsymbol{\theta}) = \ell(\langle \boldsymbol{\beta}(\boldsymbol{\theta}), \boldsymbol{\mu} \rangle).$$

For each $i \in [L]$, the gradient is

$$\nabla_{\boldsymbol{w}^{(i)}} \mathcal{L}(\boldsymbol{\theta}) = \ell'\big(\langle \boldsymbol{\beta}(\boldsymbol{\theta}), \boldsymbol{\mu} \rangle\big) \nabla_{\boldsymbol{w}^{(i)}} \langle \boldsymbol{\beta}(\boldsymbol{\theta}), \boldsymbol{\mu} \rangle = \ell'\big(\langle \boldsymbol{\beta}(\boldsymbol{\theta}), \boldsymbol{\mu} \rangle\big) \boldsymbol{\mu} \odot \Big( \bigodot_{\ell \neq i} \boldsymbol{w}^{(\ell)} \Big). \tag{5}$$

Then, we have the $\ell_p$-SAM flow of $\boldsymbol{w}^{(i)}$ as

$$\dot{\boldsymbol{w}}^{(i)}(t) = -\nabla_{\boldsymbol{w}^{(i)}} \mathcal{L}(\hat{\boldsymbol{\theta}}(t)) = -\ell'\big(\langle \boldsymbol{\beta}(\hat{\boldsymbol{\theta}}(t)), \boldsymbol{\mu} \rangle\big) \boldsymbol{\mu} \odot \Big( \bigodot_{\ell \neq i} \hat{\boldsymbol{w}}^{(\ell)}(t) \Big).$$

Since $\ell'(u) = -\frac{1}{1+\exp(u)} < 0$, it has the same spatial trajectory (up to reparameterization of time):

$$\dot{\boldsymbol{w}}^{(i)}(t) = \boldsymbol{\mu} \odot \left( \bigodot_{\ell \neq i} \hat{\boldsymbol{w}}^{(\ell)}(t) \right) = \boldsymbol{\mu} \odot \left( \bigodot_{\ell \neq i} \left( \boldsymbol{w}^{(\ell)}(t) + \boldsymbol{\varepsilon}_p^{(\ell)}(\boldsymbol{\theta}(t)) \right) \right).$$

This derivation works for any $p$, not just $p = 2$ and $p = \infty$.

## A.4 GD AND $\ell_\infty$-SAM DO NOT EXHIBIT SEQUENTIAL FEATURE AMPLIFICATION

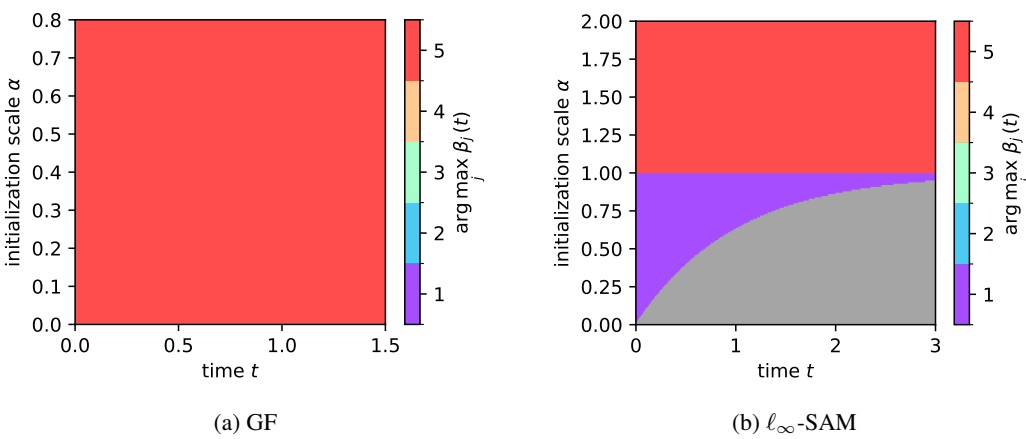

(a) GF

(b) $\ell_\infty$-SAM

Figure 7: Dominant index $j^\dagger := \arg\max_j \beta_j(t)$ for GF and $\ell_\infty$-SAM flow over $(t, \alpha)$ on $\mathcal{D}_{\boldsymbol{\mu}}$ with $\boldsymbol{\mu} = (4, 5, 6, 7, 8) \in \mathbb{R}^5$.

## A.5 INTERESTING TRAJECTORY IN REGIME 2 OF THEOREM 4.4

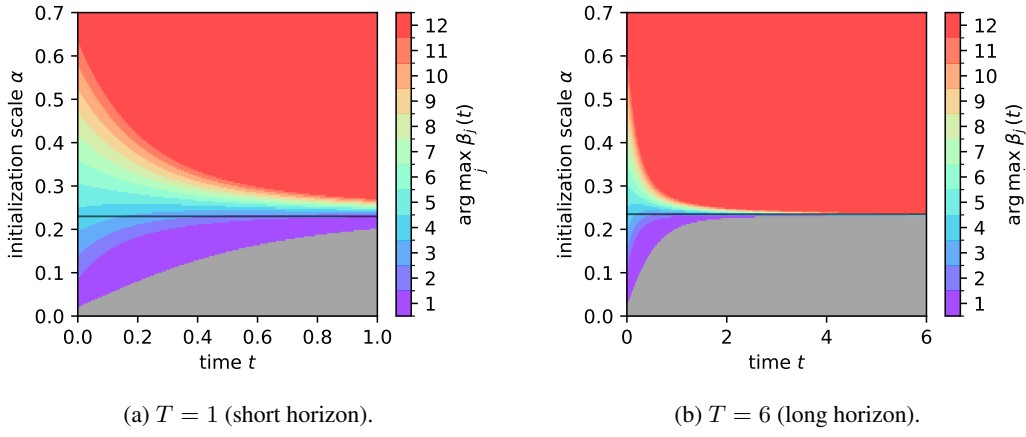

(a) $T = 1$ (short horizon).

(b) $T = 6$ (long horizon).

Figure 8: Dominant index for $\ell_2$-SAM flow with $\boldsymbol{\mu} = (1, 2, \ldots, 12)$. The black line indicates the interesting trajectory.

In Regime 2 of Theorem 4.4, there is also an interesting sub-regime that corresponds to smaller values of $\alpha$ with the range of Regime 2. Define a critical threshold $\alpha_{\mathrm{crit}} := \frac{\rho \|\boldsymbol{\mu}\|_4^4}{\sqrt{2} \|\boldsymbol{\mu}\|_2 \|\boldsymbol{\mu}\|_3^3} \in (\alpha_1, \alpha_2)$. When $\alpha_1 < \alpha < \alpha_{\mathrm{crit}}$, the trajectory $m_c(t)$ initially decreases to a minimum above $\frac{\mu_1}{2}$ and then increases. During this decreasing phase, the $\ell_2$-SAM flow amplifies coordinates with smaller indices $j < j_c(0)$ than the most-amplified index at initialization $j_c(0) \in \arg\min_j |\mu_j - m_c(0)|$, enabling an aggressive exploration of weaker features before transitioning to the standard minor-first-major-last sequential amplification pattern. Along the black path in Figure 8, this manifests as the most-

amplified coordinate starting at $\beta_4$, then stepping down to $\beta_1$ sequentially during the initial decrease, and—after sufficient time—stepping back up sequentially toward $\beta_d$ as $m_{\mathrm{c}}(t)$ increases.

# B  CORE LEMMA FOR SAM ON DEPTH-1 NETWORKS

Although our argument is inspired by the simple proof of Theorem 9 in Soudry et al. (2018), extending that analysis from gradient descent to the SAM flow is far from straightforward. In GD the gradient has a clean exponential form and all coefficients are fixed, which makes the support/non-support decomposition almost immediate.

In contrast, SAM evaluates the gradient at the perturbed point $\hat{w}(t)$, introducing the time–dependent factors $\gamma_n(t)$ and the perturbed margins $\widehat{m}_n(t)$, neither of which appear in GD. Controlling these additional terms turns out to be technically delicate: one must show that the SAM-induced coefficients remain uniformly bounded, that the perturbed margins stay within a fixed range, and that the resulting two-variable function $\psi(z, \delta)$ admits a uniform upper bound. Only after establishing these new ingredients can the GD-style argument be recovered. The proof below develops these steps and shows that, despite the additional complexity, the SAM flow converges to the same $\ell_2$ max-margin direction as GD.

**Lemma B.1.** *For almost every dataset which is linearly separable, any perturbation radius $\rho$ and any initialization, consider the linear model $f(x) = \langle w, x \rangle$ trained with logistic loss. For any SAM perturbation of the form*

$$\hat{w} = w + \varepsilon(w)$$

*with a perturbation direction $\varepsilon(w)$ satisfying*

$$\|\varepsilon(w)\|_2 \leq B \quad \text{for some finite constant } B < \infty \text{ and all } w,$$

*the resulting SAM flow converges in $\ell_2$ max-margin direction.*

*Proof.* Let $\{(x_n, y_n)\}_{n=1}^N \subset \mathbb{R}^d \times \{\pm 1\}$ be a linearly separable dataset, that is, there exists a vector $w_*$ such that

$$y_n\, x_n^\top w_* > 0 \quad \text{for all } n.$$

As usual in this setting, we absorb the labels into the inputs and assume without loss of generality that all labels are $y_n = 1$. In other words, we redefine $x_n \leftarrow y_n x_n$ and work with a dataset $\{x_n\}_{n=1}^N$ such that

$$\exists w_* \text{ with } x_n^\top w_* > 0 \quad \text{for all } n.$$

For the linear model $f(x) = x^\top w$, the logistic loss is

$$\mathcal{L}(w) = \sum_{n=1}^N \ell(x_n^\top w), \qquad \ell(u) = \log(1 + e^{-u}), \qquad \ell'(u) = -\frac{e^{-u}}{1 + e^{-u}}.$$

The SAM flow with perturbation $\varepsilon(w)$ is the gradient flow

$$\dot{w}(t) = -\nabla\mathcal{L}(\widehat{w}(t)), \qquad \widehat{w}(t) = w(t) + \varepsilon(w). \tag{6}$$

Let $m_n(t) = x_n^\top w(t)$ and $\widehat{m}_n(t) = x_n^\top \widehat{w}(t)$. Then

$$\nabla\mathcal{L}(\widehat{w}(t)) = -\sum_{n=1}^N \frac{e^{-\widehat{m}_n(t)}}{1 + e^{-\widehat{m}_n(t)}}\, x_n = -\sum_{n=1}^N \gamma_n(t) e^{-m_n(t)} x_n,$$

with

$$\gamma_n(t) = \frac{e^{-(\widehat{m}_n(t) - m_n(t))}}{1 + e^{-\widehat{m}_n(t)}} \geq 0.$$

Because $\widehat{w}(t) - w(t) = \varepsilon(w(t))$ and $\|\varepsilon(w(t))\|_2 \leq B$, if the data are bounded, say $\|x_n\|_2 \leq R$, then

$$|\widehat{m}_n(t) - m_n(t)| = |x_n^\top(\widehat{w}(t) - w(t))| \leq BR =: C \tag{7}$$

for all $n, t$. Hence there is a constant $A > 0$ such that

$$0 \leq \gamma_n(t) \leq A \quad \text{for all } n, t.$$

The SAM flow equation 6 can therefore be written as

$$\dot{\boldsymbol{w}}(t) = \sum_{n=1}^{N} \gamma_n(t) e^{-m_n(t)} \boldsymbol{x}_n, \qquad 0 \le \gamma_n(t) \le A. \tag{8}$$

Let $\boldsymbol{w}^*$ denote the $\ell_2$ max-margin solution

$$\boldsymbol{w}^* = \arg\min_{\boldsymbol{w}} \|\boldsymbol{w}\|_2 \quad \text{s.t.} \quad \boldsymbol{x}_n^\top \boldsymbol{w} \ge 1 \text{ for all } n.$$

Let $S = \{n : \boldsymbol{x}_n^\top \boldsymbol{w}^* = 1\}$ be the support set. Standard KKT conditions yield coefficients $b_n > 0$ for $n \in S$ with $\sum_{n \in S} b_n = 1$ such that

$$\boldsymbol{w}^* = \sum_{n \in S} b_n \boldsymbol{x}_n.$$

Define the residual

$$\boldsymbol{r}(t) = \boldsymbol{w}(t) - \boldsymbol{w}^* \log t.$$

Our goal is to show that $\boldsymbol{r}(t)$ is bounded. This will imply that

$$\frac{\boldsymbol{w}(t)}{\|\boldsymbol{w}(t)\|} = \frac{\boldsymbol{w}^* \log t + \boldsymbol{r}(t)}{\|\boldsymbol{w}^*\| \log t + o(\log t)} \to \frac{\boldsymbol{w}^*}{\|\boldsymbol{w}^*\|},$$

that is, the SAM flow converges in the $\ell_2$ max-margin direction.

Differentiating and substituting equation 8, we obtain

$$\dot{\boldsymbol{r}}(t) = \dot{\boldsymbol{w}}(t) - \frac{\boldsymbol{w}^*}{t} = \sum_{n=1}^{N} \gamma_n(t) e^{-m_n(t)} \boldsymbol{x}_n - \frac{\boldsymbol{w}^*}{t}.$$

We split the sum over the support and non-support points:

$$\dot{\boldsymbol{r}}(t) = \sum_{n \in S} \gamma_n(t) e^{-m_n(t)} \boldsymbol{x}_n + \sum_{n \notin S} \gamma_n(t) e^{-m_n(t)} \boldsymbol{x}_n - \frac{\boldsymbol{w}^*}{t}.$$

For $n \in S$ we have $\boldsymbol{x}_n^\top \boldsymbol{w}^* = 1$, so

$$m_n(t) = \boldsymbol{x}_n^\top \boldsymbol{w}(t) = \boldsymbol{x}_n^\top \boldsymbol{w}^* \log t + \boldsymbol{x}_n^\top \boldsymbol{r}(t) = \log t + \boldsymbol{x}_n^\top \boldsymbol{r}(t),$$

and therefore

$$t e^{-m_n(t)} = e^{-\boldsymbol{x}_n^\top \boldsymbol{r}(t)}.$$

For $n \notin S$ we have

$$e^{-m_n(t)} = e^{-\boldsymbol{x}_n^\top \boldsymbol{w}^* \log t - \boldsymbol{x}_n^\top \boldsymbol{r}(t)} = t^{-\boldsymbol{x}_n^\top \boldsymbol{w}^*} e^{-\boldsymbol{x}_n^\top \boldsymbol{r}(t)}.$$

Using $\boldsymbol{w}^* = \sum_{n \in S} b_n \boldsymbol{x}_n$ we rewrite

$$\dot{\boldsymbol{r}}(t) = \frac{1}{t} \sum_{n \in S} b_n \Big[ \frac{\gamma_n(t)}{b_n} e^{-\boldsymbol{x}_n^\top \boldsymbol{r}(t)} - 1 \Big] \boldsymbol{x}_n + \sum_{n \notin S} \gamma_n(t) t^{-\boldsymbol{x}_n^\top \boldsymbol{w}^*} e^{-\boldsymbol{x}_n^\top \boldsymbol{r}(t)} \boldsymbol{x}_n. \tag{9}$$

Consider the squared norm:

$$\frac{1}{2} \frac{d}{dt} \|\boldsymbol{r}(t)\|^2 = \boldsymbol{r}(t)^\top \dot{\boldsymbol{r}}(t) = T_1(t) + T_2(t),$$

where $T_1(t)$ and $T_2(t)$ are the contributions of the two terms in equation 9. For the non-support term $T_2(t)$ in equation 9, we have

$$T_2(t) = \sum_{n \notin S} \gamma_n(t) t^{-\boldsymbol{x}_n^\top \boldsymbol{w}^*} e^{-\boldsymbol{x}_n^\top \boldsymbol{r}(t)} \boldsymbol{x}_n^\top \boldsymbol{r}(t).$$

There is a margin gap $\theta > 0$ such that $\boldsymbol{x}_n^\top \boldsymbol{w}^* \ge 1 + \theta$ when $n \notin S$. Then

$$t^{-\boldsymbol{x}_n^\top \boldsymbol{w}^*} \le t^{-(1+\theta)},$$

and using $\gamma_n(t) \le A$ and $\forall z \; e^{-z} z \le 1$, we have

$$T_2(t) \le \frac{A}{t^{1+\theta}}.$$

For the support points, write $z_n(t) = \boldsymbol{x}_n^\top \boldsymbol{r}(t)$ and define

$$\delta_n(t) := \frac{\gamma_n(t)}{b_n}, \qquad \psi_n(t) = \big(\delta_n(t) e^{-z_n(t)} - 1\big) z_n(t),$$

so that

$$T_1(t) = \frac{1}{t} \sum_{n \in S} b_n \, \psi_n(t).$$

We first justify that the coefficients $\delta_n(t) = \gamma_n(t)/b_n$ remain in a fixed compact interval. By equation 7,

$$|\widehat{m}_n(t) - m_n(t)| \le C.$$

Since

$$\gamma_n(t) = \frac{e^{-(\widehat{m}_n(t) - m_n(t))}}{1 + e^{-\widehat{m}_n(t)}},$$

and the denominator satisfies $1 + e^{-\widehat{m}_n(t)} \ge 1$, we obtain the uniform bound

$$0 \le \gamma_n(t) \le e^{-(\widehat{m}_n(t) - m_n(t))} \le e^C \qquad \text{for all } n, t.$$

Thus each $\gamma_n(t)$ lies in the compact interval

$$[0, e^C].$$

Next, since every $b_n > 0$ for $n \in S$ and $S$ is a finite set, define

$$b_{\min} := \min_{n \in S} b_n > 0, \qquad b_{\max} := \max_{n \in S} b_n.$$

Therefore

$$\delta_n(t) = \frac{\gamma_n(t)}{b_n} \qquad \Longrightarrow \qquad 0 \le \delta_n(t) \le \frac{e^C}{b_{\min}} \quad \text{for all } n \in S \text{ and all } t.$$

Hence $\delta_n(t)$ ranges over the compact interval

$$[\delta_{\min}, \delta_{\max}] = \Big[0, \; \frac{e^C}{b_{\min}}\Big].$$

For each fixed $\delta > 0$, consider the function

$$\psi(z, \delta) := (\delta e^{-z} - 1) z.$$

As $z \to \pm\infty$ we have $\psi(z, \delta) \to -\infty$, and therefore $\psi(z, \delta)$ attains a finite global maximum on $\mathbb{R}$. Since $\delta_n(t) \in [\delta_{\min}, \delta_{\max}]$ for all $t$, there exists a constant $C_\psi > 0$ such that

$$\psi(z, \delta) \le C_\psi \qquad \forall z \in \mathbb{R}, \; \forall \delta \in [\delta_{\min}, \delta_{\max}].$$

Consequently,

$$\psi_n(t) = \psi(z_n(t), \delta_n(t)) \le C_\psi \qquad \forall n \in S, \; \forall t,$$

and therefore

$$T_1(t) \le \frac{C_1}{t}, \qquad C_1 := C_\psi \sum_{n \in S} b_n.$$

Combining the two bounds on $T_1(t), T_2(t)$, for sufficiently large $t$,

$$\frac{1}{2} \frac{d}{dt} \|\boldsymbol{r}(t)\|^2 = T_1(t) + T_2(t) \le \frac{C_1}{t} + \frac{A}{t^{1+\theta}} \le \frac{C_2}{t},$$

for some constant $C_2 > 0$.

Integrating from $t_0$ to $t$ gives

$$\|\boldsymbol{r}(t)\|^2 \leq \|\boldsymbol{r}(t_0)\|^2 + 2C_2 \int_{t_0}^{t} u^{-1} du = \|\boldsymbol{r}(t_0)\|^2 + 2C_2 \log\left(\frac{t}{t_0}\right),$$

so

$$\|\boldsymbol{r}(t)\| = O(\sqrt{\log t}) = o(\log t).$$

Since

$$\boldsymbol{w}(t) = \boldsymbol{w}^* \log t + \boldsymbol{r}(t), \qquad \|\boldsymbol{r}(t)\| = o(\log t),$$

we obtain

$$\frac{\boldsymbol{w}(t)}{\|\boldsymbol{w}(t)\|} = \frac{\boldsymbol{w}^*}{\|\boldsymbol{w}^*\|} + o(1),$$

which proves

$$\frac{\boldsymbol{w}(t)}{\|\boldsymbol{w}(t)\|} \to \frac{\boldsymbol{w}^*}{\|\boldsymbol{w}^*\|_2}.$$

Thus $\ell_2$-SAM flow converges in the $\ell_2$ max-margin direction for any initialization and any fixed $\rho > 0$. $\qquad\square$

## C   SAM WITH $\ell_\infty$-PERTURBATIONS: PROOF OF SECTION 3

### C.1   DEPTH-1 NETWORKS: PROOF OF THEOREM 3.1

**Theorem 3.1.** *For almost every dataset which is linearly separable, any perturbation radius $\rho$ and any initialization, consider the linear model $f(\boldsymbol{x}) = \langle \boldsymbol{w}, \boldsymbol{x} \rangle$ trained with logistic loss. Then, $\ell_\infty$-SAM flow directionally converges in the $\ell_2$ max-margin direction.*

*Proof.* Apply Lemma B.1 with $\boldsymbol{\varepsilon}(\boldsymbol{w}) = \rho \, \mathrm{sign}(\nabla \mathcal{L}(\boldsymbol{\theta}))$. Then $\|\boldsymbol{\varepsilon}(\boldsymbol{w})\|_2 \leq \rho\sqrt{d}$ for all $\boldsymbol{w}$, so the conditions of Lemma B.1 hold. Thus, the flow converges to the $\ell_2$ max-margin direction. □

**Theorem C.1.** *Consider the linear model $f(\boldsymbol{x}) = \langle \boldsymbol{w}, \boldsymbol{x} \rangle$ trained on the dataset $\mathcal{D}_{\boldsymbol{\mu}}$ with loss $\mathcal{L}(\boldsymbol{w}) = \ell(\langle \boldsymbol{w}, \boldsymbol{x} \rangle)$ where $\ell'(u) < 0$ for all $u$. Then, GF and $\ell_\infty$-SAM flow, starting from any $\boldsymbol{w}(0)$, evolve on the same affine line $\boldsymbol{w}(0) + \mathrm{span}\{\boldsymbol{\mu}\}$ and have the same spatial trajectory.*

*Proof.* The model is $f(\boldsymbol{x}) = \langle \boldsymbol{w}, \boldsymbol{x} \rangle = \boldsymbol{w}^\top \boldsymbol{x}$. The loss is $\mathcal{L}(\boldsymbol{w}) = \ell(\boldsymbol{w}^\top \boldsymbol{\mu})$. The gradient is $\nabla_{\boldsymbol{w}} \mathcal{L}(\boldsymbol{w}) = \ell'(\boldsymbol{w}^\top \boldsymbol{\mu}) \cdot \boldsymbol{\mu}$ with $\ell'(s) < 0$.

**Gradient Descent**   The GF is

$$\dot{\boldsymbol{w}} = -\nabla_{\boldsymbol{w}} \mathcal{L}(\boldsymbol{w})$$
$$= -\ell'(\boldsymbol{w}^\top \boldsymbol{\mu}) \cdot \boldsymbol{\mu}.$$

**SAM with $\ell_\infty$ perturbation**   The ascent point is

$$\hat{\boldsymbol{w}} = \boldsymbol{w} + \rho \boldsymbol{\varepsilon}_\infty(\boldsymbol{w})$$
$$= \boldsymbol{w} + \rho \, \mathrm{sign}(\nabla_{\boldsymbol{w}} \mathcal{L}(\boldsymbol{w}))$$
$$= \boldsymbol{w} - \rho \, \mathrm{sign}(\boldsymbol{\mu}).$$

The equation of $\ell_\infty$-SAM flow is

$$\dot{\boldsymbol{w}} = -\nabla_{\boldsymbol{w}} \mathcal{L}(\hat{\boldsymbol{w}})$$
$$= -\nabla_{\boldsymbol{w}} \mathcal{L}(\boldsymbol{w} - \rho \, \mathrm{sign}(\boldsymbol{\mu}))$$
$$= -\ell'(\boldsymbol{w}^\top \boldsymbol{\mu} - \rho \, \mathrm{sign}(\boldsymbol{\mu})^\top \boldsymbol{\mu}) \cdot \boldsymbol{\mu}$$
$$= -\ell'(\boldsymbol{w}^\top \boldsymbol{\mu} - \rho \|\boldsymbol{\mu}\|_1) \cdot \boldsymbol{\mu}.$$

Therefore, they have the same spatial trajectory as:

$$\dot{\boldsymbol{w}} = \boldsymbol{\mu}.$$

The term $-\ell'(\boldsymbol{w}^\top \boldsymbol{\mu} - \rho \|\boldsymbol{\mu}\|_1)$ is the accelation in terms of $t$ since $-\ell'(s)$ is decreasing in $s$.  □

### C.2   TECHNICAL CHALLENGES FOR MULTI-POINT DATASETS

In the multi-point setting, as $\boldsymbol{w}(t)$ diverges the SAM perturbation becomes asymptotically negligible, so SAM and GD share the same long-term behavior. The regime where they differ is precisely when the $\rho$-perturbation is non-negligible, but in the multi-point case the resulting gradients (and thus SAM updates) become considerably complex for a tractable characterization of the SAM flow in the regime where SAM and GD diverge. This motivates our focus on the single-example dataset $\mathcal{D}_{\boldsymbol{\mu}} = \{(\boldsymbol{\mu}, +1)\}$, where the SAM dynamics admit a tractable dynamical characterization while still capturing depth-dependent phenomena unique to SAM. In Appendix C.6, we empirically verify that these behaviors persist under multi-point datasets and discrete SAM updates, indicating that our insights extend beyond the single-point setting.

### C.3 PROOF OF THEOREM 3.2

**Theorem 3.2.** *For $i \in [L]$, suppose $\boldsymbol{w}^{(i)}(0) = \boldsymbol{\alpha} \in \mathbb{R}_+^d$. Let $\boldsymbol{w}^{(i)}(t)$ follow the rescaled $\ell_\infty$-SAM flow (2) with perturbation radius $\rho > 0$ on the dataset $\mathcal{D}_{\boldsymbol{\mu}}$. Then, for the $j$-th coordinate of $\boldsymbol{\beta}(t)$:*

- *If $\alpha_j < \rho$, then $\beta_j(t)$ converges to $0$ if $L$ is even, or to $\rho^L$ if $L$ is odd.*
- *If $\alpha_j = \rho$, then $\beta_j(t) = \rho^L$ for all $t \geq 0$.*
- *If $\alpha_j > \rho$ and $L = 2$, then $\beta_j(t)$ grows exponentially: $\beta_j(t) = \Theta(\exp(2\mu_j t))$.*
- *If $\alpha_j > \rho$ and $L > 2$, let $J := \arg\max_{j:\alpha_j > \rho} \mu_j(\alpha_j - \rho)^{L-2}$, and also let $T := \min_{k \in J} 1/(L-2)\mu_k(\alpha_k - \rho)^{L-2}$. If $j \in J$, then $\beta_j(t) \to \infty$ as $t \to T$; otherwise, $\beta_j(t)$ stays bounded for all $t < T$.*

*Proof.* Since we suppose $\boldsymbol{w}^{(i)}(0) = \boldsymbol{\alpha} \in \mathbb{R}_+^d$ for all $i \in [L]$, and the dynamics of the linear diagonal network are invariant under any permutation of the layer indices $\{1, \ldots, L\}$, we obtain

$$\boldsymbol{w}^{(1)}(t) = \boldsymbol{w}^{(2)}(t) = \cdots = \boldsymbol{w}^{(L)}(t) =: \boldsymbol{w}(t) \quad \text{for all } t \geq 0.$$

With $\ell_\infty$ perturbation, the rescaled $\ell_\infty$-SAM flow (2) becomes

$$\dot{\boldsymbol{w}}^{(i)}(t) = \boldsymbol{\mu} \odot \left( \bigodot_{\ell \neq i} \left( \boldsymbol{w}^{(\ell)}(t) + \boldsymbol{\varepsilon}_\infty^{(\ell)}(\boldsymbol{\theta}(t)) \right) \right)$$

$$= \boldsymbol{\mu} \odot \left( \bigodot_{\ell \neq i} \left( \boldsymbol{w}^{(\ell)}(t) + \rho \operatorname{sign}(\nabla_{\boldsymbol{w}^{(\ell)}} \mathcal{L}(\boldsymbol{\theta}(t))) \right) \right).$$

Recall the gradient (5)

$$\nabla_{\boldsymbol{w}^{(\ell)}} \mathcal{L}(\boldsymbol{\theta}(t)) = \ell'\left(\langle \boldsymbol{\beta}(\boldsymbol{\theta}(t)), \boldsymbol{\mu}\rangle\right) \boldsymbol{\mu} \odot \left( \bigodot_{\ell \neq i} \boldsymbol{w}^{(\ell)}(t) \right),$$

where $\ell'(u) = -\frac{1}{1+\exp(u)} < 0$. Since we also have $\boldsymbol{\mu} > 0$ (element-wise), we have

$$\operatorname{sign}(\nabla_{\boldsymbol{w}^{(\ell)}} \mathcal{L}(\boldsymbol{\theta}(t))) = -\operatorname{sign}\left( \bigodot_{\ell \neq i} \boldsymbol{w}^{(\ell)}(t) \right)$$

$$\underset{(a)}{=} -\operatorname{sign}\left( \bigodot_{\ell=1}^{L-1} \boldsymbol{w}(t) \right),$$

where (a) follows from the fact that $\boldsymbol{w}^{(i)}(t) = \boldsymbol{w}(t)$ for all $i \in [L]$. Using this fact again, we have the ODE

$$\dot{\boldsymbol{w}}(t) = \dot{\boldsymbol{w}}^{(i)}(t) = \boldsymbol{\mu} \odot \left( \bigodot_{\ell \neq i} \left( \boldsymbol{w}(t) - \rho \operatorname{sign}\left( \bigodot_{\ell=1}^{L-1} \boldsymbol{w}(t) \right) \right) \right)$$

$$= \boldsymbol{\mu} \odot \left( \bigodot_{\ell=1}^{L-1} \left( \boldsymbol{w}(t) - \rho \operatorname{sign}\left( \bigodot_{\ell=1}^{L-1} \boldsymbol{w}(t) \right) \right) \right).$$

This can be written as coordinate-wise as

$$\dot{w}_j(t) = \mu_j \left( w_j(t) - \rho \operatorname{sign}\left( w_j(t)^{L-1} \right) \right)^{L-1} \quad \text{for } j \in [d].$$

Divide into three cases:

**Case 1:** $L = 2$.

$$\dot{w}_j(t) = \mu_j \left( w_j(t) - \rho \operatorname{sign}(w_j(t)) \right).$$

By Lemma C.2, we have

$$w_j(t) = \begin{cases} \rho + (w_j(0) - \rho)e^{\mu_j t} & \text{if } w_j(0) > \rho, \\ \rho & \text{if } w_j(0) = \rho, \\ \rho + (w_j(0) - \rho)e^{\mu_j t} \ (t < T), \quad 0 \ (t \geq T) & \text{if } w_j(0) < \rho, \\ 0 & \text{if } w_j(0) = 0, \end{cases}$$

where $T := \frac{1}{\mu_j} \log \left( \frac{\rho}{\rho - w_j(0)} \right)$. Then, we have

$$\beta_j(t) = w_j(t)^L \to \begin{cases} \Theta(e^{2\mu_j t}) & \text{if } \alpha_j > \rho, \\ \rho^L & \text{if } \alpha_j = \rho, \quad \text{as } t \to \infty. \\ 0 & \text{if } \alpha_j < \rho, \end{cases}$$

**Case 2:** $L > 2$ **and** $L$ **is even.**

$$\dot{w}_j(t) = \mu_j \left( w_j(t) - \rho \operatorname{sign}(w_j(t)) \right)^{L-1}.$$

By Lemma C.3, we have

$$w_j(t) = \begin{cases} \rho + \left( -(L-2)\mu_j t + \frac{1}{(w_j(0)-\rho)^{L-2}} \right)^{-\frac{1}{L-2}} & \text{if } w_j(0) > \rho, \\ \rho & \text{if } w_j(0) = \rho, \\ \rho - \left( -(L-2)\mu_j t + \frac{1}{(w_j(0)-\rho)^{L-2}} \right)^{-\frac{1}{L-2}} \ (t < T), \quad 0 \ (t \geq T) & \text{if } w_j(0) < \rho, \\ 0 & \text{if } w_j(0) = 0, \end{cases}$$

where $T := \frac{(\rho - w_j(0))^{-(L-2)} - \rho^{-(L-2)}}{(L-2)\mu_j}$. Then, we have

$$\beta_j(t) = w_j(t)^L \to \begin{cases} \Theta\left( (t^* - t)^{-\frac{L}{L-2}} \right) & \text{if } \alpha_j > \rho, \text{ as } t \to t^*, \\ \rho^L & \text{if } \alpha_j = \rho, \text{ as } t \to \infty, \\ 0 & \text{if } \alpha_j < \rho, \text{ as } t \to \infty, \end{cases}$$

where $t^* = 1/(L-2)\mu_j(w_j(0)-\rho)^{L-2}$

**Case 3:** $L > 2$ **and** $L$ **is odd.**

$$\dot{w}_j(t) = \mu_j \left( w_j(t) - \rho \right)^{L-1}.$$

By Lemma C.4, we have

$$w_j(t) = \begin{cases} \rho & \text{if } w_j(0) = \rho, \\ \rho + \left( -(L-2)\mu_j t + \frac{1}{(w_j(0)-\rho)^{L-2}} \right)^{-\frac{1}{L-2}} & \text{if } w_j(0) \neq \rho. \end{cases}$$

Then, we have

$$\beta_j(t) = w_j(t)^L \to \begin{cases} \Theta\left( (t^* - t)^{-\frac{L}{L-2}} \right) & \text{if } \alpha_j > \rho, \text{ as } t \to t^*, \\ \rho^L & \text{if } \alpha_j \leq \rho, \text{ as } t \to \infty, \end{cases}$$

where $t^* = 1/(L-2)\mu_j(w_j(0)-\rho)^{L-2}$.

These cases of $L$ cover all possible cases in Theorem 3.2.

$\square$

The following three lemmas (Lemmas C.2 to C.4) are used in the proof of Theorem 3.2 and correspond, respectively, to the three cases.

**Lemma C.2.** *Let $\mu > 0$ and $\rho > 0$. Consider*
$$\dot{w}(t) = \mu \left( w(t) - \rho \operatorname{sign}(w(t)) \right).$$
*Then, there exists the solution $w$ such that it is absolutely continuous (AC) and satisfies*
$$w(t) = w(0) + \int_0^t \dot{w}(s)ds. \tag{10}$$

*In particular,*
$$w(t) = \begin{cases} \rho + (w(0) - \rho)e^{\mu t} & \text{if } w(0) > \rho, \\ \rho & \text{if } w(0) = \rho, \\ \rho + (w(0) - \rho)e^{\mu t} \; (t < T), \quad 0 \; (t \geq T) & \text{if } w(0) < \rho, \\ 0 & \text{if } w(0) = 0, \end{cases}$$
*where $T := \frac{1}{\mu} \log \left( \frac{\rho}{\rho - w(0)} \right)$.*

*Proof.* **Case 1:** $w(0) = 0$. The constant function $w(t) = 0$ is AC, and
$$\int_0^t \mu \left( 0 - \rho \operatorname{sign}(0) \right) ds = \int_0^t 0 \, ds = 0.$$
Thus, Equation (10) holds.

**Case 2:** $w(0) = \rho$. The constant function $w(t) = \rho$ is AC, and since $\operatorname{sign}(w(t)) = 1$, we have
$$\int_0^t \mu \left( \rho - \rho \cdot 1 \right) ds = \int_0^t 0 \, ds = 0.$$
Thus, Equation (10) holds.

**Case 3:** $w(0) > \rho$. At $t = 0$, we have $\dot{w}(0) = \mu \left( w(0) - \rho \right) > 0$. Assume, for contradiction, that there exists $t_\star > 0$ with $w(t_\star) = \rho$. Then on $[0, t_\star)$ we have $w(t) > \rho$ and hence $\dot{w}(t) = \mu \left( w(t) - \rho \right) > 0$, so $w$ is strictly increasing on $[0, t_\star)$. An increasing function cannot reach the smaller value $\rho$ starting from $w(0) > \rho$: contradiction. Thus $w(t) > \rho$ for all $t \geq 0$. On the region $\{w(t) > \rho\}$, $\operatorname{sign}(w(t)) = 1$ and the ODE reduces to the linear equation
$$\dot{w} = \mu(w - \rho).$$
Then, we have
$$\frac{\dot{w}(t)}{w(t) - \rho} = \mu$$
$$\Rightarrow \int_0^t \frac{\dot{w}(s)}{w(s) - \rho} ds = \int_0^t \mu ds$$
$$\Rightarrow \log \left| \frac{w(t) - \rho}{w(0) - \rho} \right| = \mu t$$
$$\Rightarrow w(t) = \rho + \left( w(0) - \rho \right) e^{\mu t}.$$

This function is AC and satisfies Equation (10).

**Case 4:** $0 < w(0) < \rho$. Initially $\operatorname{sign}(w(0)) = 1$, so again $\dot{w} = \mu(w - \rho)$ and
$$w(t) = \rho + \left( w(0) - \rho \right) e^{\mu t}.$$
Since $w(0) - \rho < 0$, the function $w$ is strictly decreasing and reaches 0 exactly once at
$$T := \frac{1}{\mu} \log \left( \frac{\rho}{\rho - w(0)} \right) > 0.$$
On $[0, T]$, this solution is AC and satisfies Equation (10). Define $w(t) := 0$ for all $t \geq T$. Then, using $\operatorname{sign}(0) = 0$,
$$w(t) = w(T) + \int_T^t \mu \left( 0 - \rho \operatorname{sign}(0) \right) ds = 0 + \int_T^t 0 \, ds = 0,$$
so Equation (10) also holds on $[T, \infty)$. The function $w$ is AC on $[0, T]$ and on $[T, \infty)$, and it is continuous at $t = T$, hence it is absolutely continuous. $\qquad \square$

**Lemma C.3.** *Let $\mu > 0$, $\rho > 0$, and L is even. Consider*

$$\dot{w}(t) = \mu\left(w(t) - \rho\operatorname{sign}(w(t))\right)^{L-1}.$$

*Then, there exists the solution $w$ such that it is absolutely continuous (AC) and satisfies Equation (10). In particular,*

$$w(t) = \begin{cases} \rho + \left(-(L-2)\mu t + \frac{1}{(w(0)-\rho)^{L-2}}\right)^{-\frac{1}{L-2}} & \text{if } w(0) > \rho, \\ \rho & \text{if } w(0) = \rho, \\ \rho - \left(-(L-2)\mu t + \frac{1}{(w(0)-\rho)^{L-2}}\right)^{-\frac{1}{L-2}} \ (t < T), \quad 0 \ (t \geq T) & \text{if } w(0) < \rho, \\ 0 & \text{if } w(0) = 0, \end{cases}$$

*where $T := \frac{(\rho-w(0))^{-(L-2)}-\rho^{-(L-2)}}{(L-2)\mu}$.*

*Proof.* The proof is similar to the proof of Lemma C.2.

**Case 1:** $w(0) = 0$**.** The constant function $w(t) = 0$ is AC, and

$$\int_0^t \mu\left(0 - \rho\operatorname{sign}(0)\right)^{L-1} ds = \int_0^t \mu \cdot 0^{L-1} ds = 0.$$

Thus, Equation (10) holds.

**Case 2:** $w(0) = \rho$**.** The constant function $w(t) = \rho$ is AC, and since $\operatorname{sign}(w(t)) = 1$, we have

$$\int_0^t \mu\left(\rho - \rho \cdot 1\right)^{L-1} ds = \int_0^t \mu \cdot 0^{L-1} ds = 0.$$

Thus, Equation (10) holds.

**Case 3:** $w(0) > \rho$**.** At $t = 0$, we have $\dot{w}(0) = \mu\left(w(0) - \rho\right)^{L-1} > 0$. Assume, for contradiction, that there exists $t_\star > 0$ with $w(t_\star) = \rho$. Then on $[0, t_\star)$ we have $w(t) > \rho$ and hence $\dot{w}(t) = \mu\left(w(t) - \rho\right) > 0$, so $w$ is strictly increasing on $[0, t_\star)$. An increasing function cannot reach the smaller value $\rho$ starting from $w(0) > \rho$: contradiction. Thus $w(t) > \rho$ for all $t \geq 0$. On the region $\{w(t) > \rho\}$, $\operatorname{sign}(w(t)) = 1$ and the ODE reduces to

$$\dot{w} = \mu(w - \rho)^{L-1}.$$

Then, we have

$$\frac{\dot{w}(t)}{(w(t) - \rho)^{L-1}} = \mu$$

$$\Rightarrow \int_0^t \frac{\dot{w}(s)}{(w(s) - \rho)^{L-1}} ds = \int_0^t \mu ds$$

$$\Rightarrow -\frac{1}{L-2}\left(\frac{1}{(w(t) - \rho)^{L-2}} - \frac{1}{(w(0) - \rho)^{L-2}}\right) = \mu t$$

$$\Rightarrow (w(t) - \rho)^{L-2} = \left(-(L-2)\mu t + \frac{1}{(w(0) - \rho)^{L-2}}\right)^{-1}$$

$$\underset{(a)}{\Rightarrow} w(t) = \rho + \left(-(L-2)\mu t + \frac{1}{(w(0) - \rho)^{L-2}}\right)^{-\frac{1}{L-2}},$$

where $(a)$ follows from $w(t) - rho > 0$. This function is AC and satisfies Equation (10).

**Case 4:** $0 < w(0) < \rho$**.** Initially $\operatorname{sign}(w(0)) = 1$, so again $\dot{w} = \mu(w - \rho)^{L-1}$ and

$$(w(t) - \rho)^{L-2} = \left(-(L-2)\mu t + \frac{1}{(w(0) - \rho)^{L-2}}\right)^{-1}.$$

Since $w(0) - \rho < 0$ and $L$ is even, we have

$$w(t) = \rho - \left(-(L-2)\mu t + \frac{1}{(w(0)-\rho)^{L-2}}\right)^{-\frac{1}{L-2}}.$$

The function $w$ is strictly decreasing and reaches $0$ exactly once at

$$T := \frac{(\rho - w(0))^{-(L-2)} - \rho^{-(L-2)}}{(L-2)\mu} > 0.$$

On $[0, T]$, this solution is AC and satisfies Equation (10). Define $w(t) := 0$ for all $t \geq T$. Then, using $\text{sign}(0) = 0$,

$$w(t) = w(T) + \int_T^t \mu\big(0 - \rho\,\text{sign}(0)\big)^{L-1} ds = 0 + \int_T^t 0\, ds = 0,$$

so Equation (10) also holds on $[T, \infty)$. The function $w$ is AC on $[0, T]$ and on $[T, \infty)$, and it is continuous at $t = T$, hence it is absolutely continuous. $\qquad\square$

**Lemma C.4.** *Let $\mu > 0$, $\rho > 0$ and $L$ is odd. Consider*

$$\dot{w}(t) = \mu\,(w(t) - \rho)^{L-1}.$$

*Then, there exists the solution $w$ such that it is absolutely continuous (AC) and satisfies Equation (10). In particular,*

$$w(t) = \begin{cases} \rho & \text{if } w(0) = \rho, \\ \rho + \left(-(L-2)\mu t + \frac{1}{(w(0)-\rho)^{L-2}}\right)^{-\frac{1}{L-2}} & \text{if } w(0) \neq \rho, \end{cases}$$

*Proof.* The proof is similar to the proof of Lemma C.2.

**Case 1:** $w(0) = \rho$**.** The constant function $w(t) = \rho$ is AC, and

$$\int_0^t \mu\big(\rho - \rho\big)\, ds = \int_0^t 0\, ds = 0.$$

Thus, Equation (10) holds.

**Case 2:** $w(0) \neq \rho$**.** Separate variables:

$$\frac{dw}{(w-\rho)^{L-1}} = \mu\, dt.$$

Integrating from $0$ to $t$ gives

$$-\frac{1}{L-2}\left(\frac{1}{(w(t)-\rho)^{L-2}} - \frac{1}{(w(0)-\rho)^{L-2}}\right) = \mu t.$$

Solving for $w$ yields

$$w(t) = \rho + \left(-(L-2)\mu t + \frac{1}{(w(0)-\rho)^{L-2}}\right)^{-\frac{1}{L-2}}.$$

The function is AC and satisfies Equation (10).

$\qquad\square$

## C.4 PROOF OF COROLLARY 3.5

**Corollary 3.5.** *Under the assumptions of Theorem 3.2, let $S := \{j : \alpha_j > \rho\}$ and assume $S \neq \varnothing$. If there is a unique maximizing index $j^* := \arg\max_{j \in S} \mu_j (\alpha_j - \rho)^{L-2}$, then the $\ell_\infty$-SAM flow converges in the $\boldsymbol{e}_{j^*}$ direction. In particular, when $L = 2$, we have $j^* := \arg\max_{j \in S} \mu_j$.*

*Proof.* Work under the assumptions of Theorem 3.2 and let

$$S := \{j : \alpha_j > \rho\} \neq \varnothing, \qquad j^* := \arg\max_{j \in S} \mu_j (\alpha_j - \rho)^{L-2},$$

where the maximizer is unique. We prove that the (rescaled) $\ell_\infty$–SAM flow satisfies

$$\frac{\boldsymbol{\beta}(t)}{\|\boldsymbol{\beta}(t)\|_2} \longrightarrow \boldsymbol{e}_{j^*}.$$

**Case $L = 2$.** By Theorem 3.2, for $j \in S$,

$$\beta_j(t) = \Theta(e^{2\mu_j t}),$$

whereas for $j \notin S$ we have either $\beta_j(t) \to 0$ (if $L$ even) or $\beta_j(t) \equiv \rho^L$ when $\alpha_j = \rho$; in any event these coordinates stay bounded. Since the maximizer is unique and $L - 2 = 0$,

$$j^* = \arg\max_{j \in S} \mu_j,$$

hence for every $k \in S \setminus \{j^*\}$,

$$\frac{\beta_k(t)}{\beta_{j^*}(t)} = \Theta\!\left(e^{-2(\mu_{j^*} - \mu_k)t}\right) \longrightarrow 0,$$

and for $k \notin S$ we also have $\beta_k(t)/\beta_{j^*}(t) \to 0$ because the denominator grows exponentially while the numerator is bounded. Therefore $\boldsymbol{\beta}(t)/\|\boldsymbol{\beta}(t)\|_2 \to \boldsymbol{e}_{j^*}$.

**Case $L > 2$.** By Theorem 3.2, for each $j \in S$ there is a blow-up time

$$t_j^* = \frac{1}{(L-2)\,\mu_j\,(\alpha_j - \rho)^{L-2}},$$

and as $t \uparrow t_j^*$,

$$\beta_j(t) = \Theta\!\left((t_j^* - t)^{-1/(L-2)}\right).$$

If $j \notin S$, then $\beta_j(t)$ is bounded (either converging to 0 when $L$ is even, or equal to $\rho^L$ when $\alpha_j = \rho$). The uniqueness of $j^*$ implies

$$t_{j^*}^* = \min_{j \in S} t_j^* \quad \text{and} \quad t_{j^*}^* < t_k^* \ \forall k \in S \setminus \{j^*\}.$$

Hence, for any fixed $t < t_{j^*}^*$, all coordinates with $k \neq j^*$ are finite; moreover,

$$\lim_{t \uparrow t_{j^*}^*} \frac{\beta_k(t)}{\beta_{j^*}(t)} = 0 \qquad \text{for every } k \neq j^*,$$

because $\beta_{j^*}(t) \to \infty$ while $\beta_k(t)$ remains finite as $t < t_k^*$. Consequently,

$$\lim_{t \uparrow t_{j^*}^*} \frac{\boldsymbol{\beta}(t)}{\|\boldsymbol{\beta}(t)\|_2} = \boldsymbol{e}_{j^*}.$$

Combining the two cases establishes the claim. In particular, when $L = 2$ we have $j^* = \arg\max_{j \in S} \mu_j$. $\qquad \square$

### C.5 FINITE-TIME BLOW-UP

In the setting of Theorem C.1, the $\ell_\infty$-SAM flow evolves independently across coordinates. In the rescaled $\ell_\infty$-SAM flow, each coordinate indeed admits a finite blow-up time. However, as explained in Remark 3.3, the smallest of these blow-up times corresponds to $t_{\mathrm{orig}} = \infty$ in the original SAM time scale. Consequently, both the original flow and the rescaled flow terminate at this same time and cannot be extended beyond it.

To illustrate this behavior concretely, we provide Figures 9 and 10 using $\mu = (1, 2, 3, 4, 5)$, $\rho = 1$, and a depth-$L = 3$ network. In the original flow, only one coordinate diverges as $t_{\mathrm{orig}} \to \infty$. As shown in Figure 9b, the normalized trajectories $\beta_j(t)/\|\beta(t)\|$ show that the remaining coordinates grow much more slowly than the dominant one—indeed, they remain bounded. Because their growth is negligible compared to the blow-up coordinate, their normalized values converge to zero. Thus, in this example, the trajectory converges to the direction $e_5$.

In contrast, Figure 10a shows that in the rescaled $\ell_\infty$-SAM flow, each coordinate $\beta_j(t)$ has its own finite blow-up time. However, Theorem 3.2 identifies the blow-up time $T = \frac{1}{(L-2)\mu_j(\alpha_j-\rho)^{L-2}}$ for any $j \in J$, which is the minimum of these blow-up times—only the coordinates in $J$ blow up at $T$, while all remaining coordinates stay bounded. Since this rescaled time $T$ corresponds to $t_{\mathrm{orig}} = \infty$, the flow cannot proceed past $T$. In this example, $T \approx 0.25$.

Because the rescaled system is simply a time reparameterization of the original one, the two plots differ only in their $x$-axis scaling. Before reaching $T$, the two flows exhibit the same evolution along the $y$-axis. Indeed, reparameterizing the original trajectory (Figure 9) by $\tau(t)$ reproduces the same curve as shown in Figure 10 before $T$.

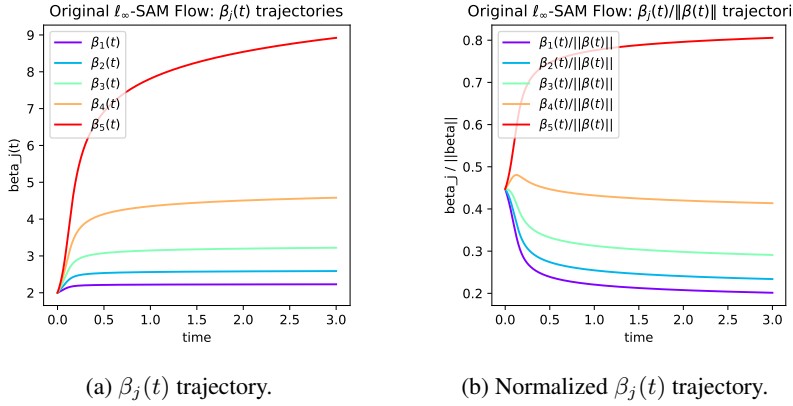

(a) $\beta_j(t)$ trajectory.   (b) Normalized $\beta_j(t)$ trajectory.

Figure 9: $\beta_j(t)$ and normalized $\beta_j(t)$ trajectory of the original $\ell_\infty$-SAM flow.

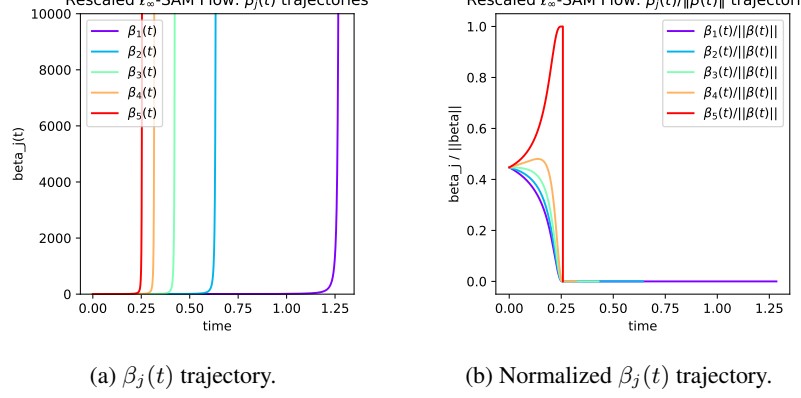

(a) $\beta_j(t)$ trajectory.   (b) Normalized $\beta_j(t)$ trajectory.

Figure 10: $\beta_j(t)$ and normalized $\beta_j(t)$ trajectory of the rescaled $\ell_\infty$-SAM flow.

### C.6 EMPIRICAL VERIFICATION

Our theoretical analysis (Theorem 3.2 and Corollary 3.5) establishes the behavior of the $\ell_\infty$-SAM flow in the one-point setting $\mathcal{D}_\mu$. In this section, we investigate whether these phenomena extend beyond the idealized one-point regime. We first examine the discrete-time dynamics (GD and discrete $\ell_\infty$-SAM) on the one-point dataset and verify that they exhibit exactly the same trajectory patterns predicted by the continuous-time theory. We then turn to multi-point datasets and demonstrate that the same qualitative behaviors persist in both the continuous-time flows and their discrete counterparts. Taken together, these experiments empirically confirm that the insights obtained from $\mathcal{D}_\mu$ carry over robustly to multi-point datasets and to practical discrete SAM updates.

For reproducibility, we detail the exact initialization used in all experiments. We adopt the layer-wise balanced initialization $\boldsymbol{w}^{(i)}(0) = \boldsymbol{\alpha}$ for every $i \in [L]$, consistent with the setup of Theorem 3.2. The black-edged dot in Figures 11 and 13 indicates the initial predictor $\boldsymbol{\beta}(0)$. We set $\boldsymbol{w}^{(i)}(0) = \boldsymbol{\beta}(0)^{1/L}$ element-wise so that $\boldsymbol{\beta}(0) = \bigodot_{i=1}^{L} \boldsymbol{w}^{(i)}(0)$ holds exactly. For the continuous-time trajectories, we approximate the flow using the corresponding discrete updates with a small step size $\eta = 10^{-3}$ via an explicit Euler scheme.

#### C.6.1 ONE-POINT CASE: DISCRETE VS. CONTINUOUS DYNAMICS

To verify that our continuous-time analysis faithfully predicts the behavior of the corresponding discrete algorithms, we repeat the experiments in Figure 2 using exactly the same initializations, SAM radius $\rho$, and feature vector $\boldsymbol{\mu}$. We simulate both the gradient flows (black curves) and their discrete counterparts (blue dots), including GD and discrete $\ell_\infty$-SAM updates. As shown below, the discrete trajectories closely trace the qualitative evolution of their continuous-time versions.

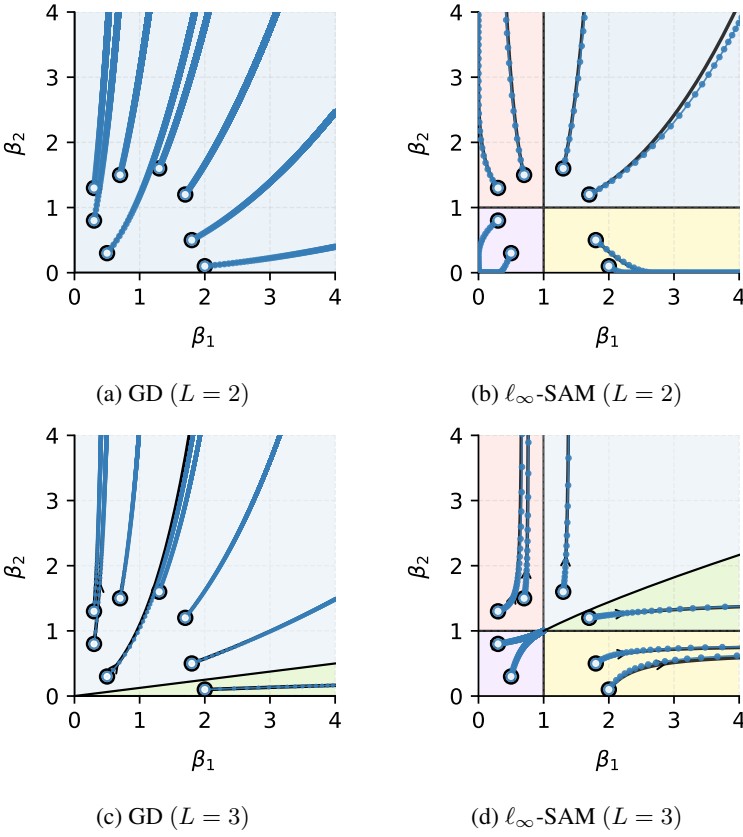

(a) GD ($L = 2$)   (b) $\ell_\infty$-SAM ($L = 2$)

(c) GD ($L = 3$)   (d) $\ell_\infty$-SAM ($L = 3$)

Figure 11: Trajectories $\boldsymbol{\beta}(t)$ under GF, $\ell_\infty$-SAM flow (black line), GD, and discrete $\ell_\infty$-SAM updates (blue dots) for $d = 2$ and $\boldsymbol{\mu} = (1, 2)$. For SAM, we set $\rho = 1$. For GD and discrete $\ell_\infty$-SAM, we use step size $\eta = 0.1$.

### C.6.2 MULTI-POINT CASE: PERSISTENCE OF ONE-POINT BEHAVIOR

To examine whether the qualitative behaviors identified in the one-point analysis persist on more realistic datasets, we construct random linearly separable binary data by sampling two Gaussian clusters centered at $+\boldsymbol{\mu}$ and $-\boldsymbol{\mu}$ as shown in Figure 12. Specifically, we draw

$$\boldsymbol{x}_n^{(+)} = \boldsymbol{\mu} + \boldsymbol{\varepsilon}_n, \quad y_n = +1, \qquad \boldsymbol{x}_n^{(-)} = -\boldsymbol{\mu} + \boldsymbol{\varepsilon}_n, \quad y_n = -1,$$

with $\varepsilon_n \sim \mathcal{N}(0, \sigma^2 \boldsymbol{I}_d)$ and use $N/2$ samples per class (with $\boldsymbol{\mu} = (1, 2), N = 100, \sigma = 0.5$).

Figures 11 and 13 show that the same qualitative patterns predicted by our one-point theory—such as the asymptotic trajectory structure—also emerge clearly in this multi-point setting. Importantly, these behaviors are observed not only in the continuous-time flows but also in their discrete counterparts (GD and discrete $\ell_\infty$-SAM). This empirical evidence demonstrates that the phenomena described in Theorem 3.2 and Corollary 3.5 extend robustly beyond the one-point setting to general linearly separable datasets.

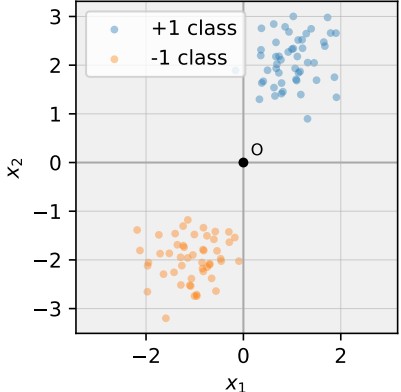

Figure 12: A randomly generated linearly separable dataset used in our multi-point experiments. We sample two Gaussian clusters centered at $\pm\boldsymbol{\mu} = \pm(1, 2)$ with isotropic noise ($\varepsilon \sim \mathcal{N}(0, 0.5^2 \boldsymbol{I}_2)$) and assign labels $+1$ and $-1$ accordingly. This dataset is used to evaluate whether the one-point phenomena from Theorem 3.2 and Corollary 3.5 persist in the multi-point regime.

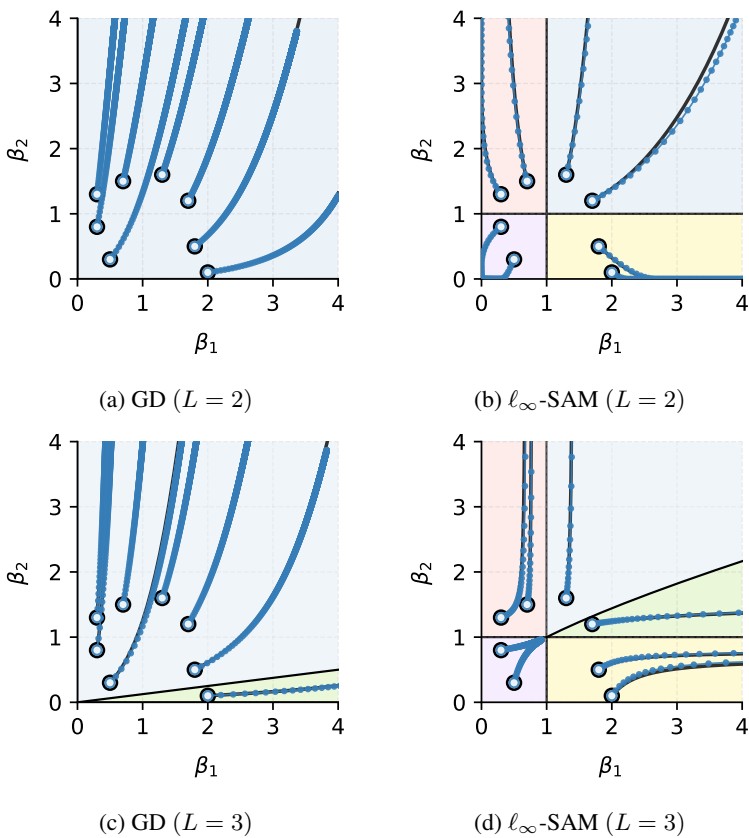

(a) GD ($L = 2$)

(b) $\ell_\infty$-SAM ($L = 2$)

(c) GD ($L = 3$)

(d) $\ell_\infty$-SAM ($L = 3$)

Figure 13: Trajectories $\boldsymbol{\beta}(t)$ under GF, $\ell_\infty$-SAM flow (black line), GD, and discrete $\ell_\infty$-SAM updates (blue dots) for $d = 2$ on random multi-point dataset in Figure 12. For SAM, we set $\rho = 1$. For GD and discrete $\ell_\infty$-SAM, we use step size $\eta = 0.1$.

# D  SAM with $\ell_2$-perturbations: Proof of Section 4

## D.1  Depth-1 Networks: Proof of Theorem 4.1

**Theorem 4.1.** *For almost every dataset which is linearly separable, any perturbation radius $\rho$ and any initialization, consider the linear model $f(\boldsymbol{x}) = \langle \boldsymbol{w}, \boldsymbol{x} \rangle$ trained with logistic loss. Then, $\ell_2$-SAM flow directionally converges in the $\ell_2$ max-margin direction.*

*Proof.* Apply Lemma B.1 with $\boldsymbol{\varepsilon}(\boldsymbol{w}) = \rho \frac{\nabla \mathcal{L}(\boldsymbol{\theta})}{\|\nabla \mathcal{L}(\boldsymbol{\theta})\|_2}$. Then $\|\boldsymbol{\varepsilon}(\boldsymbol{w})\|_2 \leq \rho$ for all $\boldsymbol{w}$, so the conditions of Lemma B.1 hold. Thus, the flow converges to the $\ell_2$ max-margin direction. $\qquad\square$

**Theorem D.1.** *Consider the linear model $f(\boldsymbol{x}) = \langle \boldsymbol{w}, \boldsymbol{x} \rangle$ trained on the dataset $\mathcal{D}_{\boldsymbol{\mu}}$ with loss $\mathcal{L}(\boldsymbol{w}) = \ell(\langle \boldsymbol{w}, \boldsymbol{x} \rangle)$ where $\ell'(u) < 0$ for all $u$. Then, GF and $\ell_2$-SAM flow, starting from any $\boldsymbol{w}(0)$, evolve on the same affine line $\boldsymbol{w}(0) + \mathrm{span}\{\boldsymbol{\mu}\}$ and have the same spatial trajectory.*

*Proof.* The model is $f(\boldsymbol{x}) = \langle \boldsymbol{w}, \boldsymbol{x} \rangle = \boldsymbol{w}^\top \boldsymbol{x}$. The loss is $\mathcal{L}(\boldsymbol{w}) = \ell(\boldsymbol{w}^\top \boldsymbol{\mu})$. The gradient is $\nabla_{\boldsymbol{w}} \mathcal{L}(\boldsymbol{w}) = \ell'(\boldsymbol{w}^\top \boldsymbol{\mu}) \cdot \boldsymbol{\mu}$ with $\ell'(s) < 0$.

**Gradient Descent**  GF is

$$
\dot{\boldsymbol{w}} = -\nabla_{\boldsymbol{w}} \mathcal{L}(\boldsymbol{w})
$$
$$
= -\ell'(\boldsymbol{w}^\top \boldsymbol{\mu}) \cdot \boldsymbol{\mu}.
$$

**SAM with $\ell_2$ perturbation**  The ascent point is

$$
\hat{\boldsymbol{w}} = \boldsymbol{w} + \rho \boldsymbol{\varepsilon}_2(\boldsymbol{w})
$$
$$
= \boldsymbol{w} + \rho \frac{\nabla_{\boldsymbol{w}} \mathcal{L}(\boldsymbol{w})}{\|\nabla_{\boldsymbol{w}} \mathcal{L}(\boldsymbol{w})\|_2}
$$
$$
= \boldsymbol{w} - \rho \frac{\boldsymbol{\mu}}{\|\boldsymbol{\mu}\|_2}.
$$

The update of $\ell_2$-SAM flow is

$$
\dot{\boldsymbol{w}} = -\nabla_{\boldsymbol{w}} \mathcal{L}(\hat{\boldsymbol{w}})
$$
$$
= -\nabla_{\boldsymbol{w}} \mathcal{L}(\boldsymbol{w} - \rho \frac{\boldsymbol{\mu}}{\|\boldsymbol{\mu}\|_2})
$$
$$
= -\ell'(\boldsymbol{w}^\top \boldsymbol{\mu} - \rho \frac{\boldsymbol{\mu}^\top \boldsymbol{\mu}}{\|\boldsymbol{\mu}\|_2}) \cdot \boldsymbol{\mu}
$$
$$
= -\ell'(\boldsymbol{w}^\top \boldsymbol{\mu} - \rho \|\boldsymbol{\mu}\|_2) \cdot \boldsymbol{\mu}.
$$

Therefore, they have the same spatial trajectory as:

$$
\dot{\boldsymbol{w}} = \boldsymbol{\mu}.
$$

The term $-\ell'(\boldsymbol{w}^\top \boldsymbol{\mu} - \rho \|\boldsymbol{\mu}\|_2)$ is the accelation in terms of $t$ since $-\ell'(s)$ is decreasing in $s$. $\qquad\square$

## D.2  Derivation of $\ell_2$-SAM flow

Let us get the $\ell_2$-SAM flow. The gradient is

$$
\nabla_{\boldsymbol{w}^{(i)}} L(\boldsymbol{\theta}) = \ell'\big(\langle \boldsymbol{\beta}(\boldsymbol{\theta}), \boldsymbol{\mu} \rangle\big) \nabla_{\boldsymbol{w}^{(i)}} \langle \boldsymbol{\beta}(\boldsymbol{\theta}), \boldsymbol{\mu} \rangle
$$
$$
= \ell'\big(\langle \boldsymbol{\beta}(\boldsymbol{\theta}), \boldsymbol{\mu} \rangle\big) \boldsymbol{\mu} \odot \boldsymbol{w}^{(\ell)} \qquad\qquad \text{for } (i, l) \in \{(1, 2), (2, 1)\}.
$$

From the gradient, we have

$$
\boldsymbol{\varepsilon}_2^{(i)}(\boldsymbol{\theta}) = \rho \frac{\nabla_{\boldsymbol{w}^{(i)}} \mathcal{L}(\boldsymbol{\theta})}{\|\nabla \mathcal{L}(\boldsymbol{\theta})\|_2} \underset{(a)}{=} -\rho \frac{\boldsymbol{\mu} \odot \boldsymbol{w}^{(\ell)}}{\sqrt{\|\boldsymbol{\mu} \odot \boldsymbol{w}^{(1)}\|_2^2 + \|\boldsymbol{\mu} \odot \boldsymbol{w}^{(2)}\|_2^2}} = -\rho \frac{\boldsymbol{\mu} \odot \boldsymbol{w}^{(\ell)}}{n_{\boldsymbol{\theta}}}
$$

for $(i, l) \in \{(1,2), (2,1)\}$, where $n_{\boldsymbol{\theta}} = \sqrt{\|\boldsymbol{\mu} \odot \boldsymbol{w}^{(1)}\|_2^2 + \|\boldsymbol{\mu} \odot \boldsymbol{w}^{(2)}\|_2^2}$ and (a) follows from $\ell'(u) = -\frac{1}{1+e^u} < 0$.

We consider the initialization $\boldsymbol{w}^{(1)}(0) = \boldsymbol{w}^{(2)}(0) \in \mathbb{R}_+^d$. Then, since the loss function and dynamics are invariant under exchanging $\boldsymbol{w}^{(1)}$ and $\boldsymbol{w}^{(2)}$, we have $\boldsymbol{w}^{(1)}(t) = \boldsymbol{w}^{(2)}(t) =: \boldsymbol{w}(t)$ for all $t \geq 0$. Therefore, the update on $\boldsymbol{w}(t)$ by rescaled $\ell_2$-SAM flow is given as

$$\dot{\boldsymbol{w}}(t) = \boldsymbol{\mu} \odot \left( \boldsymbol{w}(t) - \rho \frac{\boldsymbol{\mu} \odot \boldsymbol{w}(t)}{n_{\boldsymbol{\theta}}(t)} \right).$$

## D.3 Proof of Theorem 4.2

**Theorem 4.2.** *For almost every dataset which is linearly separable, and any perturbation radius $\rho$, consider the linear diagonal network of depth 2, $f(\boldsymbol{x}) = \langle \boldsymbol{w}^{(1)} \odot \boldsymbol{w}^{(2)}, \boldsymbol{x} \rangle$ trained with logistic loss. Let $(\boldsymbol{w}^{(1)}(t), \boldsymbol{w}^{(2)}(t))$ follow the $\ell_2$-SAM flow with $\boldsymbol{w}^{(1)}(0) = \boldsymbol{w}^{(2)}(0)$. Assume (a) the loss vanishes $\mathcal{L}(\boldsymbol{w}^{(1)}(t), \boldsymbol{w}^{(2)}(t)) \to 0$, (b) the predictor $\boldsymbol{\beta}(t) := \boldsymbol{w}^{(1)}(t) \odot \boldsymbol{w}^{(2)}(t)$ converges in direction. Then the limit direction of $\boldsymbol{\beta}(t)$ is the $\ell_1$ max-margin direction.*

*Proof.* Let $\{(\boldsymbol{x}_n, y_n)\}_{n=1}^N \subset \mathbb{R}^d \times \{\pm 1\}$ be a linearly separable dataset, meaning that there exists $\boldsymbol{w}_* \in \mathbb{R}^d$ such that

$$y_n \boldsymbol{x}_n^\top \boldsymbol{w}_* > 0 \qquad \forall n.$$

As usual, we absorb the labels into the inputs by redefining $\boldsymbol{x}_n \leftarrow y_n \boldsymbol{x}_n$, so that we may assume $y_n = 1$ for all $n$ and

$$\exists \boldsymbol{w}_* \text{ such that } \boldsymbol{x}_n^\top \boldsymbol{w}_* > 0 \ \forall n.$$

We consider a depth-2 diagonal linear network with parameters $\boldsymbol{w}_1, \boldsymbol{w}_2 \in \mathbb{R}^d$, defining the predictor

$$f(\boldsymbol{x}; \boldsymbol{w}_1, \boldsymbol{w}_2) = (\boldsymbol{w}_1 \odot \boldsymbol{w}_2)^\top \boldsymbol{x} = \boldsymbol{\beta}^\top \boldsymbol{x}, \qquad \boldsymbol{\beta} := \boldsymbol{w}_1 \odot \boldsymbol{w}_2.$$

The loss function is logistic:

$$\mathcal{L}(\boldsymbol{w}_1, \boldsymbol{w}_2) = \sum_{n=1}^N \ell(\boldsymbol{\beta}^\top \boldsymbol{x}_n), \qquad \ell(u) = \log(1 + e^{-u}), \qquad \ell'(u) = -\frac{e^{-u}}{1 + e^{-u}}.$$

We study the $\ell_2$-SAM flow with fixed perturbation radius $\rho > 0$:

$$\dot{\boldsymbol{w}}_1(t) = -\nabla_{\boldsymbol{w}_1} \mathcal{L}(\widehat{\boldsymbol{w}}_1(t), \widehat{\boldsymbol{w}}_2(t)), \qquad \dot{\boldsymbol{w}}_2(t) = -\nabla_{\boldsymbol{w}_2} \mathcal{L}(\widehat{\boldsymbol{w}}_1(t), \widehat{\boldsymbol{w}}_2(t)),$$

where

$$\widehat{\boldsymbol{w}}_i(t) = \boldsymbol{w}_i(t) + \rho \frac{\nabla_{\boldsymbol{w}_i} \mathcal{L}(\boldsymbol{w}_1(t), \boldsymbol{w}_2(t))}{\|\nabla_{\boldsymbol{w}_i} \mathcal{L}(\boldsymbol{w}_1(t), \boldsymbol{w}_2(t))\|_2}, \qquad i = 1, 2.$$

**Step 1: Balanced initialization removes layer imbalance.** Let

$$z_j(t) := w_j^{(1)}(t) - w_j^{(2)}(t).$$

From the SAM flow and

$$\frac{\partial \mathcal{L}}{\partial w_j^{(1)}}(\widehat{\boldsymbol{w}}) = \sum_{n=1}^N \ell'(\widehat{\boldsymbol{\beta}}^\top \boldsymbol{x}_n) x_{n,j} \widehat{w}_j^{(2)}, \qquad \frac{\partial \mathcal{L}}{\partial w_j^{(2)}}(\widehat{\boldsymbol{w}}) = \sum_{n=1}^N \ell'(\widehat{\boldsymbol{\beta}}^\top \boldsymbol{x}_n) x_{n,j} \widehat{w}_j^{(1)},$$

one obtains

$$\dot{z}_j(t) = -G_j(t)\big(w_j^{(2)}(t) - w_j^{(1)}(t)\big)(1 + o(1)), \qquad G_j(t) = \sum_{n=1}^N \ell'(\widehat{\boldsymbol{\beta}}^\top \boldsymbol{x}_n) x_{n,j}.$$

Here the factor $1 + o(1)$ arises because the gradients in the SAM update are evaluated at the perturbed parameter

$$\widehat{\boldsymbol{w}}(t) = \boldsymbol{w}(t) + \rho \frac{\nabla \mathcal{L}(\boldsymbol{w}(t))}{\|\nabla \mathcal{L}(\boldsymbol{w}(t))\|_2},$$

rather than at $\boldsymbol{w}(t)$ itself. Since the perturbation has fixed magnitude $\rho$ while the parameter norm satisfies $\|\boldsymbol{w}(t)\| \to \infty$ along any vanishing-loss trajectory of a 2-homogeneous model, the relative perturbation decays:

$$\frac{\|\widehat{\boldsymbol{w}}(t) - \boldsymbol{w}(t)\|_2}{\|\boldsymbol{w}(t)\|_2} = \frac{\rho}{\|\boldsymbol{w}(t)\|_2} \longrightarrow 0.$$

Consequently, the gradients $\nabla\mathcal{L}(\widehat{\boldsymbol{w}}(t))$ and $\nabla\mathcal{L}(\boldsymbol{w}(t))$ become asymptotically colinear, and replacing the latter by the former introduces only a vanishing multiplicative error $1 + o(1)$ in the imbalance ODE for $z_j(t)$.

Since $z_j(0) = 0$ under balanced initialization and the ODE $\dot{z}_j(t) = -G_j(t)z_j(t)(1 + o(1))$ is linear with a Lipschitz right-hand side, uniqueness of solutions implies $z_j(t) \equiv 0$ for all $t$. Hence for all $t$

$$w_j^{(1)}(t) = w_j^{(2)}(t) =: a_j(t), \qquad \beta_j(t) = a_j(t)^2.$$

**Step 2: Predictor ODE.** From the SAM ODE,

$$\dot{a}_j(t) = -a_j(t)\, G_j(t)\, (1 + o(1)).$$

Hence

$$\dot{\beta}_j(t) = 2a_j(t)\dot{a}_j(t) = -2a_j(t)^2 G_j(t)(1 + o(1)) = -2\beta_j(t)G_j(t)(1 + o(1)).$$

Noting that

$$\nabla_{\boldsymbol{\beta}}\mathcal{L}(\boldsymbol{\beta})_j = \sum_{n=1}^{N} \ell'(\boldsymbol{\beta}^\top \boldsymbol{x}_n)\, x_{n,j},$$

since

$$G_j(t) = \sum_{n=1}^{N} \ell'(\widehat{\boldsymbol{\beta}}^\top \boldsymbol{x}_n)\, x_{n,j} = \sum_{n=1}^{N} \ell'(\boldsymbol{\beta}(t)^\top \boldsymbol{x}_n)\, x_{n,j}\, (1 + o(1)),$$

we have

$$G_j(t) = \nabla_{\beta_j}\mathcal{L}(\boldsymbol{\beta}(t))\, (1 + o(1)).$$

Hence the coordinate-wise predictor dynamics

$$\dot{\beta}_j(t) = -2\, \beta_j(t)\, G_j(t)\, (1 + o(1))$$

become

$$\dot{\beta}_j(t) = -2\, \beta_j(t)\, \nabla_{\beta_j}\mathcal{L}(\boldsymbol{\beta}(t))\, (1 + o(1)).$$

Writing this in vector form using $\mathrm{diag}(\boldsymbol{\beta})\nabla_{\boldsymbol{\beta}}\mathcal{L} = (\beta_1\nabla_{\beta_1}\mathcal{L}, \ldots, \beta_d\nabla_{\beta_d}\mathcal{L})^\top$, we obtain

$$\dot{\boldsymbol{\beta}}(t) = -2\, \mathrm{diag}(\boldsymbol{\beta}(t))\, \nabla_{\boldsymbol{\beta}}\mathcal{L}(\boldsymbol{\beta}(t))\, (1 + o(1)). \tag{11}$$

**Step 3: Geometry induced by the diagonal parameterization.** To characterize the optimization geometry associated with the depth-2 diagonal model, we invoke Lemma D.2. The lemma shows that, for the parameterization

$$\boldsymbol{\beta} = \boldsymbol{w}^{(1)} \odot \boldsymbol{w}^{(2)} \qquad \text{and} \qquad R(\boldsymbol{w}^{(1)}, \boldsymbol{w}^{(2)}) = \tfrac{1}{2}\big(\|\boldsymbol{w}^{(1)}\|_2^2 + \|\boldsymbol{w}^{(2)}\|_2^2\big),$$

the induced predictor norm is exactly the $\ell_1$ norm:

$$\|\boldsymbol{\beta}\|_{\mathcal{N}} := \min_{\boldsymbol{w}^{(1)} \odot \boldsymbol{w}^{(2)} = \boldsymbol{\beta}} R(\boldsymbol{w}^{(1)}, \boldsymbol{w}^{(2)}) = \|\boldsymbol{\beta}\|_1.$$

Moreover, on the balanced submanifold $\boldsymbol{w}^{(1)} = \boldsymbol{w}^{(2)} = \boldsymbol{a}$ with $\boldsymbol{\beta} = \boldsymbol{a}^{\odot 2}$, the lemma establishes that the Riemannian metric induced on predictor space is

$$\langle \boldsymbol{u}, \boldsymbol{v} \rangle_{\mathcal{N}} = \boldsymbol{u}^\top M(\boldsymbol{\beta})\boldsymbol{v}, \qquad M(\boldsymbol{\beta}) = 2\,\mathrm{diag}(\boldsymbol{\beta}).$$

Therefore, the natural-gradient steepest-descent flow with respect to the induced norm $\|\cdot\|_{\mathcal{N}}$ takes the form

$$\dot{\boldsymbol{\beta}}(t) = -M(\boldsymbol{\beta}(t))\, \nabla_{\boldsymbol{\beta}}\mathcal{L}(\boldsymbol{\beta}(t)) = -2\,\mathrm{diag}(\boldsymbol{\beta}(t))\, \nabla_{\boldsymbol{\beta}}\mathcal{L}(\boldsymbol{\beta}(t)).$$

We next compare this asymptotic steepest-descent flow with the predictor ODE arising from the $\ell_2$-SAM dynamics.

**Step 4: Asymptotic identification with $\ell_1$ steepest descent.** Comparing equation 11 with the steepest-descent flow above shows that the SAM predictor dynamics coincide with the $\ell_1$ steepest-descent dynamics up to a multiplicative factor $1 + o(1)$ and a vanishing perturbation. Assumptions (a) and (b) guarantee that these perturbations do not change the limiting direction of $\boldsymbol{\beta}(t)/\|\boldsymbol{\beta}(t)\|_2$.

**Step 5: Conclude $\ell_1$ max-margin.** By the max-margin theorem for steepest descent in a given norm (Gunasekar et al. (2018a), Thm. 5; extended to logistic loss by Lyu & Li (2019)), any trajectory following $\ell_1$ steepest descent and satisfying $\mathcal{L}(\boldsymbol{\beta}(t)) \to 0$ converges in direction to the $\ell_1$ max-margin solution. Since the SAM predictor dynamics are asymptotically equivalent to $\ell_1$ steepest descent, and by (b) the direction limit exists, we obtain

$$\bar{\boldsymbol{\beta}} \parallel \boldsymbol{\beta}^\star, \qquad \boldsymbol{\beta}^\star \in \arg\min_{\boldsymbol{\beta}} \|\boldsymbol{\beta}\|_1 \text{ s.t. } \boldsymbol{\beta}^\top \boldsymbol{x}_n \geq 1.$$

$\square$

**Lemma D.2** (Induced Norm and Natural Gradient Metric for Depth-2 Diagonal Models). *Consider the depth-2 diagonal parameterization*

$$\boldsymbol{\beta} = \boldsymbol{w}^{(1)} \odot \boldsymbol{w}^{(2)} \in \mathbb{R}^d,$$

*and the quadratic parameter regularizer*

$$R(\boldsymbol{w}^{(1)}, \boldsymbol{w}^{(2)}) := \frac{1}{2} \left( \|\boldsymbol{w}^{(1)}\|_2^2 + \|\boldsymbol{w}^{(2)}\|_2^2 \right).$$

*Then the induced predictor norm*

$$\|\boldsymbol{\beta}\|_{\mathcal{N}} := \min_{\boldsymbol{w}^{(1)} \odot \boldsymbol{w}^{(2)} = \boldsymbol{\beta}} R(\boldsymbol{w}^{(1)}, \boldsymbol{w}^{(2)})$$

*satisfies*

$$\|\boldsymbol{\beta}\|_{\mathcal{N}} = \|\boldsymbol{\beta}\|_1.$$

*Moreover, on the submanifold where $\boldsymbol{w}^{(1)} = \boldsymbol{w}^{(2)} = \boldsymbol{a}$ and $\boldsymbol{\beta} = \boldsymbol{a}^{\odot 2}$, the Riemannian metric induced on the predictor space by $R$ is*

$$\langle \boldsymbol{u}, \boldsymbol{v} \rangle_{\mathcal{N}} = \boldsymbol{u}^\top M(\boldsymbol{\beta}) \boldsymbol{v}, \qquad M(\boldsymbol{\beta}) = 2 \operatorname{diag}(\boldsymbol{\beta}).$$

*Consequently, the natural-gradient steepest-descent flow w.r.t. $\|\cdot\|_{\mathcal{N}}$ is*

$$\dot{\boldsymbol{\beta}} = -M(\boldsymbol{\beta}) \nabla_{\boldsymbol{\beta}} \mathcal{L}(\boldsymbol{\beta}) = -2 \operatorname{diag}(\boldsymbol{\beta}) \nabla_{\boldsymbol{\beta}} \mathcal{L}(\boldsymbol{\beta}).$$

*Proof.* **(i) Computation of the induced norm.** For each coordinate $j$, the constraint $\beta_j = w_j^{(1)} w_j^{(2)}$ decouples. If $\beta_j = 0$, the minimum is attained at $(w_j^{(1)}, w_j^{(2)}) = (0, 0)$ and equals $0 = |\beta_j|$.

For $\beta_j \neq 0$, eliminate $w_j^{(2)}$ via $w_j^{(2)} = \beta_j / w_j^{(1)}$ and minimize

$$\phi_j(w) := \frac{1}{2} \left( w^2 + \frac{\beta_j^2}{w^2} \right), \qquad w \neq 0.$$

Differentiation yields $\phi_j'(w) = w - \beta_j^2 w^{-3}$, whose nonzero roots satisfy $w^4 = \beta_j^2$, so that $|w| = |\beta_j|^{1/2}$. Substitution gives $\phi_j(w^\star) = |\beta_j|$. Summing over $j$ yields the induced norm

$$\|\boldsymbol{\beta}\|_{\mathcal{N}} = \sum_{j=1}^d |\beta_j| = \|\boldsymbol{\beta}\|_1.$$

**(ii) Local parametrization and Jacobian.** On the balanced submanifold $\boldsymbol{w}^{(1)} = \boldsymbol{w}^{(2)} = \boldsymbol{a} \in \mathbb{R}^d$, the predictor is

$$\beta_j = a_j^2.$$

Hence the Jacobian of the map $\boldsymbol{a} \mapsto \boldsymbol{\beta}$ is diagonal:

$$\frac{\partial \beta_j}{\partial a_k} = 2 a_j \, \delta_{jk}.$$

**(iii) Riemannian metric induced from $R$.** The regularizer restricted to $\boldsymbol{a}$ becomes

$$R(\boldsymbol{a}, \boldsymbol{a}) = \|\boldsymbol{a}\|_2^2.$$

Thus the parameter-space metric is Euclidean on $\boldsymbol{a}$. For a tangent predictor perturbation $\mathrm{d}\boldsymbol{\beta}$, the corresponding parameter perturbation is

$$\mathrm{d}a_j = \frac{\mathrm{d}\beta_j}{2a_j} = \frac{\mathrm{d}\beta_j}{2\sqrt{\beta_j}}.$$

Thus the squared parameter differential is

$$\|\mathrm{d}\boldsymbol{a}\|_2^2 = \sum_{j=1}^{d} \left( \frac{\mathrm{d}\beta_j}{2\sqrt{\beta_j}} \right)^2 = \sum_{j=1}^{d} \frac{(\mathrm{d}\beta_j)^2}{4\,\beta_j}.$$

Therefore the predictor-space inner product induced by $R$ is

$$\langle \boldsymbol{u}, \boldsymbol{v} \rangle_{\mathcal{N}} = \sum_{j=1}^{d} \frac{u_j v_j}{4\beta_j}.$$

Equivalently,

$$M(\boldsymbol{\beta})^{-1} = \frac{1}{4}\,\mathrm{diag}(\beta_1^{-1}, \ldots, \beta_d^{-1}).$$

Inverting yields

$$M(\boldsymbol{\beta}) = 4\,\mathrm{diag}(\beta_1, \ldots, \beta_d).$$

**(iv) Removal of irrelevant constant factor.** Steepest-descent flows are invariant to multiplication of $M$ by any positive scalar constant. Thus $M(\boldsymbol{\beta})$ is equivalent, for optimization dynamics, to

$$M(\boldsymbol{\beta}) = 2\,\mathrm{diag}(\boldsymbol{\beta}),$$

which is the conventional normalization in the induced-norm literature.

**(v) Natural gradient flow.** By definition of steepest descent under the induced norm,

$$\dot{\boldsymbol{\beta}} = -M(\boldsymbol{\beta})\,\nabla_{\boldsymbol{\beta}}\mathcal{L}(\boldsymbol{\beta}) = -2\,\mathrm{diag}(\boldsymbol{\beta})\,\nabla_{\boldsymbol{\beta}}\mathcal{L}(\boldsymbol{\beta}).$$

$\square$

## D.4 PROOFS FOR SECTION 4.2.3

In this section, we provide detailed proofs for the trajectory analysis of SAM flow, with a focus on the roles of the initialization scale $\alpha$, the perturbation radius $\rho$, and the feature vector $\boldsymbol{\mu}$. For notational simplicity, we omit the time dependence $(t)$ when the context is clear. The full SAM flow differs from the rescaled flow by the positive scalar factor $\hat{\lambda}(t)$, so the coordinate-wise ordering of growth rates is unchanged. For the rescaled flow, this factor is omitted.

**Assumption D.3.** The initial weight parameters are positive and symmetric: $\boldsymbol{w}^{(1)}(0) = \boldsymbol{w}^{(2)}(0) = \alpha\mathbf{1}$ for some scaling factor $\alpha > 0$.

**Assumption D.4.** The vector $\boldsymbol{\mu}$ has strictly positive, increasing coordinates: $0 < \mu_1 < \cdots < \mu_d$. (Equivalently, up to a fixed permutation we may assume the coordinates are monotone.)

We introduce two auxiliary quantities. Define the normalized weights $p_j(t) := \frac{\mu_j^2 \beta_j(t)}{\sum_{k=1}^{d} \mu_k^2 \beta_k(t)}$ and their moments $M_k(t) := \sum_{j=1}^{d} \mu_j^k p_j(t)$. Using these, we set the thresholds

$$m_{\mathrm{L}} := \frac{\mu_1}{2}, \qquad m_{\mathrm{H}}(t) := \frac{M_2(t)}{2M_1(t)}.$$

In the proof, we consider $\ell(\langle \boldsymbol{\beta}, \boldsymbol{\mu} \rangle)$ term, so not only considering the spatial trajectory but full gradient flow without any reparameterization. We define the margins at the current and perturbed parameters as $s(t) := \langle \boldsymbol{\beta}(t), \boldsymbol{\mu} \rangle$ and $\hat{s}(t) := \langle \hat{\boldsymbol{\beta}}(t), \boldsymbol{\mu} \rangle$. Set $\hat{\lambda}(t) := |\ell'(\hat{s}(t))|$, the slope of the loss with respect to the margin evaluated at the perturbed margin.

### D.4.1 RECAP: BASIC NOTATION

Recall the margin $s = \langle \boldsymbol{\beta}, \boldsymbol{\mu} \rangle$ and the loss $\mathcal{L}(s) = \log\left(1 + \exp(-s)\right)$. The derivatives of the loss with respect to the margin $s$ are:

$$\frac{d\mathcal{L}}{ds} = -\sigma(-s) = -\frac{1}{1 + \exp(s)},$$

$$\frac{d^2\mathcal{L}}{ds^2} = \sigma(s)\sigma(-s) > 0,$$

where $\sigma(s) = (1 + \exp(-s))^{-1}$ is the sigmoid function. We define $\lambda := \sigma(-s) \in (0,1)$ as the logistic loss slope magnitude. The gradients with respect to the weight parameters, obtained via the chain rule, are:

$$\frac{d\mathcal{L}}{dw_j^{(1)}} := \frac{d\mathcal{L}}{ds}\frac{ds}{dw_j^{(1)}} = -\lambda\mu_j w_j^{(2)}, \qquad \frac{d\mathcal{L}}{dw_j^{(2)}} := \frac{d\mathcal{L}}{ds}\frac{ds}{dw_j^{(2)}} = -\lambda\mu_j w_j^{(1)}.$$

The squared norm of the gradient vector is:

$$\|\nabla_{\boldsymbol{\theta}}\mathcal{L}\|^2 = \sum_{j=1}^{d}\lambda^2\mu_j^2\left(\left(w_j^{(2)}\right)^2 + \left(w_j^{(1)}\right)^2\right) = \lambda^2 n_{\boldsymbol{\theta}}^2,$$

where $n_{\boldsymbol{\theta}} := \sqrt{\sum_{j=1}^{d}\mu_j^2\left(\left(w_j^{(1)}\right)^2 + \left(w_j^{(2)}\right)^2\right)}$. SAM perturbs parameters by taking a step of size $\rho$ along the normalized gradient direction.

$$\varepsilon_2 := \rho\frac{\nabla_{\boldsymbol{\theta}}\mathcal{L}}{\|\nabla_{\boldsymbol{\theta}}\mathcal{L}\|_2},$$

$$(\varepsilon_2)_{w_j^{(1)}} = -\frac{\rho\mu_j w_j^{(2)}}{n_{\boldsymbol{\theta}}},$$

$$(\varepsilon_2)_{w_j^{(2)}} = -\frac{\rho\mu_j w_j^{(1)}}{n_{\boldsymbol{\theta}}}.$$

The perturbed weight parameters are

$$(\hat{w}_1)_j := w_j^{(1)} - \frac{\rho\mu_j w_j^{(2)}}{n_{\boldsymbol{\theta}}}, \qquad (\hat{w}_2)_j := w_j^{(2)} - \frac{\rho\mu_j w_j^{(1)}}{n_{\boldsymbol{\theta}}}.$$

The perturbed $\beta_j$ becomes

$$\begin{aligned}
\hat{\beta}_j &:= \hat{w}_j^{(1)}\hat{w}_j^{(2)} \\
&= w_j^{(1)}w_j^{(2)} - \frac{\rho\mu_j}{n_{\boldsymbol{\theta}}}\left(\left(w_j^{(1)}\right)^2 + \left(w_j^{(2)}\right)^2\right) + \frac{\rho^2\mu_j^2}{n_{\boldsymbol{\theta}}^2}w_j^{(1)}w_j^{(2)} \\
&= \beta_j\left(1 + \frac{\rho^2\mu_j^2}{n_{\boldsymbol{\theta}}^2}\right) - \frac{\rho\mu_j}{n_{\boldsymbol{\theta}}}\left(\left(w_j^{(1)}\right)^2 + \left(w_j^{(2)}\right)^2\right).
\end{aligned}$$

The perturbed margin and loss slope magnitude are

$$\hat{s} := \langle\hat{\boldsymbol{\beta}}, \boldsymbol{\mu}\rangle = \sum_{j=1}^{d}\mu_j\hat{\beta}_j, \qquad \hat{\lambda} := \sigma(-\hat{s}).$$

Recall that the SAM flow dynamics are given by:

$$\dot{\boldsymbol{\theta}} = -\nabla_{\boldsymbol{\theta}}\mathcal{L}(\hat{\boldsymbol{\theta}}).$$

### D.4.2 PRELIMINARY ANALYSIS

We first establish a key property of the SAM flow: the balancedness of the weights.

**Lemma D.5.** *Under Assumption D.4, the SAM flow decays the quantity $w_j^{(1)}(t) - w_j^{(2)}(t)$ exponentially to zero.*

*Proof.* Define $\Delta_j := w_j^{(1)} - w_j^{(2)}$. The SAM dynamics yield

$$\dot{w}_j^{(1)} = \hat{\lambda}\mu_j \hat{w}_j^{(2)}, \qquad \dot{w}_j^{(2)} = \hat{\lambda}\mu_j \hat{w}_j^{(1)}.$$

The time derivative of $\Delta_j$ is

$$\begin{aligned}
\dot{\Delta}_j &= \dot{w}_j^{(1)} - \dot{w}_j^{(2)} \\
&= \hat{\lambda}\mu_j \hat{w}_j^{(2)} - \hat{\lambda}\mu_j \hat{w}_j^{(1)} \\
&= \hat{\lambda}\mu_j \left( w_j^{(2)} - \frac{\rho\mu_j w_j^{(1)}}{n_{\boldsymbol{\theta}}} \right) - \hat{\lambda}\mu_j \left( w_j^{(1)} - \frac{\rho\mu_j w_j^{(2)}}{n_{\boldsymbol{\theta}}} \right) \\
&= -\hat{\lambda}\mu_j \left( 1 + \frac{\rho\mu_j}{n_{\boldsymbol{\theta}}} \right) \Delta_j.
\end{aligned}$$

Since $\hat{\lambda}$ is positive and $\mu_j > 0$, it gives exponential decay.

$$\Delta_j(T) = \Delta_j(0) \cdot \exp\left( -\mu_j \int_0^T \hat{\lambda}\left( 1 + \frac{\rho\mu_j}{n_{\boldsymbol{\theta}}} \right) \mathrm{d}t \right).$$

Hence, the quantity $w_j^{(1)}(t) - w_j^{(2)}(t)$ decays exponentially. $\qquad\square$

**Proposition D.6.** *Under initialization with $w_j^{(1)}(0) = w_j^{(2)}(0)$ and Assumption D.4, the equality $w_j^{(1)}(t) = w_j^{(2)}(t)$ is preserved for all $t \geq 0$. Furthermore, the sign of $w_j^{(1)}(t)$ and $w_j^{(2)}(t)$ remains unchanged throughout the dynamics.*

*Proof.* With $w_j^{(1)}(0) = w_j^{(2)}(0)$, we have $\Delta_j(0) = w_j^{(1)}(0) - w_j^{(2)}(0) = 0$. By Lemma D.5, $\Delta_j(t) = 0$ for all $t \geq 0$. Given this balancedness, each coordinate evolves multiplicatively according to

$$\dot{w}_j^{(1)} = \hat{\lambda}\mu_j \hat{w}_j^{(2)} = \hat{\lambda}\mu_j \left( w_j^{(1)} - \frac{\rho\mu_j w_j^{(1)}}{n_{\boldsymbol{\theta}}} \right) = \hat{\lambda}\mu_j \left( 1 - \frac{\rho\mu_j}{n_{\boldsymbol{\theta}}} \right) w_j^{(1)}.$$

This differential equation has the unique solution

$$w_j^{(1)}(T) = w_j^{(1)}(0) \cdot \exp\left( \mu_j \cdot \int_0^T \hat{\lambda}(t) \left( 1 - \frac{\rho\mu_j}{n_{\boldsymbol{\theta}}} \right) \mathrm{d}t \right).$$

Since the exponential function is always positive, $w_j^{(1)}(t)$ and $w_j^{(2)}(t)$ maintain the same sign as their initial values throughout the dynamics. $\qquad\square$

### D.4.3 PROOF OF LEMMA 4.3

We begin by restating Lemma 4.3.

**Lemma 4.3.** *The rescaled $\ell_2$-SAM flow (2) is $\dot{\beta}_j(t) = r_j(t)\beta_j(t)$ with $r_j(t) := 2\mu_j \left( 1 - \frac{\rho\mu_j}{n_{\boldsymbol{\theta}}(t)} \right)$.*

*Proof.* Under Assumption D.3 and Assumption D.4, the Proposition D.6 holds, which ensures that $w_j^{(1)} = w_j^{(2)} = \sqrt{\beta_j}$ for all $t \geq 0$. So we have

$$\left( w_j^{(1)} \right)^2 + \left( w_j^{(2)} \right)^2 = 2\beta_j, \qquad n_{\boldsymbol{\theta}}^2 = 2\sum_{j=1}^d \mu_j^2 \beta_j.$$

The evolution equation for $\beta_j$ is

$$\dot{\beta}_j = \dot{w}_j^{(1)} w_j^{(2)} + w_j^{(1)} \dot{w}_j^{(2)}$$
$$= 2\hat{\lambda}\mu_j\beta_j\left(1 - \frac{\rho\mu_j}{n_{\boldsymbol{\theta}}}\right). \tag{12}$$

This yields

$$\beta_j(T) = \beta_j(0)\cdot\exp\left(2\mu_j\int_0^T\hat{\lambda}\left(1 - \frac{\rho\mu_j}{n_{\boldsymbol{\theta}}}\right)\mathrm{d}t\right).$$

Let $r_j := 2\hat{\lambda}\mu_j\left(1 - \frac{\rho\mu_j}{n_{\boldsymbol{\theta}}}\right)$. When $r_j > 0$, $\beta_j$ grows locally exponentially. Otherwise, it decays locally exponentially. The key insight is that each $\beta_j$'s growth rate depends on the interaction between the gradient magnitude $\hat{\lambda}$ and the perturbation term $\frac{\rho\mu_j}{n_{\boldsymbol{\theta}}}$. This interaction drives SAM's implicit bias. $\qquad\square$

### D.4.4 PRELIMINARY ANALYSIS FOR $m_{\mathrm{c}}(t)$ TRAJECTORY ANALYSIS

Before proving Theorem 4.4, we establish some preliminary results that will be used in the proof.

**Lemma D.7.** *Under Assumption D.3 and Assumption D.4, the time derivative of $m_{\mathrm{c}}(t)$ is given by*

$$\dot{m}_{\mathrm{c}} = \hat{\lambda}(t)M_1(t)\left(m_{\mathrm{c}}(t) - m_H(t)\right).$$

*Proof.* Recall that $m_{\mathrm{H}} = \frac{M_2}{2M_1}$, where

$$M_r := \sum_{j=1}^d p_j\mu_j^r, \qquad p_j := \frac{\mu_j^2\beta_j}{\sum_{k=1}^d\mu_k^2\beta_k}. \tag{13}$$

Substituting the definition of $p_j$, we obtain

$$M_2 = \frac{\sum_j\mu_j^4\beta_j}{\sum_k\mu_k^2\beta_k} = \frac{2\sum_j\mu_j^4\beta_j}{n_{\boldsymbol{\theta}}^2}, \qquad M_1 = \frac{\sum_j\mu_j^3\beta_j}{\sum_k\mu_k^2\beta_k} = \frac{2\sum_j\mu_j^3\beta_j}{n_{\boldsymbol{\theta}}^2}.$$

Since $\mu_1 < \cdots < \mu_d$ and $p_j \geq 0$ with $\sum_j p_j = 1$, we have $\frac{\mu_1}{2} \leq m_{\mathrm{H}} = \frac{M_2}{2M_1} \leq \frac{\mu_d}{2}$. We define a new expression for $m_{\mathrm{c}}$.

$$m_{\mathrm{c}}(t) = \frac{\sqrt{S}}{2\rho}, \qquad \text{where } S := n_{\boldsymbol{\theta}}^2. \tag{14}$$

Taking the time derivative of $S$, we have

$$\dot{S} = 2\sum_{j=1}^d\mu_j^2\dot{\beta}_j.$$

From Lemma 4.3, we have $\dot{\beta}_j = r_j\beta_j$ where $r_j = 2\hat{\lambda}\cdot\mu_j\left(1 - \frac{\rho\mu_j}{n_{\boldsymbol{\theta}}}\right) = 2\hat{\lambda}\cdot\left(\mu_j - \frac{\mu_j^2}{2m_{\mathrm{c}}}\right)$. Substituting this into the expression for $\dot{S}$, we get

$$\dot{S} = 2\sum_{j=1}^d\mu_j^2\cdot 2\hat{\lambda}\cdot\left(\mu_j - \frac{\mu_j^2}{2m_{\mathrm{c}}}\right)\cdot\beta_j$$
$$= 4\hat{\lambda}\sum_{j=1}^d\mu_j^2\beta_j\left(\mu_j - \frac{\mu_j^2}{2m_{\mathrm{c}}}\right)$$
$$= 4\hat{\lambda}\sum_{j=1}^d\left(\mu_j^3\beta_j - \frac{\mu_j^4\beta_j}{2m_{\mathrm{c}}}\right).$$

Recalling that $M_1 = \frac{2\sum_{j=1}^{d}\mu_j^3\beta_j}{S}$ and $M_2 = \frac{2\sum_{j=1}^{d}\mu_j^4\beta_j}{S}$, we can rewrite the sums as

$$\sum_{j=1}^{d}\mu_j^3\beta_j = \frac{M_1 S}{2}, \qquad \sum_{j=1}^{d}\mu_j^4\beta_j = \frac{M_2 S}{2}.$$

Therefore, we have

$$\dot{S} = 4\hat{\lambda}\left(\frac{M_1 S}{2} - \frac{M_2 S}{2\cdot 2m_c}\right)$$
$$= 2\hat{\lambda}S\left(M_1 - \frac{M_2}{2m_c}\right).$$

Since $m_c = \frac{\sqrt{S}}{2\rho}$, we have:

$$\dot{m}_c = \frac{1}{2\rho}\cdot\frac{\dot{S}}{2\sqrt{S}} = \frac{\dot{S}}{4\rho\sqrt{S}}.$$

Substituting our expression for $\dot{S}$:

$$\dot{m}_c = \frac{2\hat{\lambda}S\left(M_1 - \frac{M_2}{2m_c}\right)}{4\rho\sqrt{S}}$$
$$= \frac{\hat{\lambda}\sqrt{S}}{2\rho}\left(M_1 - \frac{M_2}{2m_c}\right)$$
$$= \hat{\lambda}m_c\left(M_1 - \frac{M_2}{2m_c}\right)$$
$$= \hat{\lambda}M_1\left(m_c - \frac{M_2}{2M_1}\right)$$
$$= \hat{\lambda}M_1\left(m_c - m_H\right).$$

$\square$

Next, we derive the time derivative of $m_H$.

**Lemma D.8.** *Under Assumption D.3 and Assumption D.4, the time derivative of $m_H$ is given by*

$$\dot{m}_H = \frac{\hat{\lambda}}{2(M_1)^2 m_c}\left(2m_c\Gamma_1 - \Gamma_2\right),$$

*where $\Gamma_1 := M_1 M_3 - M_2^2$ and $\Gamma_2 := M_1 M_4 - M_2 M_3$.*

*Proof.* Starting from $m_H = \frac{M_2}{2M_1}$, we have

$$\dot{m}_H = \frac{\dot{M}_2 M_1 - M_2 \dot{M}_1}{2(M_1)^2}$$
$$= \frac{1}{2M_1}\left(\dot{M}_2 - \frac{M_2}{M_1}\dot{M}_1\right)$$
$$= \frac{1}{2M_1}\left(\sum_{j=1}^{d}\dot{p}_j\mu_j^2 - \frac{M_2}{M_1}\cdot\sum_{j=1}^{d}\dot{p}_j\mu_j\right)$$
$$= \frac{1}{2M_1}\sum_{j=1}^{d}\dot{p}_j\left(\mu_j^2 - 2m_H\mu_j\right).$$

Since $\dot{\beta}_j = r_j\beta_j$ where $r_j = 2\hat{\lambda}\left(\mu_j - \frac{\mu_j^2}{2m_c}\right)$, we can compute

$$\dot{p}_j = \frac{(\mu_j^2\beta_j)\cdot r_j\cdot\left(\sum_{k=1}^{d}\mu_k^2\beta_k\right) - (\mu_j^2\beta_j)\cdot\left(\sum_{k=1}^{d}\mu_k^2\beta_k r_k\right)}{\left(\sum_{k=1}^{d}\mu_k^2\beta_k\right)^2}$$

$$= p_j \left( r_j - \sum_{k=1}^{d} p_k r_k \right)$$

$$= p_j \cdot 2\hat{\lambda} \left( \left( \mu_j - \frac{\mu_j^2}{2m_c} \right) - \sum_{k=1}^{d} p_k \cdot \left( \mu_k - \frac{\mu_k^2}{2m_c} \right) \right)$$

$$= p_j \cdot 2\hat{\lambda} \left( (\mu_j - M_1) - \frac{1}{2m_c} \left( \mu_j^2 - M_2 \right) \right).$$

Substituting this into the expression for $\dot{m}_H$, we have

$$\dot{m}_H = \frac{\hat{\lambda}}{M_1} \sum_{j=1}^{d} p_j \left( (\mu_j - M_1) - \frac{1}{2m_c} (\mu_j^2 - M_2) \right) (\mu_j^2 - 2m_H \mu_j).$$

We split the sum into two components:

$$\text{First term:} \quad C_1 = \sum_j p_j (\mu_j - M_1) \left( \mu_j^2 - 2m_H \mu_j \right),$$

$$\text{Second term:} \quad C_2 = \sum_j p_j \left( \mu_j^2 - M_2 \right) \left( \mu_j^2 - 2m_H \mu_j \right).$$

For the first term,

$$C_1 = \sum_j p_j \mu_j^3 - 2m_H \sum_j p_j \mu_j^2 - M_1 \sum_j p_j \mu_j^2 + 2m_H M_1 \sum_j p_j \mu_j$$

$$= M_3 - 2m_H M_2 - M_1 M_2 + 2m_H M_1^2$$

$$= M_3 - \frac{M_2^2}{M_1} = \frac{M_1 M_3 - M_2^2}{M_1} = \frac{\Gamma_1}{M_1}.$$

For the second term,

$$C_2 = \sum_j p_j \mu_j^4 - 2m_H \sum_j p_j \mu_j^3 - M_2 \sum_j p_j \mu_j^2 + 2m_H M_2 \sum_j p_j \mu_j$$

$$= M_4 - 2m_H M_3 - M_2^2 + 2m_H M_1 M_2$$

$$= M_4 - \frac{M_2 M_3}{M_1} = \frac{M_1 M_4 - M_2 M_3}{M_1} = \frac{\Gamma_2}{M_1}.$$

Therefore, we have

$$\dot{m}_H = \frac{\hat{\lambda}}{M_1} \sum_{j=1}^{d} p_j \cdot \left( (\mu_j - M_1) - \frac{1}{2m_c} \left( \mu_j^2 - M_2 \right) \right) (\mu_j^2 - 2m_H \mu_j)$$

$$= \frac{\hat{\lambda}}{M_1} \left( \frac{\Gamma_1}{M_1} - \frac{\Gamma_2}{2m_c M_1} \right)$$

$$= \frac{\hat{\lambda}}{2 (M_1)^2 m_c} (2m_c \Gamma_1 - \Gamma_2).$$

$\square$

Next, we establish a key inequalities involving the threshold $m_H$.

**Proposition D.9.** $\Gamma_1 \geq 0$ and $\Gamma_2 \geq 0$.

*Proof.* $\Gamma_1$ and $\Gamma_2$ are defined in Lemma D.8. $M_r$ and $p_j$ are defined in Equation 13. Let $M_r := \sum_{j=1}^{d} p_j \mu_j^r = \mathbb{E}_p [\mu_j^r]$. By Cauchy–Schwarz with $X = \boldsymbol{\mu}^{1/2}$ and $Y = \boldsymbol{\mu}^{3/2}$,

$$\left( \mathbb{E}_p \left[ \boldsymbol{\mu}^2 \right] \right)^2 \leq \mathbb{E}_p [\boldsymbol{\mu}] \, \mathbb{E}_p \left[ \boldsymbol{\mu}^3 \right] \implies \Gamma_1 = M_1 M_3 - M_2^2 \geq 0.$$

By Cauchy–Schwarz with $X = \boldsymbol{\mu}$ and $Y = \boldsymbol{\mu}^2$,

$$\left(\mathbb{E}_{\boldsymbol{p}}\left[\boldsymbol{\mu}^3\right]\right)^2 \le \mathbb{E}_{\boldsymbol{p}}\left[\boldsymbol{\mu}^2\right]\mathbb{E}_{\boldsymbol{p}}\left[\boldsymbol{\mu}^4\right].$$

Multiplying the two inequalities gives

$$\mathbb{E}_{\boldsymbol{p}}\left[\boldsymbol{\mu}^2\right]\mathbb{E}_{\boldsymbol{p}}\left[\boldsymbol{\mu}^3\right] \le \mathbb{E}_{\boldsymbol{p}}\left[\boldsymbol{\mu}\right]\mathbb{E}_{\boldsymbol{p}}\left[\boldsymbol{\mu}^4\right] \quad \implies \quad \Gamma_2 = M_1 M_4 - M_2 M_3 \ge 0.$$

$\square$

**Proposition D.10.** *Let* $m_D := \frac{\Gamma_2}{2\Gamma_1}$. *We have* $m_D \ge m_H$ *for all* $t \ge 0$.

*Proof.* We use same notation as in the proof of Proposition D.9. Let $a := \frac{M_2}{M_1}$. $\Gamma_1 \ge 0$ and $\Gamma_2 \ge 0$ by Proposition D.9. Then we have

$$\mathbb{E}_p\left[(\boldsymbol{\mu}^2 - a\boldsymbol{\mu})^2\right] = \mathbb{E}_p[\boldsymbol{\mu}^4] - 2a\,\mathbb{E}_p[\boldsymbol{\mu}^3] + a^2\,\mathbb{E}_p[\boldsymbol{\mu}^2]$$
$$= M_4 - 2aM_3 + a^2 M_2.$$

Substituting $a = \frac{M_2}{M_1}$ and multiplying by $M_1^2$ gives

$$M_1^2 \mathbb{E}_p\left[(\boldsymbol{\mu}^2 - \tfrac{M_2}{M_1}\boldsymbol{\mu})^2\right] = M_1^2 M_4 - 2M_1 M_2 M_3 + M_2^3.$$

Since an expectation of a square is nonnegative and $M_1^2 \ge 0$, it follows that

$$M_1^2 M_4 - 2M_1 M_2 M_3 + M_2^3 \ge 0.$$

Therefore, we have

$$\frac{\Gamma_2}{2\Gamma_1} \ge \frac{M_2}{2M_1} = m_\mathrm{H}.$$

$\square$

### D.4.5 PROOF OF THEOREM 4.4

We begin by restating Theorem 4.4 for convenience.

**Theorem 4.4.** *There exists a unique* $\alpha_1$ *such that* $\alpha_0 := \rho\frac{\mu_1}{\sqrt{2}\|\boldsymbol{\mu}\|_2} < \alpha_1 < \rho\frac{\|\boldsymbol{\mu}\|_4^4}{\sqrt{2}\|\boldsymbol{\mu}\|_2\|\boldsymbol{\mu}\|_3^3} < \alpha_2 :=$
$\rho\frac{\mu_{d-1}+\mu_d}{\sqrt{2}\|\boldsymbol{\mu}\|_2}$ *and the trajectory of* $m_\mathrm{c}(t)$ *falls into one of the following three regimes.*

**(Regime 1)** *If* $\alpha < \alpha_1$, *then* $m_\mathrm{c}(t)$ *strictly decreases for all* $t \ge 0$ *and there exists* $T_1$ *such that for* $j \in [d]$, $\beta_j(t)$ *strictly decreases for all* $t \ge T_1$.

**(Regime 2)** *If* $\alpha_1 < \alpha < \alpha_2$, *there exists* $T_2$ *such that* $m_\mathrm{c}(T_2) < \frac{\mu_{d-1}+\mu_d}{2}$ *and* $m_\mathrm{c}(t)$ *strictly increases for all* $t \ge T_2$.

**(Regime 3)** *If* $\alpha > \alpha_2$, *then* $m_\mathrm{c}(t) > \frac{\mu_{d-1}+\mu_d}{2}$, *and* $\beta_d(t)$ *has the largest growth rate for all* $t \ge 0$.

*Proof.* Recall from Appendix D.4 that

$$p_j := \frac{\mu_j^2 \beta_j}{\sum_{k=1}^d \mu_k^2 \beta_k}, \quad M_r := \sum_{j=1}^d p_j \mu_j^r,$$

$$m_\mathrm{H} := \frac{M_2}{2M_1}, \quad m_\mathrm{L} := \frac{\mu_1}{2}, \quad \hat{\lambda} := |\ell'(\hat{s})| > 0.$$

From Lemma D.8, we further have

$$\Gamma_1 := M_1 M_3 - M_2^2, \quad \Gamma_2 := M_1 M_4 - M_2 M_3, \quad m_\mathrm{D} := \frac{\Gamma_2}{2\Gamma_1}.$$

Also note that

$$m_\mathrm{c} := \frac{\sqrt{S}}{2\rho}, \quad S := n_{\boldsymbol{\theta}}^2.$$

From Lemma D.7 and Lemma D.8, we have

$$\dot{m}_{\text{c}} = \hat{\lambda} M_1 \left( m_{\text{c}} - m_{\text{H}} \right),$$

$$\dot{m}_{\text{H}} = \frac{\hat{\lambda}}{2 \left( M_1 \right)^2 m_{\text{c}}} \left( 2 m_{\text{c}} \Gamma_1 - \Gamma_2 \right).$$

We partition Regime 1 into sub-regimes 1-a ($\alpha < \alpha_0$) and 1-b ($\alpha_0 < \alpha < \alpha_1$), and similarly partition Regime 2 into sub-regimes 2-a $\left( \alpha_1 < \alpha < \rho \frac{\|\boldsymbol{\mu}\|_4^4}{\sqrt{2}\|\boldsymbol{\mu}\|_2^2\|\boldsymbol{\mu}\|_3^3} \right)$ and 2-b $\left( \rho \frac{\|\boldsymbol{\mu}\|_4^4}{\sqrt{2}\|\boldsymbol{\mu}\|_2^2\|\boldsymbol{\mu}\|_3^3} < \alpha < \alpha_2 \right)$. The proof is then structured into three distinct cases: Regime 1-a, Regimes 1-b and 2-a treated jointly, and Regimes 2-b and 3 treated jointly.

**Regime 1-a.** $n_{\boldsymbol{\theta}}(0) = \sqrt{2}\|\boldsymbol{\mu}\|_2 \cdot \alpha$, so $\alpha < \alpha_0$ implies $m_{\text{c}}(0) = \frac{n_{\boldsymbol{\theta}}(0)}{2\rho} < \frac{\mu_1}{2} = m_{\text{L}}$. For any $t \geq 0$, if $m_{\text{c}}(t) < m_{\text{L}}$, then $m_{\text{c}}(t) < \frac{\mu_1}{2} < m_{\text{H}}(t)$. Hence $B(t) < 0$, and therefore $\dot{m}_{\text{c}}(t) < 0$. Consequently, for any $t \geq 0$, whenever $m_{\text{c}}(t) < m_{\text{L}}$, the function $m_{\text{c}}(\cdot)$ is strictly decreasing. Since $m_{\text{c}}(0) < m_{\text{L}}$, we have $m_{\text{c}}(t) < m_{\text{L}}$ for all $t \geq 0$, and it is strictly decreasing.

Moreover, we have $2 m_{\text{c}}(t) < \mu_1 \leq \mu_j$. Therefore,

$$r_j(t) = 2\hat{\lambda}(t) \cdot \left( \mu_j - \frac{\mu_j^2}{2 m_{\text{c}}(t)} \right) < 0,$$

Thus $\dot{\beta}_j(t) = \beta_j(t) r_j(t) < 0$, and $\beta_j(t)$ decays exponentially for all $t \geq 0$.

**Regimes 1-b and 2-a.** We define $A(t) := \hat{\lambda} M_1(t)$ and $B(t) := m_{\text{c}}(t) - m_{\text{H}}(t)$. Then we get the following equalities:

$$\dot{m}_{\text{c}} = AB,$$

$$\dot{B} = \dot{m}_{\text{c}} - \dot{m}_{\text{H}} = AB - \dot{m}_{\text{H}}.$$

Let $I(t) := \exp\left( -\int_0^t A(\tau) d\tau \right)$. Then:

$$I\dot{B} = IAB - I\dot{m}_{\text{H}}, \tag{15}$$

$$\frac{\text{d}}{\text{d}t}(IB) = \dot{I}B + I\dot{B} = -IAB + I\dot{B} = -I\dot{m}_{\text{H}}, \tag{16}$$

$$I(t)B(t) - I(0)B(0) = -\int_0^t I(u)\dot{m}_{\text{H}}(u)\text{d}u. \tag{17}$$

$n_{\boldsymbol{\theta}}(0) = \sqrt{2}\|\boldsymbol{\mu}\|_2 \cdot \alpha$, so $\alpha_0 < \alpha < \rho \frac{\|\boldsymbol{\mu}\|_4^4}{\sqrt{2}\|\boldsymbol{\mu}\|_2^2\|\boldsymbol{\mu}\|_3^3}$ implies $m_{\text{L}} < m_{\text{c}}(0) < m_{\text{H}}(0)$. In this case, we have $B(0) < 0$ and thus $\dot{m}_{\text{c}}(0) = A(0)B(0) < 0$, so $m_{\text{c}}$ initially drifts downward.

For an initial condition $m_0 \in (m_{\text{L}}, m_{\text{H}}(0))$, let $m_{\text{c}}(\cdot; m_0)$ denote the solution with $m_{\text{c}}(0) = m_0$. We define the *floor hitting time* as $\tau_F(m_0) := \inf\{t \geq 0 : m_{\text{c}}(t; m_0) = m_{\text{L}}\} \in [0, \infty]$, and the *ceiling hitting time* as $\tau_G(m_0) := \inf\{t \geq 0 : B(t; m_0) = 0\} \in [0, \infty]$. Define the *first exit time* as $\tau(m_0) := \min\{\tau_F(m_0), \tau_G(m_0)\}$.

Now let

$$\mathcal{F} := \{m_0 \in (m_{\text{L}}, m_{\text{H}}(0)) : \tau_F(m_0) < \tau_G(m_0)\},$$
$$\mathcal{G} := \{m_0 \in (m_{\text{L}}, m_{\text{H}}(0)) : \tau_G(m_0) < \tau_F(m_0)\}.$$

We will prove:

- $\mathcal{F}$ and $\mathcal{G}$ are open in $(m_{\text{L}}, m_{\text{H}}(0))$.

- $\mathcal{F} \neq \emptyset$ and $\mathcal{G} \neq \emptyset$.

- therefore there exists $m_0^* \in (m_{\text{L}}, m_{\text{H}}(0)) \setminus (\mathcal{F} \cup \mathcal{G})$, and for such $m_0^*$, we must have $\tau_F(m_0^*) = \tau_G(m_0^*) = \infty$.

Since $\alpha > 0$, we have $\beta_j(0) > 0$ for all $j \in [d]$, and from the explicit solution of the $\beta_j$ dynamics in Lemma 4.3, we have that for each finite time $t$, $\beta_j(t) > 0$. Therefore, $p_j(t) > 0$ for all finite $t$ and $j \in [d]$, which implies that $p(t)$ is not collapsed onto a single feature at any finite time. Based on this observation, we make the following Lemma:

**Lemma D.11.** $\dot{m}_{\mathrm{c}}(\tau_F(m_0)) < 0$ for all $m_0 \in \mathcal{F}$, and $\dot{B}(\tau_G(m_0)) > 0$ for all $m_0 \in \mathcal{G}$. In other words, the floor crossing and ceiling crossing are transversal.

*Proof.* We first prove for the floor crossing case. $\tau_F(m_0) < \infty$. Let $t_f := \tau_F(m_0)$, so $m_{\mathrm{c}}(t_f; m_0) = m_{\mathrm{L}}$. Then we have

$$\dot{m}_{\mathrm{c}}(t_f; m_0) = A(t_f)B(t_f).$$

Here, $A(t_f) > 0$. Also $p(t_f)$ has positive mass on every $\mu_j$, hence $B(t_f) = m_{\mathrm{L}} - m_{\mathrm{H}}(t_f) < 0$. Thus $\dot{m}_{\mathrm{c}}(t_f; m_0) < 0$, in particular $\dot{m}_{\mathrm{c}}(t_f) \neq 0$. So the intersection with the floor is transversal.

Now prove for the ceiling crossing case. $\tau_G(m_0) < \infty$. Let $t_g := \tau_G(m_0)$, so $B(t_g) = 0$ which means $m_{\mathrm{c}}(t_g) = m_{\mathrm{H}}(t_g)$. Then we have

$$\dot{B}(t_g) = A(t_g)B(t_g) - \dot{m}_{\mathrm{H}}(t_g) = -\dot{m}_{\mathrm{H}}(t_g).$$

From Proposition D.10, we have $m_{\mathrm{H}} \leq m_{\mathrm{D}}$, and the equality holds if and only if $p$ is collapsed onto a single feature. But at any finite time $t_g$, $p(t_g)$ is not collapsed onto a single feature, so $m_{\mathrm{H}}(t_g) < m_{\mathrm{D}}$. Therefore, we get $m_{\mathrm{c}}(t_g) - m_{\mathrm{D}}(t_g) = m_{\mathrm{H}}(t_g) - m_{\mathrm{D}}(t_g) < 0$.

Using the formula

$$\dot{m}_{\mathrm{H}} = \frac{\hat{\lambda}}{2\left(M_1\right)^2 m_{\mathrm{c}}}\left(2m_{\mathrm{c}}\Gamma_1 - \Gamma_2\right) = \frac{\hat{\lambda}}{2\left(M_1\right)^2 m_{\mathrm{c}}}2\Gamma_1\left(m_{\mathrm{c}} - m_{\mathrm{D}}\right),$$

and noting $\Gamma_1 > 0$ for non-collapsed $p$, we conclude

$$\dot{m}_{\mathrm{H}}(t_g) < 0.$$

Hence,

$$\dot{B}(t_g) = -\dot{m}_{\mathrm{H}}(t_g) > 0.$$

So the intersection with the ceiling is transversal. $\qquad\square$

Next, we make a lemma for the openness of $\mathcal{F}$ and $\mathcal{G}$.

**Lemma D.12.** The sets $\mathcal{F}$ and $\mathcal{G}$ are open subsets of $(m_L, m_H(0))$.

*Proof.* We use continuous dependence of ODE solutions on initial data on compact time intervals.

**1. $\mathcal{F}$ is open.** Take any $m_0 \in \mathcal{F}$ and set $t_f := \tau_F(m_0)$. Then $t_f < \infty$ and $m_{\mathrm{c}}(t_f; m_0) = m_{\mathrm{L}}$. By Lemma D.11, the floor crossing is transversal: $\dot{m}_{\mathrm{c}}(t_f; m_0) < 0$.

Consider $F(t, m) := m_{\mathrm{c}}(t; m) - m_{\mathrm{L}}$. We have $F(t_f, m_0) = 0$ and $\partial_t F(t_f, m_0) = \dot{m}_{\mathrm{c}}(t_f; m_0) \neq 0$. By the Implicit Function Theorem, there exists a neighborhood $U_1$ of $m_0$ and a continuous map $m \mapsto \tau_F(m)$ such that for all $m \in U_1$,

$$m_{\mathrm{c}}(\tau_F(m); m) = m_{\mathrm{L}}, \qquad |\tau_F(m) - t_f| < \delta$$

for some $\delta > 0$.

Next we show that the ceiling (gap) cannot be hit before $\tau_F(m)$ for $m$ near $m_0$, without appealing to continuity of $\tau_G$. Since $m_0 \in \mathcal{F}$, we have $\tau_G(m_0) > t_f$, hence

$$B(t; m_0) < 0 \quad \text{for all } t \in [0, t_f].$$

Moreover, at $t_f$ we have $B(t_f; m_0) = m_{\mathrm{L}} - m_{\mathrm{H}}(t_f; m_0) < 0$. By continuity in $t$, there exists $\delta_0 \in (0, \delta]$ such that

$$B(t; m_0) < 0 \quad \text{for all } t \in [0, t_f + \delta_0].$$

Since $[0, t_f + \delta_0]$ is compact and $B(\cdot; m_0)$ is continuous and strictly negative on it, we can define the margin

$$\eta := - \max_{t \in [0, t_f + \delta_0]} B(t; m_0) > 0,$$

so that $B(t; m_0) \le -\eta$ for all $t \in [0, t_f + \delta_0]$.

By continuous dependence on initial data on $[0, t_f + \delta_0]$, there exists a neighborhood $U_2$ of $m_0$ such that for all $m \in U_2$,

$$\sup_{t \in [0, t_f + \delta_0]} |B(t; m) - B(t; m_0)| < \eta/2.$$

Hence for all $m \in U_2$ and all $t \in [0, t_f + \delta_0]$,

$$B(t; m) \le B(t; m_0) + \eta/2 \le -\eta/2 < 0.$$

In particular, $B(t; m)$ cannot reach $0$ before time $t_f + \delta_0$, so

$$\tau_G(m) > t_f + \delta_0.$$

Finally, take $m$ in $U := U_1 \cap U_2$ and shrink $U_1$ if needed so that $|\tau_F(m) - t_f| < \delta_0$. Then $\tau_F(m) < t_f + \delta_0 < \tau_G(m)$, i.e. $m \in \mathcal{F}$. Therefore $\mathcal{F}$ is open.

**2. $\mathcal{G}$ is open.** Take any $m_0 \in \mathcal{G}$ and set $t_g := \tau_G(m_0)$. Then $t_g < \infty$ and $B(t_g; m_0) = 0$. By Lemma D.11, the ceiling crossing is transversal: $\dot{B}(t_g; m_0) > 0$.

Let $G(t, m) := B(t; m)$. We have $G(t_g, m_0) = 0$ and $\partial_t G(t_g, m_0) = \dot{B}(t_g; m_0) \ne 0$. By the Implicit Function Theorem, there exists a neighborhood $V_1$ of $m_0$ and a continuous map $m \mapsto \tau_G(m)$ such that for all $m \in V_1$,

$$B(\tau_G(m); m) = 0, \qquad |\tau_G(m) - t_g| < \delta$$

for some $\delta > 0$.

Next we show that the floor cannot be hit before $\tau_G(m)$ for $m$ near $m_0$. Since $m_0 \in \mathcal{G}$, we have $\tau_F(m_0) > t_g$, hence

$$m_c(t; m_0) > m_L \quad \text{for all } t \in [0, t_g].$$

Because $m_c(\cdot; m_0)$ is continuous and $[0, t_g]$ is compact, define

$$\varepsilon_0 := \min_{t \in [0, t_g]} \big( m_c(t; m_0) - m_L \big) > 0, \qquad \varepsilon := \varepsilon_0/2.$$

Then $m_c(t; m_0) \ge m_L + 2\varepsilon$ for all $t \in [0, t_g]$. By continuity in $t$ at $t = t_g$, there exists $\delta_0 \in (0, \delta]$ such that

$$m_c(t; m_0) \ge m_L + \varepsilon \quad \text{for all } t \in [0, t_g + \delta_0].$$

By continuous dependence on initial data on $[0, t_g + \delta_0]$, there exists a neighborhood $V_2$ of $m_0$ such that for all $m \in V_2$,

$$\sup_{t \in [0, t_g + \delta_0]} |m_c(t; m) - m_c(t; m_0)| < \varepsilon/2.$$

Hence $m_c(t; m) \ge m_L + \varepsilon/2$ on $[0, t_g + \delta_0]$, so in particular $\tau_F(m) > t_g + \delta_0$ for all $m \in V_2$.

Finally, take $m$ in $V := V_1 \cap V_2$ and shrink $V_1$ if needed so that $|\tau_G(m) - t_g| < \delta_0$. Then $\tau_G(m) < t_g + \delta_0 < \tau_F(m)$, i.e. $m \in \mathcal{G}$. Therefore $\mathcal{G}$ is open. $\qquad\square$

Next, we make a lemma for the no simultaneous finite exit.

**Lemma D.13** (No simultaneous finite exit). *There is no $m_0 \in (m_L, m_H(0))$ such that*

$$\tau_F(m_0) = \tau_G(m_0) < \infty.$$

*Proof.* Suppose for contradiction that there exists $m_0 \in (m_L, m_H(0))$ and a finite time $t^\star := \tau_F(m_0) = \tau_G(m_0) < \infty$. Then by definition,

$$m_c(t^\star; m_0) = m_L \quad \text{and} \quad B(t^\star; m_0) = 0.$$

Since $B(t) = m_c(t) - m_H(t)$, the second equality implies $m_H(t^\star; m_0) = m_c(t^\star; m_0) = m_L$.

However, $m_{\mathrm{H}}(t) = m_{\mathrm{L}} = \mu_1/2$ holds if and only if $p(t)$ collapses onto the first feature (i.e., $p_1(t) = 1$). Under the uniform positive initialization, we have $\beta_j(0) > 0$ for all $j$, and by the explicit $\beta$-dynamics (Lemma 4.3) we have $\beta_j(t) > 0$ for every finite $t$. Hence $p_j(t) > 0$ for all $j$ at every finite time, so $p(t)$ cannot be collapsed at time $t^\star < \infty$. This contradicts $m_{\mathrm{H}}(t^\star) = m_{\mathrm{L}}$.

Therefore, no such $m_0$ exists. $\qquad\square$

Next, we make the non-emptiness of $\mathcal{F}$ and $\mathcal{G}$.

**Lemma D.14.** *The set $\mathcal{F}$ is non-empty.*

*Proof.* We show that there exists at least one initial condition $m_0 \in (m_{\mathrm{L}}, m_{\mathrm{H}}(0))$ that hits the floor before the gap closes.

Consider an initial condition $m_0$ arbitrarily close to $m_{\mathrm{L}}$ (with $m_0 > m_{\mathrm{L}}$). At time $t = 0$, we have $B(0) = m_0 - m_{\mathrm{H}}(0)$. Since $m_{\mathrm{H}}(0) > m_{\mathrm{L}}$, the initial gap $B(0)$ is strictly negative and bounded away from 0 as $m_0 \to m_{\mathrm{L}}$. Because $A(0) = \hat{\lambda}(0)M_1(0) > 0$, we have $\dot{m}_{\mathrm{c}}(0) = A(0)B(0) < 0$, meaning $m_{\mathrm{c}}(t)$ strictly decreases initially. Because the velocity $\dot{m}_{\mathrm{c}}(t)$ is continuous and bounded away from zero in a small neighborhood of $t = 0$, the time needed for the trajectory to travel the distance to the floor scales with the initial distance $m_0 - m_{\mathrm{L}}$. Thus, we can make the floor hitting time $\tau_F(m_0)$ arbitrarily small by choosing $m_0$ sufficiently close to $m_{\mathrm{L}}$.

On the other hand, the gap $B(t)$ is a continuous function of time and starts from a strictly negative value $B(0) \approx m_{\mathrm{L}} - m_{\mathrm{H}}(0) < 0$. By continuity, it cannot reach 0 instantly. Therefore, there exists some uniform constant $\varepsilon_0 > 0$ such that for all $m_0$ sufficiently close to $m_{\mathrm{L}}$, we have $B(t) < 0$ for all $t \in [0, \varepsilon_0]$. This implies that the ceiling hitting time satisfies $\tau_G(m_0) \geq \varepsilon_0$ for all such $m_0$.

By choosing $m_0 > m_{\mathrm{L}}$ close enough to $m_{\mathrm{L}}$, we can guarantee that $\tau_F(m_0) < \varepsilon_0 \leq \tau_G(m_0)$. This strict inequality implies $\tau_F(m_0) < \tau_G(m_0)$, meaning $m_0 \in \mathcal{F}$. Hence, $\mathcal{F} \neq \emptyset$. $\qquad\square$

**Lemma D.15.** *The set $\mathcal{G}$ is non-empty.*

*Proof.* We show there exists at least one initial condition that closes the gap $B(t) \to 0$ before reaching the floor. Consider an initial condition $m_0 = m_{\mathrm{H}}(0) - \delta$ with $\delta > 0$ small. Thus, the initial gap is $B(0) = -\delta$, which is strictly negative but arbitrarily close to 0.

First, we establish that $\dot{B}(0)$ is strictly positive and bounded below for small $\delta$. From the corresponding ODE, at $t = 0$:

$$\dot{B}(0) = A(0)B(0) - \dot{m}_{\mathrm{H}}(0) = -A(0)\delta - \dot{m}_{\mathrm{H}}(0).$$

We evaluate the $-\dot{m}_{\mathrm{H}}(0)$ term. Under the uniform positive initialization, $\beta_j(0) > 0$ for all $j$, meaning the initial distribution $p(0)$ has full support across all features and is not collapsed. Because $m_{\mathrm{H}} = m_{\mathrm{D}}$ if and only if $p$ collapses onto a single feature, we have a strict inequality at $t = 0$:

$$m_{\mathrm{D}}(0) > m_{\mathrm{H}}(0).$$

Using the formula for $\dot{m}_{\mathrm{H}}$ evaluated at $t = 0$:

$$\dot{m}_{\mathrm{H}}(0) = \frac{\hat{\lambda}(0)}{2\left(M_1(0)\right)^2 m_0} 2\Gamma_1(0)\left(m_0 - m_{\mathrm{D}}(0)\right).$$

For $m_0 \in (m_{\mathrm{H}}(0) - \delta_0, m_{\mathrm{H}}(0))$, we have $m_0 < m_{\mathrm{H}}(0) < m_{\mathrm{D}}(0)$, which implies $m_0 - m_{\mathrm{D}}(0) < m_{\mathrm{H}}(0) - m_{\mathrm{D}}(0) < 0$. Furthermore, $\hat{\lambda}(0) > 0$, $M_1(0) > 0$, $m_0 > m_{\mathrm{L}} > 0$, and $\Gamma_1(0) > 0$ (by Cauchy-Schwarz on a non-collapsed distribution $p$). Therefore, there exists a constant $c_0 > 0$, independent of sufficiently small $\delta$, such that:

$$-\dot{m}_{\mathrm{H}}(0) \geq c_0 > 0.$$

For $\delta$ chosen small enough, the magnitude of the $-A(0)\delta$ term becomes strictly less than $c_0/2$. Consequently, we obtain $\dot{B}(0) \geq c_0/2 > 0$.

Next, we bound the time required to close the gap versus the time required to hit the floor. Because $\dot{B}(0) \geq c_0/2 > 0$ and the vector field is continuous, there exists a short, finite time interval $[0, T]$

over which $\dot{B}(t) \geq c_0/4$. Starting from $B(0) = -\delta$, the trajectory must hit the ceiling (i.e., $B(t_g) = 0$) at some time $t_g$ bounded by:

$$t_g \leq \frac{\delta}{c_0/4} = \frac{4\delta}{c_0}.$$

We now control how much the center $m_c$ can drop during this interval $[0, t_g]$. Since $\dot{m}_c(t) = A(t)B(t)$, and knowing that $B(t) \in [-\delta, 0]$ on this interval while $A(t)$ is bounded by some constant $A_{\max} > 0$, the total drop is bounded by:

$$m_c(t_g) = m_0 + \int_0^{t_g} \dot{m}_c(t)\mathrm{d}t \geq m_0 - \int_0^{t_g} A_{\max}\delta\mathrm{d}t = m_0 - A_{\max}\delta t_g \geq m_0 - A_{\max}\delta\frac{4\delta}{c_0}.$$

This reveals that $m_c$ decreases by at most $\mathcal{O}(\delta^2)$ in the time it takes the gap to close. Since the initial position $m_0 = m_H(0) - \delta$ is separated from the floor $m_L$ by an $\mathcal{O}(1)$ distance (specifically, $m_H(0) - m_L > 0$), we can choose $\delta$ sufficiently small such that:

$$m_c(t_g) \geq m_0 - \mathcal{O}(\delta^2) > m_L.$$

Thus, the trajectory hits the ceiling at time $t_g$ before it could possibly reach the floor. This implies $\tau_G(m_0) = t_g < \tau_F(m_0)$, meaning $m_0 \in \mathcal{G}$. Hence, $\mathcal{G} \neq \emptyset$. $\qquad\square$

We have now established the following topological properties:

- $\mathcal{F}$ and $\mathcal{G}$ are open subsets of $(m_L, m_H(0))$ (Lemma D.12).

- $\mathcal{F} \neq \emptyset$ (Lemma D.14) and $\mathcal{G} \neq \emptyset$ (Lemma D.15).

- $\mathcal{F} \cap \mathcal{G} = \emptyset$ by their definitions (a trajectory cannot hit the floor first and the ceiling first simultaneously).

- The interval $(m_L, m_H(0))$ is a connected topological space.

A connected space cannot be written as the union of two disjoint, non-empty open sets. Therefore, we must have $(m_L, m_H(0)) \setminus (\mathcal{F} \cup \mathcal{G}) \neq \emptyset$.

Let us pick an initial condition $m_0^* \in (m_L, m_H(0)) \setminus (\mathcal{F} \cup \mathcal{G})$. By the definition of $\mathcal{F}$ and $\mathcal{G}$, this means that the trajectory does not hit the floor first in finite time, nor does it hit the gap (ceiling) first in finite time. It implies that:

$$\tau_F(m_0^*) = \tau_G(m_0^*) = \infty.$$

Thus, the trajectory starting at $m_0^*$ never reaches $m_L$ and never reaches $B = 0$ in finite time. Consequently, it satisfies:

$$m_L < m_c(t; m_0^*) < m_H(t; m_0^*) \quad \forall t \geq 0.$$

This boundary trajectory exactly corresponds to the case where the trajectory reaches neither boundary, completing the proof of existence at the $m_0$ level.

**Existence of the Regime 2 Threshold $\alpha_1$.** Finally, we map the boundary initial condition $m_0^*$ back into the initialization scale parameter $\alpha$. Under the uniform positive initialization, the initial scale determines $n_\theta(0) = \sqrt{2}\|\boldsymbol{\mu}\|_2\alpha$, which in turn dictates the initial center $m_c(0)$ via:

$$m_c(0) = \frac{n_\theta(0)}{2\rho} = \frac{\sqrt{2}\|\boldsymbol{\mu}\|_2}{2\rho}\alpha.$$

Because $m_c(0)$ is a continuous and strictly increasing linear function of $\alpha$, the existence of $m_0^* \in (m_L, m_H(0))$ guarantees the existence of a corresponding initialization scale $\alpha_1$ such that $m_c(0; \alpha_1) = m_0^*$. This $\alpha_1$ is precisely the boundary threshold defining Regime 2.

**Uniqueness of the Regime 2 Threshold $\alpha_1$.** We now show that within the interval $\alpha \in (\alpha_0, \alpha_2)$, there can be *at most one* initialization scale $\alpha_1$ for which the corresponding trajectory is a *boundary trajectory*, i.e. it never reaches either the floor $m_L$ or the ceiling $m_H(t)$ in finite time.

Suppose for contradiction that there exist two distinct boundary initializations $\alpha_a < \alpha_b$ in $(\alpha_0, \alpha_2)$. Let $(m_c^{(a)}(t), p^{(a)}(t))$ and $(m_c^{(b)}(t), p^{(b)}(t))$ denote their trajectories. Since both are boundary trajectories, they satisfy

$$m_L < m_c^{(a)}(t) < m_H^{(a)}(t) \quad \text{and} \quad m_L < m_c^{(b)}(t) < m_H^{(b)}(t) \qquad \forall t \geq 0. \tag{18}$$

In particular, $B^{(\cdot)}(t) := m_c^{(\cdot)}(t) - m_H^{(\cdot)}(t) < 0$ for all $t$. Since $\dot{m}_c = A(t) B(t)$ with $A(t) = \hat{\lambda}(t) M_1(t) > 0$, we have $\dot{m}_c^{(a)}(t) < 0$ and $\dot{m}_c^{(b)}(t) < 0$, so both $m_c^{(a)}(t)$ and $m_c^{(b)}(t)$ are strictly decreasing and bounded below by $m_L$; hence they converge.

Next, along the boundary corridor we have $m_c(t) < m_H(t) \leq m_D(t)$ (using Proposition D.10), and therefore

$$\dot{m}_H(t) = \frac{\hat{\lambda}(t)}{2(M_1(t))^2 m_c(t)} \big( 2m_c(t)\Gamma_1(t) - \Gamma_2(t) \big) = \frac{\hat{\lambda}(t)\Gamma_1(t)}{(M_1(t))^2 m_c(t)} \big( m_c(t) - m_D(t) \big) < 0,$$

so $m_H^{(a)}(t)$ and $m_H^{(b)}(t)$ are also strictly decreasing and bounded below by $m_L$, hence both converge as well.

Since $m_c^{(\cdot)}(t)$ converges while $\dot{m}_c^{(\cdot)}(t) = A^{(\cdot)}(t) B^{(\cdot)}(t) < 0$, we must have $\dot{m}_c^{(\cdot)}(t) \to 0$, hence $B^{(\cdot)}(t) \to 0$ along each boundary trajectory:

$$m_c^{(\cdot)}(t) - m_H^{(\cdot)}(t) \to 0. \tag{19}$$

Likewise, since $m_H^{(\cdot)}(t)$ converges and $\dot{m}_H^{(\cdot)}(t) < 0$, we must have $\dot{m}_H^{(\cdot)}(t) \to 0$. Because $\hat{\lambda}(t) > 0$, $M_1(t) > 0$, $m_c(t) > 0$, and $\Gamma_1(t) \geq 0$, this forces

$$m_c^{(\cdot)}(t) - m_D^{(\cdot)}(t) \to 0. \tag{20}$$

Combining equation 19 and equation 20 yields $m_H^{(\cdot)}(t) - m_D^{(\cdot)}(t) \to 0$. By the equality case of Proposition D.10, this implies that $p^{(\cdot)}(t)$ collapses to a single feature as $t \to \infty$.

We now rule out collapse to any major coordinate $k > 1$. If $m_c(t) \to \mu_k/2$ for some $k > 1$, then the minor feature $j = 1$ has asymptotic growth rate

$$r_1(t) = 2\mu_1 \Big( 1 - \frac{\mu_1}{2m_c(t)} \Big) \to 2\mu_1 \Big( 1 - \frac{\mu_1}{\mu_k} \Big) > 0,$$

so any "collapse-to-major" state is unstable (the minor coordinate gets re-amplified). Therefore, the only possible boundary limit is the *minor-feature collapse* equilibrium

$$E_1: \qquad p(t) \to e_1, \qquad m_c(t) \to m_L = \frac{\mu_1}{2}.$$

In particular, *both* boundary trajectories (for $\alpha_a$ and $\alpha_b$) must converge to $E_1$.

Now use the special structure of the uniform initialization. Under uniform positive initialization, the initial proportions $p(0)$ depend only on $\boldsymbol{\mu}$ (and not on $\alpha$), so the two trajectories share identical $p(0)$ and differ only in their initial $m_c(0)$:

$$m_c(0; \alpha_b) > m_c(0; \alpha_a).$$

We examine the stability of $E_1$ in the $m_c$-direction. At $E_1$ we have $p = e_1$, hence $M_1 = \mu_1$ and $m_H = m_L$. Thus the $m_c$-dynamics

$$\dot{m}_c = \hat{\lambda} M_1 (m_c - m_H)$$

has strictly positive linearization along the $m_c$ axis:

$$\frac{\partial \dot{m}_c}{\partial m_c}\Big|_{E_1} = \hat{\lambda}(E_1) M_1(E_1) = \hat{\lambda}(E_1) \mu_1 > 0.$$

Hence $E_1$ is *repelling* along the $m_c$-direction.

Finally, by uniqueness of solutions to smooth ODEs, trajectories cannot cross in phase space. Since the two trajectories start with the *same* initial proportions $p(0)$ but different initial height $m_c(0)$, and since the limiting equilibrium $E_1$ repels along the $m_c$-direction, it is impossible for *both* trajectories to remain on the boundary corridor equation 18 and still converge to $E_1$.

Therefore, there is at most one boundary initialization scale $\alpha_1$ in $(\alpha_0, \alpha_2)$. Combined with the existence shown above, the threshold $\alpha_1$ is unique.

Since we checked the existence and uniqueness of the boundary threshold $\alpha_1$ in Regime 1-b and 2-a, we can now conclude that for $\alpha < \alpha_1$, the dynamics goes to floor-first behavior(Regime 1) and for $\alpha > \alpha_1$, the dynamics goes to ceiling-first behavior(Regime 2).

**Regimes 2-b and 3.** $n_{\boldsymbol{\theta}}(0) = \sqrt{2}\|\boldsymbol{\mu}\|_2 \cdot \alpha$, so $\alpha > \rho \frac{\|\boldsymbol{\mu}\|_4^4}{\sqrt{2}\|\boldsymbol{\mu}\|_2^2\|\boldsymbol{\mu}\|_3^3}$ implies $m_c(0) = \frac{n_{\boldsymbol{\theta}}(0)}{2\rho} > m_H(0)$. When $m_c(0) > m_H(0)$, we have $B(0) > 0$ and thus $\dot{m}_c(0) = A(0)B(0) > 0$, so $m_c$ initially increases. We now show that $B(t) > 0$ for all $t \geq 0$. Suppose for contradiction that there exists a first time $\tau > 0$ such that $B(\tau) = 0$ (i.e., $m_c(\tau) = m_H(\tau)$). Then

$$
\begin{aligned}
\dot{B}(\tau) &= \dot{m}_c(\tau) - \dot{m}_H(\tau) \\
&= A(\tau)B(\tau) - \dot{m}_H(\tau) \\
&= 0 - \dot{m}_H(\tau) \\
&= -\frac{\hat{\lambda}(\tau)}{2\left(M_1(\tau)\right)^2 m_c(\tau)} \left(2m_c(\tau)\Gamma_1(\tau) - \Gamma_2(\tau)\right).
\end{aligned}
$$

Proposition D.10 gives $m_D(\tau) \geq m_H(\tau)$. Therefore, $2m_c(\tau)\Gamma_1(\tau) - \Gamma_2(\tau) \leq 0$, ensuring that $\dot{B}(\tau) > 0$. Yet, if $\tau$ represents the first instance where $B$ reaches zero from above, we must have $\dot{B}(\tau) \leq 0$. This contradiction establishes that $B(t) > 0$ for all $t \geq 0$, which implies $m_c(t) > m_H(t)$ for all $t \geq 0$. Finally, since $A(t) = \hat{\lambda}M_1(t) > 0$ and $B(t) > 0$ for all $t \geq 0$, it follows that

$$
\dot{m}_c(t) = A(t)B(t) > 0
$$

for all $t \geq 0$, proving that $m_c(t)$ is strictly increasing. Additionally, $\alpha \gtrless \alpha_2$ implies $m_c(0) \gtrless \frac{\mu_{d-1} + \mu_d}{2}$, ensuring that sub-regime 2-b exhibits the dynamics claimed for Regime 2, and that Regime 3 follows its respective theorem statement. $\qquad\square$

### D.5 EXTENSION TO DEEPER DIAGONAL LINEAR NETWORKS

In this section, we extend our analysis to $L$-layer diagonal linear networks. As the depth increases $(L > 2)$, some notational adjustments are necessary.

Recall that the margin is given by

$$
s = \langle \boldsymbol{\beta}, \boldsymbol{\mu} \rangle = \left\langle \boldsymbol{w}^{(1)} \odot \boldsymbol{w}^{(2)} \odot \cdots \odot \boldsymbol{w}^{(L)}, \ \boldsymbol{\mu} \right\rangle,
$$

where $\odot$ denotes elementwise (Hadamard) product.

The gradient of the loss $\mathcal{L}$ with respect to a particular weight $w_j^{(l)}$ can be computed via the chain rule:

$$
\frac{\mathrm{d}\mathcal{L}}{\mathrm{d}w_j^{(l)}} = \frac{\mathrm{d}\mathcal{L}}{\mathrm{d}s} \cdot \frac{\mathrm{d}s}{\mathrm{d}w_j^{(l)}} = -\lambda\mu_j \prod_{k \neq l} w_j^{(k)},
$$

where $\lambda$ is as before, and $k \neq l$ indicates multiplication over all layers except $l$.

The squared Euclidean norm of the gradient vector $\nabla_{\boldsymbol{\theta}}\mathcal{L}$ is then

$$
\|\nabla_{\boldsymbol{\theta}}\mathcal{L}\|^2 = \sum_{j=1}^{d}\sum_{l=1}^{L} \left(\frac{\mathrm{d}\mathcal{L}}{\mathrm{d}w_j^{(l)}}\right)^2 = \lambda^2 \sum_{j=1}^{d}\sum_{l=1}^{L} \mu_j^2 \left(\prod_{k \neq l} w_j^{(k)}\right)^2.
$$

Accordingly, we define

$$n_{\boldsymbol{\theta}} := \sqrt{\sum_{j=1}^{d}\sum_{l=1}^{L}\mu_j^2\left(\prod_{k\neq l}w_j^{(k)}\right)^2}.$$

The resulting perturbation is:

$$\varepsilon_2 := \rho\frac{\nabla_{\boldsymbol{\theta}}\mathcal{L}}{\|\nabla_{\boldsymbol{\theta}}\mathcal{L}\|_2},$$

$$(\varepsilon_2)_{w_j^{(l)}} = -\frac{\rho\mu_j}{n_{\boldsymbol{\theta}}}\prod_{k\neq l}w_j^{(k)}.$$

Thus, the perturbed weights are given by

$$\hat{w}_j^{(l)} := w_j^{(l)} - \frac{\rho\mu_j}{n_{\boldsymbol{\theta}}}\prod_{k\neq l}w_j^{(k)}.$$

The perturbed product then takes the form

$$\hat{\beta}_j := \prod_{l=1}^{L}\hat{w}_j^{(l)}.$$

Therefore, the ODE for each coordinate is:

$$\dot{w}_j^{(l)} = -\frac{\partial\mathcal{L}(\hat{\boldsymbol{\theta}})}{\partial w_j^{(l)}} = \hat{\lambda}\mu_j\prod_{k\neq l}\hat{w}_j^{(k)}.$$

Additionally, we define an assumption on the weight initialization scheme:

**Assumption D.16.** The weights are initialized symmetrically at $t = 0$, that is, $w_j^{(1)}(0) = w_j^{(2)}(0) = \cdots = w_j^{(L)}(0) = w_j(0)$ for all $j$.

Now we show the balancedness-preserving property of the SAM flow.

**Lemma D.17.** *Suppose Assumption D.16 holds. Then for all $t \geq 0$,*

$$w_j^{(l)}(t) = w_j(t) \quad \text{for every } l, j.$$

*Furthermore, the sign of $w_j(t)$ is preserved for all $t \geq 0$.*

*Proof.* Fix $j$. Assume that at some time $t$ all weights corresponding to $j$ across the layers are equal, i.e.,

$$w_j^{(1)}(t) = w_j^{(2)}(t) = \cdots = w_j^{(L)}(t) = w_j(t).$$

Then $n_{\boldsymbol{\theta}}^2(t)$ simplifies as follows:

$$n_{\boldsymbol{\theta}}^2(t) = \sum_{j=1}^{d}\sum_{l=1}^{L}\mu_j^2\left(\prod_{k\neq l}w_j^{(k)}(t)\right)^2$$

$$= \sum_{j=1}^{d}\sum_{l=1}^{L}\mu_j^2\left(w_j(t)^{L-1}\right)^2$$

$$= \sum_{j=1}^{d}L\mu_j^2\left(w_j(t)\right)^{2L-2}.$$

Therefore, the perturbed weight for each layer $l$ simplifies to:

$$\hat{w}_j^{(l)}(t) = w_j^{(l)}(t) - \frac{\rho\mu_j}{n_{\boldsymbol{\theta}}(t)} \prod_{k \neq l} w_j^{(k)}(t)$$

$$= w_j(t) - \frac{\rho\mu_j}{n_{\boldsymbol{\theta}}(t)} w_j(t)^{L-1},$$

which is independent of $l$. Hence,

$$\hat{w}_j^{(1)}(t) = \hat{w}_j^{(2)}(t) = \cdots = \hat{w}_j^{(L)}(t) =: \hat{w}_j(t).$$

Substituting this into the SAM flow equation yields:

$$\dot{w}_j^{(l)}(t) = \hat{\lambda}(t)\mu_j \hat{w}_j(t)^{L-1},$$

which is likewise independent of $l$.

Now, for a fixed $j$, consider the $L$-dimensional vector

$$u_j(t) := \left( w_j^{(1)}(t), w_j^{(2)}(t), \ldots, w_j^{(L)}(t) \right).$$

The SAM dynamics specify the ODE:

$$\dot{u}_j(t) = F_j\left( u_j(t), \boldsymbol{\theta}(t) \right),$$

where $F_j$ is the vector whose $l$-th entry is $\hat{\lambda}(t)\mu_j \prod_{k \neq l} \hat{w}_j^{(k)}(t)$. This ODE is locally Lipschitz in $u_j$, ensuring uniqueness of solutions for given initial conditions.

Consider the one-dimensional diagonal manifold

$$\mathcal{D}_j := \left\{ (x, \ldots, x) \in \mathbb{R}^L : x \in \mathbb{R} \right\}.$$

if $u_j(t) \in \mathcal{D}_j$, then $\dot{u}_j(t) \in \mathcal{D}_j$ as well, because all coordinates have the same derivative. So $\mathcal{D}_j$ is invariant under the flow.

Since the initial condition $u_j(0)$ lies in $\mathcal{D}_j$ due to symmetric initialization, and the ODE solution is unique, we conclude that $u_j(t) \in \mathcal{D}_j$ for all $t \geq 0$. Therefore,

$$w_j^{(l)}(t) = w_j(t) \qquad \text{for all } l, j, \text{ and } t \geq 0.$$

In summary, Assumption D.16 guarantees balancedness at all times for any depth $L$.

Next, we consider the sign preservation property.

Recall that on the balanced manifold, we may write $w_j^{(l)}(t) = w_j(t)$ for all $l$, $j$, and $t \geq 0$, so the per-coordinate dynamics reduce to

$$\dot{w}_j(t) = \hat{\lambda}(t)\mu_j \left( w_j(t) - \rho\frac{\mu_j}{n_{\boldsymbol{\theta}}(t)} w_j(t)^{L-1} \right)^{L-1}.$$

We claim that the sign of $w_j(t)$ is preserved for all $t \geq 0$. To see this, observe that the right-hand side of the ODE is a smooth (in fact, polynomial) function of $w_j$, so it is locally Lipschitz in $w_j$ for each fixed $t$. In particular, if at some time $\tau$ we have $w_j(\tau) = 0$, then $\dot{w}_j(\tau) = 0$, so $w_j(t) \equiv 0$ for all $t \geq \tau$ is a solution with the same initial value. By uniqueness of solutions to ODEs with Lipschitz right-hand side, it follows that once $w_j$ reaches zero, it remains identically zero for all future time and cannot cross to the opposite sign. Therefore, if $w_j(0) \neq 0$, the sign of $w_j(t)$ is preserved for all $t \geq 0$ by continuity; if $w_j(0) = 0$, it remains zero.

In summary, the sign of $w_j(t)$ cannot change during the flow.

$\square$

Utilizing the balancedness-preserving property, we can now extend the lemma for the depth-$L$ diagonal network.

**Lemma D.18.** *Under Assumption D.16 and Assumption D.4, the rescaled $\ell_2$ SAM flow satisfies, for each coordinate $j$,*

$$\frac{\mathrm{d}}{\mathrm{d}t}\beta_j(t) = r_j^{(L)}(t)\beta_j(t),$$

*where*

$$r_j^{(L)}(t) = L\mu_j\beta_j(t)^{(1-2/L)}\left(1 - \frac{\rho\mu_j}{n_{\boldsymbol{\theta}}(t)}\beta_j(t)^{(L-2)/L}\right)^{(L-1)},$$

*and*

$$\beta_j(t) = w_j(t)^L, \qquad n_{\boldsymbol{\theta}}(t) = L\sum_{k=1}^{d}\mu_k^2 w_k(t)^{(2L-2)}.$$

*Proof.* Now define the effective coefficient per coordinate, for general depth $L$:

$$\beta_j(t) := \prod_{l=1}^{L} w_j^{(l)}(t) = w_j(t)^{(L)}.$$

Under the balanced $\ell_2$ SAM flow, the coordinate dynamics become:

$$\dot{\beta}_j(t) = \frac{\mathrm{d}}{\mathrm{d}t}\left(w_j(t)^L\right) = Lw_j(t)^{(L-1)}\dot{w}_j(t)$$
$$= Lw_j^{(L-1)}\hat{\lambda}\mu_j\hat{w}_j^{(L-1)}.$$

We first compute the perturbed weight for coordinate $j$:

$$\hat{w}_j = w_j - \frac{\rho\mu_j}{n_{\boldsymbol{\theta}}}w_j^{L-1} = w_j\left(1 - \frac{\rho\mu_j}{n_{\boldsymbol{\theta}}}w_j^{L-2}\right).$$

Substituting this into the expression for $\dot{\beta}_j(t)$ gives:

$$\dot{\beta}_j(t) = L\hat{\lambda}(t)\mu_j\,w_j^{2L-2}\left(1 - \frac{\rho\mu_j}{n_{\boldsymbol{\theta}}(t)}w_j^{L-2}\right)^{L-1}.$$

To express this in terms of $\beta_j = w_j^L$, note that

$$w_j^{2L-2} = \beta_j^{2-2/L}, \qquad w_j^{L-2} = \beta_j^{(L-2)/L}.$$

Therefore, we obtain:

$$\dot{\beta}_j(t) = L\hat{\lambda}(t)\,\mu_j\,\beta_j(t)^{2-2/L}\left(1 - \frac{\rho\mu_j}{n_{\boldsymbol{\theta}}(t)}\beta_j(t)^{(L-2)/L}\right)^{L-1}.$$

$\square$

Absorbing $\hat{\lambda}(t)$ into the time parameter yields the rescaled SAM flow equation:

$$\frac{\mathrm{d}}{\mathrm{d}t}\beta_j(t) = r_j^{(L)}(t)\,\beta_j(t),$$

where

$$r_j^{(L)}(t) := L\mu_j\,\beta_j(t)^{1-2/L}\left(1 - \frac{\rho\mu_j}{n_{\boldsymbol{\theta}}(t)}\beta_j(t)^{(L-2)/L}\right)^{L-1}.$$

This provides the Depth-$L$ generalization of the SAM feature amplification dynamics.

**Proposition D.19.** *Consider the depth-$L$ diagonal network under Assumption D.16 and Assumption D.4. Define*

$$\beta_j(t) := \prod_{l=1}^{L} w_j^{(l)}(t) = w_j(t)^L, \qquad z_j(t) := \mu_j w_j(t)^{L-2}, \qquad n_{\boldsymbol{\theta}}^2(t) := L\sum_{k=1}^{d} \mu_k^2 w_k(t)^{(2L-2)},$$

*and the critical effective scale:*

$$z_c(t) := \frac{n_{\boldsymbol{\theta}}(t)}{\rho L}.$$

*Then for each time $t$, we have*

$$r_j^{(L)}(t) = Lz_j(t)\left(1 - \frac{\rho}{n_{\boldsymbol{\theta}}(t)}z_j(t)\right)^{L-1} =: \phi_t(z_j(t)).$$

*The function $z \mapsto \phi_t(z)$ is strictly increasing on $(0, z_c(t))$, strictly decreasing on $(z_c(t), n_{\boldsymbol{\theta}}(t)/\rho)$, and possesses a unique interior maximum at $z = z_c(t)$.*

*In particular, at any fixed $t$, the coordinate(s) whose effective scale $z_j(t)$ is closest to the peak of $\phi_t$, i.e., near $z_c(t)$, experience the largest instantaneous relative growth rate.*

*Proof.* In rescaled SAM time, we have

$$r_j^{(L)}(t) = L\mu_j\,\beta_j(t)^{1-2/L}\left(1 - \frac{\rho\mu_j}{n_{\boldsymbol{\theta}}(t)}\beta_j(t)^{(L-2)/L}\right)^{L-1},$$

where

$$n_{\boldsymbol{\theta}}^2(t) = L\sum_{k=1}^{d} \mu_k^2 w_k(t)^{2L-2}.$$

Define the effective $z$-scale by

$$z_j(t) := \mu_j\,w_j(t)^{L-2}.$$

Note that

$$\mu_j\,\beta_j^{(L-2)/L} = \mu_j\,w_j^{L-2} = z_j.$$

Plugging this yields

$$r_j^{(L)}(t) = \phi_t(z_j(t)), \qquad \text{where} \quad \phi_t(z) := Lz\left(1 - \frac{\rho}{n_{\boldsymbol{\theta}}(t)}z\right)^{L-1}.$$

Define the critical effective scale:

$$z_c(t) := \frac{n_{\boldsymbol{\theta}}(t)}{\rho L}.$$

Consider $\phi_t(z) = Lz\,(1-cz)^{L-1}$, where $c = \frac{\rho}{n_{\boldsymbol{\theta}}(t)} > 0$. Its derivative with respect to $z$ is:

$$\frac{\mathrm{d}}{\mathrm{d}z}\phi_t(z) = L\,(1-cz)^{L-2}\,(1-Lcz),$$

so that:

- $\phi_t'(z) > 0$ for $0 < z < z_c(t)$,

- $\phi_t'(z) = 0$ when $z = z_c(t)$,

- $\phi_t'(z) < 0$ for $z_c(t) < z < n_{\boldsymbol{\theta}}(t)/\rho$.

Therefore, for each fixed $t$, the function $z \mapsto \phi_t(z)$ is strictly increasing on $(0, z_c(t))$, strictly decreasing on $(z_c(t), n_{\boldsymbol{\theta}}(t)/\rho)$, and has a unique interior maximum at $z = z_c(t)$.

$\square$

Unlike the depth-2 case, where each $\mu_j$ is a fixed constant and their order remains unchanged throughout training, in the depth-$L$ case the effective quantities $z_j(t)$ are time-dependent and could, in principle, change order as the SAM flow evolves. However, the following proposition establishes that the order of $z_j(t)$ is actually preserved throughout the entire SAM trajectory.

**Proposition D.20.** *Under Assumptions D.16 and D.4, the order of the $z_j(t)$ is preserved in the depth-L SAM flow. That is, if $\mu_1 < \cdots < \mu_d$, then $z_1(t) < z_2(t) < \cdots < z_d(t)$ for all $t \geq 0$.*

*Proof.* We first compute the ODE satisfied by $z_j(t)$. By definition,

$$z_j = \mu_j w_j^{L-2},$$

Taking the time derivative, we get

$$\dot{z}_j = \mu_j (L-2) w_j^{(L-3)} \dot{w}_j$$
$$= \mu_j (L-2) w_j^{(L-3)} \left( \hat{\lambda} \mu_j \hat{w}_j^{(L-1)} \right)$$

Therefore, the perturbed weight is

$$\hat{w}_j = w_j \left( 1 - \frac{\rho \mu_j}{n_{\boldsymbol{\theta}}} w_j^{(L-2)} \right).$$

Also, we get

$$w_j^{(L-3)} \hat{w}_j^{(L-1)} = w_j^{(2L-4)} \left( 1 - \frac{\rho \mu_j}{n_{\boldsymbol{\theta}}} w_j^{(L-2)} \right)^{(L-1)}.$$

Using $w_j^{(L-2)} = \frac{z_j}{\mu_j}$ and $w_j^{(2L-4)} = \frac{z_j^2}{\mu_j^2}$, we obtain

$$\dot{z}_j = (L-2) \hat{\lambda} \mu_j^2 \frac{z_j^2}{\mu_j^2} \left( 1 - \frac{\rho \mu_j}{n_{\boldsymbol{\theta}}} \frac{z_j}{\mu_j} \right)^{(L-1)} = (L-2) \hat{\lambda} z_j^2 \left( 1 - \frac{\rho z_j}{n_{\boldsymbol{\theta}}} \right)^{(L-1)}.$$

Thus, the ODE for $z_j(t)$ can be expressed as

$$\dot{z}_j(t) = f(t, z_j(t)) := (L-2) \hat{\lambda} z_j(t)^2 \left( 1 - \frac{\rho z_j(t)}{n_{\boldsymbol{\theta}}(t)} \right)^{L-1}.$$

Notice that in this expression, the dependence on $j$ appears only through $z_j(t)$; both $\hat{\lambda}$ and $n_{\boldsymbol{\theta}}(t)$ are time-dependent scalars shared across all coordinates. So each $z_j(t)$ solves the same scalar non-autonomous ODE,

$$\dot{z}(t) = f(t, z(t)),$$

with $z(t) = z_j(t)$.

Now at $t = 0$, under symmetric positive init $w_j(0) = \alpha > 0$, we have $z_j(0) = \mu_j \alpha^{L-2}$. Since $\mu_1 < \cdots < \mu_d$ and $\alpha^{L-2} > 0$, we have $z_1(0) < z_2(0) < \cdots < z_d(0)$. For this ODE with $f$ is smooth and locally Lipschitz in $z$, the two different solutions $z_j(t)$ cannot cross each other. If two solutions ever meet (same values at some time), then uniqueness makes them to be identical for all times. So the order of $z_j(t)$ is preserved for all $t \geq 0$. Thus, we have $z_1(t) < z_2(t) < \cdots < z_d(t)$ for all $t \geq 0$.

$\square$

## D.6 PROOFS FOR SECTION 4.2.4

### D.6.1 DERIVATION OF THE DYNAMICS OF $\boldsymbol{\beta}(t)$

The dynamics of $\boldsymbol{\beta}(t) = \boldsymbol{w}(t) \odot \boldsymbol{w}(t)$ is given by

$$\dot{\boldsymbol{\beta}}(t) = \dot{\boldsymbol{w}}(t) \odot \boldsymbol{w}(t) + \boldsymbol{w}(t) \odot \dot{\boldsymbol{w}}(t).$$

By Equation (3), it is given as

$$\dot{\boldsymbol{\beta}}(t) = 2\boldsymbol{\mu} \odot \boldsymbol{w}(t) \odot \left( \boldsymbol{w}(t) - \rho \frac{\boldsymbol{\mu} \odot \boldsymbol{w}(t)}{n_{\boldsymbol{\theta}}(t)} \right)$$

$$= 2\boldsymbol{\mu} \odot \left( \boldsymbol{\beta}(t) - \rho \frac{\boldsymbol{\mu} \odot \boldsymbol{\beta}(t)}{n_{\boldsymbol{\theta}}(t)} \right).$$

Coordinate-wise, we have the linear equation

$$\dot{\beta}_j(t) = 2\mu_j \left( \beta_j(t) - \rho \frac{\mu_j \beta_j(t)}{n_{\boldsymbol{\theta}}(t)} \right) = 2\mu_j \beta_j(t) \left( 1 - \rho \frac{\mu_j}{n_{\boldsymbol{\theta}}(t)} \right).$$

Therefore, separating variables and integrating, we get

$$\frac{\dot{\beta}_j(t)}{\beta_j(t)} = 2\mu_j - 2\rho \frac{\mu_j^2}{n_{\boldsymbol{\theta}}(t)}$$

$$\Rightarrow \int_0^t \frac{\dot{\beta}_j(s)}{\beta_j(s)} ds = \int_0^t \left( 2\mu_j - 2\rho \frac{\mu_j^2}{n_{\boldsymbol{\theta}}(s)} \right) ds$$

$$\Rightarrow \log \frac{\beta_j(t)}{\beta_j(0)} = 2\mu_j t - 2\rho \mu_j^2 \int_0^t \frac{1}{n_{\boldsymbol{\theta}}(s)} ds.$$

Define $I(t) := \int_0^t \frac{1}{n_{\boldsymbol{\theta}}(s)} ds$. Then, the solution is given by

$$\beta_j(t) = \beta_j(0) \exp \left( 2\mu_j t - 2\rho \mu_j^2 I(t) \right) \qquad \text{for } j \in [d].$$

### D.6.2 PROOF OF THEOREM 4.5

Before proving Theorem 4.5, we establish Theorem D.21, which provides lower and upper bounds for $I(t)$ and serves as a key ingredient in the proof of Theorem 4.5 below.

**Theorem D.21.** *Suppose* $\boldsymbol{w}^{(1)} = \boldsymbol{w}^{(2)} = \boldsymbol{\alpha} \in \mathbb{R}^d$. *Let* $(\boldsymbol{w}^{(1)}(t))_{t \geq 0}$ *and* $(\boldsymbol{w}^{(2)}(t))_{t \geq 0}$ *follow the rescaled* $\ell_2$*-SAM flow (2) reduced to (3) with perturbation radius* $\rho$ *and data point* $\boldsymbol{\mu}$. *Define* $\underline{C}_{\boldsymbol{\mu},\boldsymbol{\alpha}} = \frac{\mu_1}{\sqrt{2\sum_{j=1}^d \mu_j^2 \alpha_j^2}}$ *and* $\overline{C}_{\boldsymbol{\mu},\boldsymbol{\alpha}} = \frac{\|\boldsymbol{\mu}\|_2^2}{\sqrt{2d}(\prod_{j=1}^d \mu_j \alpha_j)^{1/d}\|\boldsymbol{\mu}\|_1}$. *Then,*

*(a)* $I(t) \geq \frac{1}{\rho \mu_1^2} \log \left( \frac{1}{\rho \underline{C}_{\boldsymbol{\mu},\boldsymbol{\alpha}} \exp(-\mu_1 t) + 1 - \rho \underline{C}_{\boldsymbol{\mu},\boldsymbol{\alpha}}} \right)$ *when* $\frac{I(t)}{t} \geq \frac{1}{\rho(\mu_1 + \mu_2)}$,

*(b)* $I(t) \leq \frac{d}{\rho \|\boldsymbol{\mu}\|_2^2} \log \left( \frac{1}{\rho \overline{C}_{\boldsymbol{\mu},\boldsymbol{\alpha}} \exp(-\frac{\|\boldsymbol{\mu}\|_1}{d} t) + 1 - \rho \overline{C}_{\boldsymbol{\mu},\boldsymbol{\alpha}}} \right)$.

*Proof.* From the definition of $I(t)$, $I(t) := \int_0^t \frac{1}{n_{\boldsymbol{\theta}}(s)} ds$, we have $I'(t) = \frac{1}{n_{\boldsymbol{\theta}}(t)}$.

Since we suppose $\boldsymbol{w}^{(1)}(0) = \boldsymbol{w}^{(2)}(0)$, and the loss function and dynamics are invariant under exchanging $\boldsymbol{w}^{(1)}$ and $\boldsymbol{w}^{(2)}$, we have $\boldsymbol{w}^{(1)}(t) = \boldsymbol{w}^{(2)}(t) =: \boldsymbol{w}(t)$ for all $t \geq 0$.

From the definition of $n_{\boldsymbol{\theta}}(t)$, we have

$$n_{\boldsymbol{\theta}}(t) = \sqrt{\|\boldsymbol{\mu} \odot \boldsymbol{w}^{(1)}(t)\|_2^2 + \|\boldsymbol{\mu} \odot \boldsymbol{w}^{(2)}(t)\|_2^2}$$

$$= \sqrt{2\|\boldsymbol{\mu} \odot \boldsymbol{w}(t)\|_2^2}$$

$$= \sqrt{2 \left( \sum_{j=1}^d \mu_j^2 w_j(t)^2 \right)}$$

$$= \sqrt{2\left(\sum_{j=1}^{d} \mu_j^2 \beta_j(t)\right)}.$$

From Equation (4), which is $\beta_j(t) = \beta_j(0)\exp\left(2\mu_j t - 2\rho\mu_j^2 I(t)\right)$, we have

$$n_{\boldsymbol{\theta}}(t) = \sqrt{2\left(\sum_{j=1}^{d} \mu_j^2 \beta_j(0)\exp\left(2\mu_j t - 2\rho\mu_j^2 I(t)\right)\right)},$$

and therefore,

$$I'(t) = \frac{1}{\sqrt{2\left(\sum_{j=1}^{d} \mu_j^2 \beta_j(0)\exp\left(2\mu_j t - 2\rho\mu_j^2 I(t)\right)\right)}}.$$

(a) When $\frac{I(t)}{t} \geq \frac{1}{\rho(\mu_1+\mu_2)} \geq \frac{1}{\rho(\mu_1+\mu_j)}$ for $j = 2,\ldots,d$, it holds that

$$(2\mu_j t - 2\rho\mu_j^2 I(t)) - (2\mu_1 t - 2\rho\mu_1^2 I(t)) = 2(\mu_j - \mu_1)(t - \rho(\mu_j + \mu_1)I(t)) \geq 0.$$

Therefore,

$$I'(t) = \frac{1}{\sqrt{2\sum_{j=1}^{d} \mu_j^2 \beta_j(0)\exp\left(2\mu_j t - 2\rho\mu_j^2 I(t)\right)}}$$

$$\leq \frac{1}{\sqrt{2\sum_{j=1}^{d} \mu_j^2 \beta_j(0)\exp\left(2\mu_1 t - 2\rho\mu_1^2 I(t)\right)}}$$

$$= \frac{1}{\sqrt{2\sum_{j=1}^{d} \mu_j^2 \beta_j(0)}\exp\left(\mu_1 t - \rho\mu_1^2 I(t)\right)}$$

Separating variables and integrating, we get

$$\exp(-\rho\mu_1^2 I(t))dI \leq \frac{1}{\sqrt{2\sum_{j=1}^{d} \mu_j^2 \beta_j(0)}}\exp(-\mu_1 t)dt$$

$$\Rightarrow \int_{I(0)}^{I(t)} \exp(-\rho\mu_1^2 u)du \leq \int_0^t \frac{1}{\sqrt{2\sum_{j=1}^{d} \mu_j(s)^2 \beta_j(0)}}\exp(-\mu_1 s)ds$$

$$\Rightarrow -\frac{1}{\rho\mu_1^2}(\exp(-\rho\mu_1^2 I(t)) - \exp(-\rho\mu_1^2 I(0))) \leq -\frac{1}{\sqrt{2\sum_{j=1}^{d} \mu_j(s)^2 \beta_j(0)}}\frac{1}{\mu_1}(\exp(-\mu_1 t) - \exp(-\mu_1 0))$$

$$\underset{(a)}{\Rightarrow} \frac{1}{\rho\mu_1^2}(\exp(-\rho\mu_1^2 I(t)) - 1) \geq \frac{1}{\sqrt{2\sum_{j=1}^{d} \mu_j(s)^2 \beta_j(0)}}\frac{1}{\mu_1}(\exp(-\mu_1 t) - 1)$$

$$\Rightarrow \exp(-\rho\mu_1^2 I(t)) \geq \rho\frac{\mu_1}{\sqrt{2\sum_{j=1}^{d} \mu_j(s)^2 \beta_j(0)}}(\exp(-\mu_1 t) - 1) + 1$$

$$\Rightarrow -\rho\mu_1^2 I(t) \geq \log\left(\rho\underline{C}_{\boldsymbol{\mu},\boldsymbol{\alpha}}(\exp(-\mu_1 t) - 1) + 1\right)$$

$$\Rightarrow I(t) \geq \frac{1}{\rho\mu_1^2}\log\left(\frac{1}{\rho\underline{C}_{\boldsymbol{\mu},\boldsymbol{\alpha}}\exp(-\mu_1 t) + 1 - \rho\underline{C}_{\boldsymbol{\mu},\boldsymbol{\alpha}}}\right),$$

where (a) holds since $I(0) = 0$ from the definition of $I(t)$.

(b) By AM-GM inequality, we have

$$I'(t) = \frac{1}{\sqrt{2\sum_{j=1}^{d} \mu_j^2 \beta_j(0)\exp\left(2\mu_j t - 2\rho\mu_j^2 I(t)\right)}}$$

$$\leq \frac{1}{\sqrt{2d \left( \prod_{j=1}^d \mu_j^2 \beta_j(0) \exp\left( 2\mu_j t - 2\rho\mu_j^2 I(t) \right) \right)^{1/d}}}$$

$$= \frac{1}{\sqrt{2d \left( \prod_{j=1}^d \mu_j^2 \beta_j(0) \right)^{1/d} \exp\left( \frac{2 \sum_{j=1}^d \mu_j}{d} t - \frac{2\rho \sum_{j=1}^d \mu_j^2}{d} I(t) \right)}}$$

$$= \frac{1}{\sqrt{2d \left( \prod_{j=1}^d \mu_j^2 \alpha_j^2 \right)^{1/d} \exp\left( \frac{2\|\boldsymbol{\mu}\|_1}{d} t - \frac{2\rho\|\boldsymbol{\mu}\|_2^2}{d} I(t) \right)}}$$

$$= \frac{1}{\sqrt{2d} \left( \prod_{j=1}^d \mu_j \alpha_j \right)^{1/d} \exp\left( \frac{\|\boldsymbol{\mu}\|_1}{d} t - \frac{\rho\|\boldsymbol{\mu}\|_2^2}{d} I(t) \right)}$$

Separating variables and integrating, we get

$$\exp(-\frac{\rho\|\boldsymbol{\mu}\|_2^2}{d} I(t)) dI \leq \frac{1}{\sqrt{2d} \left( \prod_{j=1}^d \mu_j \alpha_j \right)^{1/d}} \exp\left( -\frac{\|\boldsymbol{\mu}\|_1}{d} t \right) dt$$

$$\Rightarrow \int_{I(0)}^{I(t)} \exp(-\frac{\rho\|\boldsymbol{\mu}\|_2^2}{d} u) du \leq \int_0^t \frac{1}{\sqrt{2d} \left( \prod_{j=1}^d \mu_j \alpha_j \right)^{1/d}} \exp\left( -\frac{\|\boldsymbol{\mu}\|_1}{d} s \right) ds$$

$$\Rightarrow -\frac{d}{\rho\|\boldsymbol{\mu}\|_2^2} (\exp(-\frac{\rho\|\boldsymbol{\mu}\|_2^2}{d} I(t)) - \exp(-\frac{\rho\|\boldsymbol{\mu}\|_2^2}{d} I(0))) \leq -\frac{1}{\sqrt{2d} \left( \prod_{j=1}^d \mu_j \alpha_j \right)^{1/d}} \frac{d}{\|\boldsymbol{\mu}\|_1} (\exp(-\frac{\|\boldsymbol{\mu}\|_1}{d} t) - 1)$$

$$\Rightarrow \exp(-\frac{\rho\|\boldsymbol{\mu}\|_2^2}{d} I(t)) \geq \rho \frac{\|\boldsymbol{\mu}\|_2^2}{\sqrt{2d} \left( \prod_{j=1}^d \mu_j \alpha_j \right)^{1/d} \|\boldsymbol{\mu}\|_1} (\exp(-\frac{\|\boldsymbol{\mu}\|_1}{d} t) - 1) + 1$$

$$\Rightarrow -\rho \frac{\|\boldsymbol{\mu}\|_2^2}{d} I(t) \geq \log\left( \rho \overline{C}_{\boldsymbol{\mu},\boldsymbol{\alpha}} (\exp(-\frac{\|\boldsymbol{\mu}\|_1}{d} t) - 1) + 1 \right)$$

$$\Rightarrow I(t) \leq \frac{d}{\rho\|\boldsymbol{\mu}\|_2^2} \log\left( \frac{1}{\rho \overline{C}_{\boldsymbol{\mu},\boldsymbol{\alpha}} \exp(-\frac{\|\boldsymbol{\mu}\|_1}{d} t) + 1 - \rho \overline{C}_{\boldsymbol{\mu},\boldsymbol{\alpha}}} \right).$$

$\square$

**Theorem 4.5.** *Let $\alpha_0, \alpha_2$ be defined in Theorem 4.4 and $\alpha_1$ be the threshold from there. Suppose $\alpha_1 < \alpha \leq \rho \frac{\mu_1 + \mu_d}{\sqrt{2}\|\boldsymbol{\mu}\|_2} < \alpha_2$. Then, for $j \in [d]$, there exists $T_j$ such that*

$$\frac{\beta_j(T_j)}{\beta_d(T_j)} \geq \mathrm{LB}_j(\alpha) := \exp\left( 2R_j' \left( (R_j - 1) \log\left( \frac{1}{1 - \alpha_0/\alpha} \right) + \log\left( \frac{1}{\alpha_0/\alpha} \right) - C(R_j) \right) \right)$$

*where $R_j := (\mu_j + \mu_d)/\mu_1 > 2$, $R_j' := (\mu_d - \mu_j)/\mu_1$ and $C(R) := R \log R - (R - 1) \log(R - 1)$.*

*Proof.* By the assumption $\alpha_0 < \alpha_1 < \alpha$, we have $\underline{C}_{\boldsymbol{\mu},\boldsymbol{\alpha}} = \frac{\alpha_0}{\rho\alpha} < \frac{1}{\rho}$. We also have

$$\underline{C}_{\boldsymbol{\mu},\boldsymbol{\alpha}} = \frac{\mu_1}{\sqrt{2}\|\mu\|_2 \alpha} \geq \frac{\mu_1}{\sqrt{2}\|\mu\|_2 \rho \alpha_{\boldsymbol{\mu}}^{(2)}} = \frac{\mu_1}{\rho(\mu_1 + \mu_d)} \geq \frac{\mu_1}{\rho(\mu_j + \mu_d)} = \frac{1}{\rho R_j} \quad \text{for all } j \in [d].$$

$$\Rightarrow \frac{1 - \rho\underline{C}_{\boldsymbol{\mu},\boldsymbol{\alpha}}}{\rho\underline{C}_{\boldsymbol{\mu},\boldsymbol{\alpha}}} = \frac{1}{\rho\underline{C}_{\boldsymbol{\mu},\boldsymbol{\alpha}}} - 1 < R_j - 1 \quad \text{for all } j \in [d].$$

Let $T_j := \frac{1}{\mu_1} \log\left( \frac{\rho\underline{C}_{\boldsymbol{\mu},\boldsymbol{\alpha}}}{1 - \rho\underline{C}_{\boldsymbol{\mu},\boldsymbol{\alpha}}} (R_j - 1) \right) \geq 0$.

From Theorem D.21, we have

$$I(T_j) \geq \frac{1}{\rho\mu_1^2} \log\left( \frac{1}{\rho\underline{C}_{\boldsymbol{\mu},\boldsymbol{\alpha}} \exp(-\mu_1 T_j) + 1 - \rho\underline{C}_{\boldsymbol{\mu},\boldsymbol{\alpha}}} \right)$$

$$= \frac{1}{\rho\mu_1^2} \log \left( \frac{1}{\rho\underline{C}_{\boldsymbol{\mu},\boldsymbol{\alpha}} \exp\left(\log\left(\frac{1-\rho\underline{C}_{\boldsymbol{\mu},\boldsymbol{\alpha}}}{\rho\underline{C}_{\boldsymbol{\mu},\boldsymbol{\alpha}}(R_j-1)}\right)\right) + 1 - \rho\underline{C}_{\boldsymbol{\mu},\boldsymbol{\alpha}}} \right)$$

$$= \frac{1}{\rho\mu_1^2} \log \left( \frac{1}{\frac{1-\rho\underline{C}_{\boldsymbol{\mu},\boldsymbol{\alpha}}}{R_j-1} + 1 - \rho\underline{C}_{\boldsymbol{\mu},\boldsymbol{\alpha}}} \right)$$

$$= \frac{1}{\rho\mu_1^2} \log \left( \frac{1}{(1-\rho\underline{C}_{\boldsymbol{\mu},\boldsymbol{\alpha}})\left(1 + \frac{1}{R_j-1}\right)} \right)$$

$$= \frac{1}{\rho\mu_1^2} \log \left( \frac{1}{(1-\rho\underline{C}_{\boldsymbol{\mu},\boldsymbol{\alpha}})\left(\frac{R_j}{R_j-1}\right)} \right)$$

$$= \frac{1}{\rho\mu_1^2} \log \left( \frac{1 - \frac{1}{R_j}}{1 - \rho\underline{C}_{\boldsymbol{\mu},\boldsymbol{\alpha}}} \right).$$

Recall from Equation (4) that

$$\beta_j(T_j) = \beta_j(0) \exp\left(2\mu_j T_j - 2\rho\mu_j^2 I(T_j)\right) \text{ for } j \in [d].$$

Thus, for $j \in [d]$, we have

$$\frac{\beta_j(T_j)}{\beta_d(T_j)} = \exp\left(-2(\mu_d - \mu_j)T_j + 2\rho(\mu_d^2 - \mu_j^2)I(T_j)\right)$$

$$= \exp\left(-2\frac{\mu_d - \mu_j}{\mu_1} \log\left(\frac{\rho\underline{C}_{\boldsymbol{\mu},\boldsymbol{\alpha}}}{1 - \rho\underline{C}_{\boldsymbol{\mu},\boldsymbol{\alpha}}}(R_j - 1)\right) + 2\rho(\mu_d^2 - \mu_j^2)I(T_j)\right)$$

$$\geq \exp\left(-2\frac{\mu_d - \mu_j}{\mu_1} \log\left(\frac{\rho\underline{C}_{\boldsymbol{\mu},\boldsymbol{\alpha}}}{1 - \rho\underline{C}_{\boldsymbol{\mu},\boldsymbol{\alpha}}}(R_j - 1)\right) + 2\frac{\mu_d^2 - \mu_j^2}{\mu_1^2} \log\left(\frac{1 - \frac{1}{R_j}}{1 - \rho\underline{C}_{\boldsymbol{\mu},\boldsymbol{\alpha}}}\right)\right)$$

$$= \exp\left(2\frac{\mu_d - \mu_j}{\mu_1}\left(\frac{\mu_d + \mu_j}{\mu_1} \log\left(\frac{1 - \frac{1}{R_j}}{1 - \rho\underline{C}_{\boldsymbol{\mu},\boldsymbol{\alpha}}}\right) - \log\left(\frac{\rho\underline{C}_{\boldsymbol{\mu},\boldsymbol{\alpha}}}{1 - \rho\underline{C}_{\boldsymbol{\mu},\boldsymbol{\alpha}}}(R_j - 1)\right)\right)\right)$$

$$= \exp\left(2R_j'\left(R_j \log\left(\frac{1 - \frac{1}{R_j}}{1 - \rho\underline{C}_{\boldsymbol{\mu},\boldsymbol{\alpha}}}\right) - \log\left(\frac{\rho\underline{C}_{\boldsymbol{\mu},\boldsymbol{\alpha}}}{1 - \rho\underline{C}_{\boldsymbol{\mu},\boldsymbol{\alpha}}}(R_j - 1)\right)\right)\right)$$

$$= \exp\left(2R_j'\left(R_j \log\left(\frac{\frac{R_j-1}{R_j}}{1 - \frac{\rho\alpha_0}{\alpha}}\right) - \log\left(\frac{\frac{\rho\alpha_0}{\alpha}}{1 - \frac{\rho\alpha_0}{\alpha}}(R_j - 1)\right)\right)\right)$$

$$= \exp\left(2R_j'\left((R_j - 1)\log(R_j - 1) - R_j \log(R_j) - (R_j - 1)\log\left(1 - \frac{\rho\alpha_0}{\alpha}\right) - \log\left(\frac{\rho\alpha_0}{\alpha}\right)\right)\right)$$

$$= \exp\left(2R_j'\left(-C(R_j) - (R_j - 1)\log\left(1 - \frac{\rho\alpha_0}{\alpha}\right) - \log\left(\frac{\rho\alpha_0}{\alpha}\right)\right)\right)$$

$$= \exp\left(2R_j'\left((R_j - 1)\log\left(\frac{1}{1 - \rho\alpha_0/\alpha}\right) + \log\left(\frac{1}{\rho\alpha_0/\alpha}\right) - C(R_j)\right)\right)$$

$\square$

### D.6.3 PROOF OF PROPOSITION 4.6

**Proposition 4.6.** *Under the conditions of Theorem 4.5, define $j^*(\alpha) := \arg\max_{j\in[d]} \mathrm{LB}_j(\alpha)$ and set $\alpha_0^* := \alpha_0$. Then, there exist thresholds $\alpha_0^* < \alpha_1^* < \cdots < \alpha_m^* \leq \rho\frac{\mu_1 + \mu_d}{\sqrt{2}\|\boldsymbol{\mu}\|_2}$ for some $m \leq d - 1$ such that $j^*(\alpha) = j$ for $\alpha \in (\alpha_{j-1}^*, \alpha_j^*]$.*

*Proof.* For $\alpha \in (\alpha_0, \rho \frac{\mu_1 + \mu_d}{\sqrt{2}\|\boldsymbol{\mu}\|_2})$, let $x = \alpha_0/\alpha \in (0,1)$ and write

$$G_j(x) = \log \mathrm{LB}_j(\alpha) = 2R'_j \Phi_{R_j}(x),$$

where

$$\Phi_R(x) = (R-1)\log\frac{1}{1-x} + \log\frac{1}{x} - C(R), \qquad C(R) = R\log R - (R-1)\log(R-1),$$

and $R_j = (\mu_j + \mu_d)/\mu_1 > 1$, $R'_j = (\mu_d - \mu_j)/\mu_1 \geq 0$.

**(1) Shape of $\Phi_{R_j}$.** We have

$$\Phi'_{R_j}(x) = \frac{R_j x - 1}{x(1-x)}, \qquad \Phi''_{R_j}(x) = \frac{R_j - 1}{(1-x)^2} + \frac{1}{x^2} > 0.$$

Thus $\Phi_{R_j}$ is strictly convex on $(0,1)$ and attains its unique minimum at $x = 1/R_j$, where $\Phi_{R_j}(1/R_j) = 0$. Consequently $\Phi_{R_j}(x) \geq 0$ for all $x$ and it is strictly increasing on $[1/R_j, 1)$.

**(2) Crossing between adjacent indices.** For any $j \in \{1, \ldots, d-1\}$ define

$$H_{j+1,j}(x) = G_{j+1}(x) - G_j(x) = 2\big(R'_{j+1}\Phi_{R_{j+1}}(x) - R'_j \Phi_{R_j}(x)\big).$$

Because $R_{j+1} > R_j$, we have $\Phi_{R_{j+1}}(1/R_{j+1}) = 0$ and $\Phi_{R_j}(1/R_{j+1}) > 0$, hence $H_{j+1,j}(1/R_{j+1}) < 0$. Likewise $\Phi_{R_j}(1/R_j) = 0$ and $\Phi_{R_{j+1}}(1/R_j) > 0$, giving $H_{j+1,j}(1/R_j) > 0$. By continuity, $H_{j+1,j}$ has at least one zero $x_j^* \in (1/R_{j+1}, 1/R_j]$.

To show uniqueness, using the expression for $\Phi'_{R_j}$, we obtain

$$H'_{j+1,j}(x) = \frac{2}{x(1-x)}\big((R'_{j+1}R_{j+1} - R'_j R_j)x - (R'_{j+1} - R'_j)\big).$$

Since

$$R'_k R_k = \frac{(\mu_d - \mu_k)(\mu_k + \mu_d)}{\mu_1^2} = \frac{\mu_d^2 - \mu_k^2}{\mu_1^2},$$

we obtain $R'_{j+1}R_{j+1} - R'_j R_j = \frac{\mu_j^2 - \mu_{j+1}^2}{\mu_1^2} < 0$. Its zero occurs at

$$x_c = \frac{R'_{j+1} - R'_j}{R'_{j+1}R_{j+1} - R'_j R_j} = \frac{\mu_1}{\mu_{j+1} + \mu_j},$$

and therefore

$$H'_{j+1,j}(x) > 0 \text{ for } x < x_c, \qquad H'_{j+1,j}(x) < 0 \text{ for } x > x_c.$$

Hence $H_{j+1,j}(x)$ is strictly increasing up to $x_c$ and strictly decreasing afterward. Since $1/R_j = \mu_1/(\mu_j + \mu_d) \leq \mu_1/(\mu_{j+1} + \mu_j)$, $H_{j+1,j}$ is strictly increasing in the interval $(1/R_{j+1}, 1/R_j]$. Because $H_{j+1,j}(1/R_{j+1}) < 0$ and $H_{j+1,j}(1/R_j) > 0$, this implies that $H_{j+1,j}$ crosses zero exactly once in $(1/R_{j+1}, 1/R_j)$. Consequently the root $x_j^*$ is unique, with $H_{j+1,j}(x) < 0$ for $x < x_j^*$ and $H_{j+1,j}(x) > 0$ for $x > x_j^*$.

**(3) Thresholds and staircase structure.** As $\alpha$ increases, $x = \alpha_0/\alpha$ decreases. Define $\alpha_j^* = \alpha_0/x_j^*$. When $\alpha$ crosses $\alpha_j^*$, the maximizer between indices $j$ and $j+1$ switches once from $j$ to $j+1$. Because the intervals $(1/R_{j+1}, 1/R_j]$ are disjoint and ordered, the thresholds satisfy $\alpha_0^* < \alpha_1^* < \cdots < \alpha_m^* \leq \rho(\mu_1 + \mu_d)/(\sqrt{2}\|\boldsymbol{\mu}\|_2)$ for some $m \leq d-1$.

Thus $j^*(\alpha)$ takes constant values on each interval $(\alpha_{j-1}^*, \alpha_j^*]$, increasing step by step until the last threshold within the admissible range. $\qquad\square$

### D.6.4 Proof of Proposition 4.7

**Proposition 4.7.** *Consider $\alpha_0$ defined in Theorem 4.4. (i) If $\alpha < \alpha_0$, then $\boldsymbol{\beta}(t)$ converges to zero. (ii) If $\alpha > \rho\frac{\|\boldsymbol{\mu}\|_2^2}{\sqrt{2d}(\prod_{i=1}^d \mu_i)^{1/d}\|\boldsymbol{\mu}\|_1}$, then $\boldsymbol{\beta}(t)$ converge in $\ell_1$ max-margin direction.*

*Proof.* We use Theorem D.21 to prove the theorem. When $\boldsymbol{w}^{(1)}(0) = \boldsymbol{w}^{(2)}(0) = \alpha\mathbf{1}$, we have

$$\underline{C}_{\boldsymbol{\mu},\boldsymbol{\alpha}} = \frac{\mu_1}{\sqrt{2\sum_{j=1}^d \mu_j^2\alpha^2}} = \frac{\mu_1}{\sqrt{2\sum_{j=1}^d \mu_j^2}\alpha} = \frac{\mu_1}{\sqrt{2}\|\mu\|_2\alpha} = \frac{\alpha_0}{\alpha}$$

$$\overline{C}_{\boldsymbol{\mu},\boldsymbol{\alpha}} = \frac{\|\boldsymbol{\mu}\|_2^2}{\sqrt{2d}(\prod_{j=1}^d \mu_j\alpha)^{1/d}\|\boldsymbol{\mu}\|_1} = \frac{\|\boldsymbol{\mu}\|_2^2}{\sqrt{2d}(\prod_{j=1}^d \mu_j)^{1/d}\alpha\|\boldsymbol{\mu}\|_1}$$

**(i)** By the assumption $\alpha \leq \alpha_0$, we have $\underline{C}_{\boldsymbol{\mu},\boldsymbol{\alpha}} = \frac{\alpha_0}{\rho\alpha} \geq \frac{1}{\rho}$. Let $T := \frac{1}{\mu_1}\log\left(\frac{\rho\underline{C}_{\boldsymbol{\mu},\boldsymbol{\alpha}}}{\rho\underline{C}_{\boldsymbol{\mu},\boldsymbol{\alpha}}-1}\right) \geq 0$.

From Theorem D.21, we have

$$I(t) \geq \frac{1}{\rho\mu_1^2}\log\left(\frac{1}{\rho\underline{C}_{\boldsymbol{\mu},\boldsymbol{\alpha}}\exp(-\mu_1 t) + 1 - \rho\underline{C}_{\boldsymbol{\mu},\boldsymbol{\alpha}}}\right).$$

As $t \to T$, we have

$$\rho\underline{C}_{\boldsymbol{\mu},\boldsymbol{\alpha}}\exp(-\mu_1 t) + 1 - \rho\underline{C}_{\boldsymbol{\mu},\boldsymbol{\alpha}}$$
$$\to \rho\underline{C}_{\boldsymbol{\mu},\boldsymbol{\alpha}}\exp(-\mu_1 T) + 1 - \rho\underline{C}_{\boldsymbol{\mu},\boldsymbol{\alpha}}$$
$$= \rho\underline{C}_{\boldsymbol{\mu},\boldsymbol{\alpha}}\exp(\log\left(\frac{\rho\underline{C}_{\boldsymbol{\mu},\boldsymbol{\alpha}}-1}{\rho\underline{C}_{\boldsymbol{\mu},\boldsymbol{\alpha}}}\right)) + 1 - \rho\underline{C}_{\boldsymbol{\mu},\boldsymbol{\alpha}}$$
$$= \rho\underline{C}_{\boldsymbol{\mu},\boldsymbol{\alpha}}\left(\frac{\rho\underline{C}_{\boldsymbol{\mu},\boldsymbol{\alpha}}-1}{\rho\underline{C}_{\boldsymbol{\mu},\boldsymbol{\alpha}}}\right) + 1 - \rho\underline{C}_{\boldsymbol{\mu},\boldsymbol{\alpha}} = 0.$$

Since $\rho\underline{C}_{\boldsymbol{\mu},\boldsymbol{\alpha}}\exp(-\mu_1 t) + 1 - \rho\underline{C}_{\boldsymbol{\mu},\boldsymbol{\alpha}}$ is strictly decreasing in $t$, we have

$$\rho\underline{C}_{\boldsymbol{\mu},\boldsymbol{\alpha}}\exp(-\mu_1 t) + 1 - \rho\underline{C}_{\boldsymbol{\mu},\boldsymbol{\alpha}} \to 0+ \text{ as } t \to T.$$

Therefore, $I(t) \to +\infty$ as $t \to T$.

Recall from Equation (4) that

$$\beta_j(t) = \beta_j(0)\exp\left(2\mu_j t - 2\rho\mu_j^2 I(t)\right) \text{ for } j \in [d].$$

As $t \to T$, we have $\beta_j(t) \to 0$ for all $j \in [d]$ since $I(t) \to +\infty$. Therefore, $\boldsymbol{\beta}(t) \to \mathbf{0}$ as $t \to T$.

**(ii)** By the assumption $\alpha > \rho\frac{\|\boldsymbol{\mu}\|_2^2}{\sqrt{2d}(\prod_{i=1}^d \mu_i)^{1/d}\|\boldsymbol{\mu}\|_1}$, we have $\overline{C}_{\boldsymbol{\mu},\boldsymbol{\alpha}} < \frac{1}{\rho}$.

From Theorem D.21, we have

$$I(t) \leq \frac{d}{\rho\|\boldsymbol{\mu}\|_2^2}\log\left(\frac{1}{\rho\overline{C}_{\boldsymbol{\mu},\boldsymbol{\alpha}}\exp(-\frac{\|\boldsymbol{\mu}\|_1}{d}t) + 1 - \rho\overline{C}_{\boldsymbol{\mu},\boldsymbol{\alpha}}}\right).$$

For $t \in [0, \infty)$, we have

$$0 < 1 - \rho\overline{C}_{\boldsymbol{\mu},\boldsymbol{\alpha}} \leq \rho\overline{C}_{\boldsymbol{\mu},\boldsymbol{\alpha}}\exp(-\frac{\|\boldsymbol{\mu}\|_1}{d}t) + 1 - \rho\overline{C}_{\boldsymbol{\mu},\boldsymbol{\alpha}} < 1.$$

and as $t \to \infty$, we have

$$\rho\overline{C}_{\boldsymbol{\mu},\boldsymbol{\alpha}}\exp(-\frac{\|\boldsymbol{\mu}\|_1}{d}t) + 1 - \rho\overline{C}_{\boldsymbol{\mu},\boldsymbol{\alpha}} \to 1 - \rho\overline{C}_{\boldsymbol{\mu},\boldsymbol{\alpha}} > 0.$$

As $t \to \infty$, we have

$$I(t) \leq \frac{d}{\rho\|\boldsymbol{\mu}\|_2^2}\log\left(\frac{1}{\rho\overline{C}_{\boldsymbol{\mu},\boldsymbol{\alpha}}\exp(-\frac{\|\boldsymbol{\mu}\|_1}{d}t) + 1 - \rho\overline{C}_{\boldsymbol{\mu},\boldsymbol{\alpha}}}\right) \to \frac{d}{\rho\|\boldsymbol{\mu}\|_2^2}\log\left(\frac{1}{1 - \rho\overline{C}_{\boldsymbol{\mu},\boldsymbol{\alpha}}}\right) < \infty.$$

Therefore, $I(t) < \infty$ as $t \to \infty$.

Recall from Equation (4) that

$$\beta_j(t) = \beta_j(0) \exp\left(2\mu_j t - 2\rho\mu_j^2 I(t)\right) \text{ for } j \in [d].$$

Thus, for $j \in [d]$, we have

$$\frac{\beta_j(t)}{\beta_d(t)} = \exp\left(-2(\mu_d - \mu_j)t + 2\rho(\mu_d^2 - \mu_j^2)I(t)\right).$$

As $t \to \infty$, we have $\frac{\beta_j(t)}{\beta_d(t)} \to 0$ for all $j < d$ since $\lim_{t\to\infty} I(t) < \infty$. Therefore, $\boldsymbol{\beta}(t)$ converges to the direction of $\boldsymbol{e}_d$ as $t \to \infty$.

$\square$

## D.7 NUMERICAL EVALUATION OF THEOREM 4.5

In this section, we provide numerical illustrations of the lower bound $\mathrm{LB}_j(\alpha)$ derived in Theorem 4.5. For several choices of $\mu$, we compute the value of

$$\mathrm{LB}_j(\alpha) := \exp\left(2R_j'\left((R_j - 1)\log\left(\tfrac{1}{1 - \alpha_0/\alpha}\right) + \log\left(\tfrac{1}{\alpha_0/\alpha}\right) - C(R_j)\right)\right)$$

and visualize how much the ratio $\beta_j(t)/\beta_d(t)$ must be amplified at minimum.

Figure 14 shows that for small $\alpha$ in Regime 2 and for $\mu$ with a large spectral gap $\mu_d/\mu_1$, $\mathrm{LB}_j(\alpha)$ easily exceeds 10. Since this is only a lower bound, the actual amplification can be even larger, indicating that minor-to-intermediate coordinates can grow by substantially more than the major coordinate.

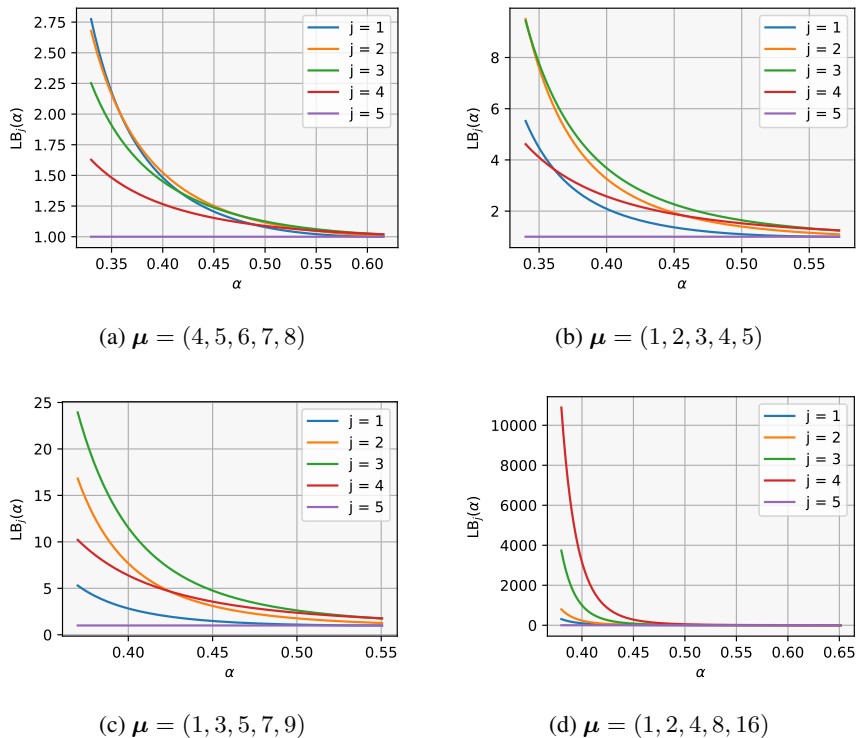

(a) $\mu = (4, 5, 6, 7, 8)$

(b) $\mu = (1, 2, 3, 4, 5)$

(c) $\mu = (1, 3, 5, 7, 9)$

(d) $\mu = (1, 2, 4, 8, 16)$

Figure 14: Numerical evaluation of $\mathrm{LB}_j(\alpha)$ for various choices of $\mu$.

For reproducibility, we describe the numerical procedure used to generate Figure 14. For each choice of $\mu$ (with $d = \dim(\mu)$), we evaluate $\mathrm{LB}_j(\alpha)$ for all $j \in [d]$ on a uniform grid of $\alpha$ values. Following the assumptions of Theorem 4.5, we first obtain the threshold $\alpha_1$ specified in Theorem 4.4. We then set $\alpha \in \left[\alpha_1, \rho \frac{\mu_1 + \mu_d}{\sqrt{2}\|\mu\|_2}\right]$ using 400 grid points. The quantities $\alpha_0$, $R_j$, $R_j'$, and $C(R_j)$ are computed directly from their definitions in Theorems 4.4 and 4.5 using the given $\mu$. The index $j \in [d]$ corresponds to the coordinate ordering $\mu_1 < \cdots < \mu_d$. Since the computation is closed-form, no randomness is involved and the plots are exactly reproducible.

## D.8 EMPIRICAL VERIFICATION

Our analysis in Section 4.2 focuses on the one-point setting $\mathcal{D}_\mu$. We begin by verifying that the sequential feature amplification occurs across multiple choices of $\mu$ in this one-point regime: both the continuous-time rescaled flows and the discrete $\ell_\infty$-SAM updates exhibit the same coordinate-wise progression, and the loss dynamics follow the theoretical prediction. We then turn to multi-point datasets and show that the sequential feature amplification persists in this more realistic setting under both the rescaled $\ell_2$-SAM flow and discrete $\ell_2$-SAM updates, as illustrated in Figure 11. Finally,

we confirm that this phenomenon is not limited to depth 2; the same coordinate-wise progression arises in deeper diagonal networks (general depth $L$). Taken together, these results demonstrate that the sequential feature amplification is a robust and widely recurring behavior: it appears consistently across different $\boldsymbol{\mu}$, across multiple multi-point datasets, across both continuous and discrete SAM dynamics, and across depths $L \geq 2$.

To clarify the heatmap visualizations (e.g., Figures 3a and 15 to 23), for each time $t$ and initialization scale $\alpha$, we compute $j^{\dagger} = \arg\min_j \beta_j(t)$ and color the grid point $(t, \alpha)$ according to this index. Grid regions where the predictor $\boldsymbol{\beta}$ becomes negligibly small are shown in gray, indicating convergence toward $\mathbf{0}$. We use the threshold $\|\boldsymbol{\beta}(t)\|_2 \leq 10^{-2}$ to define gray regions.

Following the visualization style of Figure 3a, we also partition the $\alpha$–axis into the three regimes defined in Theorem 4.4: Regime 1 (small $\alpha$), Regime 2 (intermediate $\alpha$), and Regime 3 (large $\alpha$). These regime boundaries are indicated by horizontal black dashed lines in heatmap figures.

For reproducibility, we detail the exact initialization used in all experiments. As mentioned in Section 4.2, we adopt a uniform initialization across coordinates and layers: $\boldsymbol{w}^{(1)}(0) = \boldsymbol{w}^{(2)}(0) = \alpha\mathbf{1}$ for depth-2 setup and $\boldsymbol{w}^{(1)}(0) = \cdots = \boldsymbol{w}^{(L)}(0) = \alpha\mathbf{1}$ for depth-$L$. To approximate continuous-time trajectories, we simulate the flow using an explicit Euler scheme with a small step size $\eta = 10^{-4}$. For discrete updates, we use a step size of $\eta = 0.01$.

### D.8.1 ONE-POINT CASE: CONTINUOUS VS. DISCRETE DYNAMICS

We first verify that sequential feature amplification appears robustly across multiple choices of $\boldsymbol{\mu}$ in the one-point setting. To demonstrate that this phenomenon is not limited to the continuous $\ell_2$-SAM flow, we additionally evaluate discrete $\ell_2$-SAM updates. Across all tested choices of $\boldsymbol{\mu}$, the resulting heatmaps closely match the structure in Figure 3a, showing both time–wise and initialization–wise sequential feature amplification. To better visualize the evolution of $\boldsymbol{\beta}(t)$, we also provide the loss heatmaps over $(\alpha, t)$. In the discrete $\ell_2$-SAM case, Regime 1 often appears unstable and does not become fully gray. This occurs because the relatively large step size causes the trajectory to hover near the origin without collapsing exactly to $\mathbf{0}$. As a result, the predictor norm stays above the gray threshold—so it is not colored gray—yet the loss remains large, revealing that the trajectory is still effectively stuck in the vicinity of the origin.

For comparison, we first present the results of GF and discrete GD with $\boldsymbol{\mu} = (4, 5, 6, 7, 8)$. The behavior is similar across different choices of $\boldsymbol{\mu}$. Both GF and GD consistently recover the major feature, independent of the initialization scale $\alpha$, and they do not exhibit sequential feature amplification.

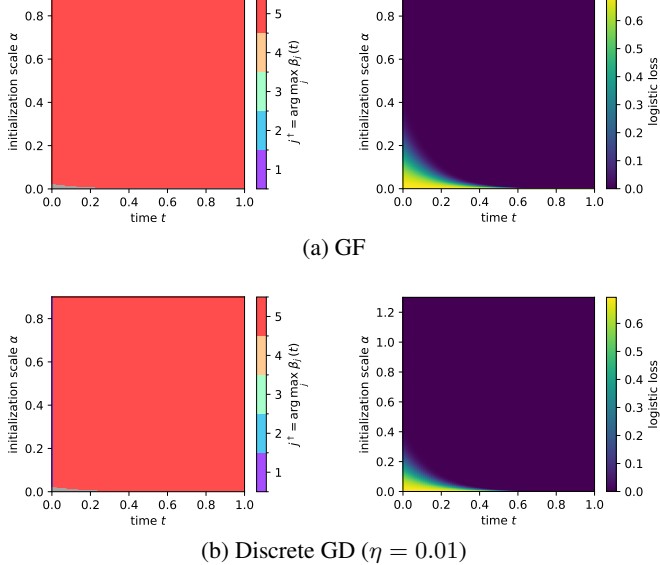

Figure 15: Dominant index $j^{\dagger}$ over $\alpha, t$ and logistic loss on $\mathcal{D}_{\boldsymbol{\mu}}$ with $\boldsymbol{\mu} = (4, 5, 6, 7, 8)$.

1. $\boldsymbol{\mu} = (4, 5, 6, 7, 8)$

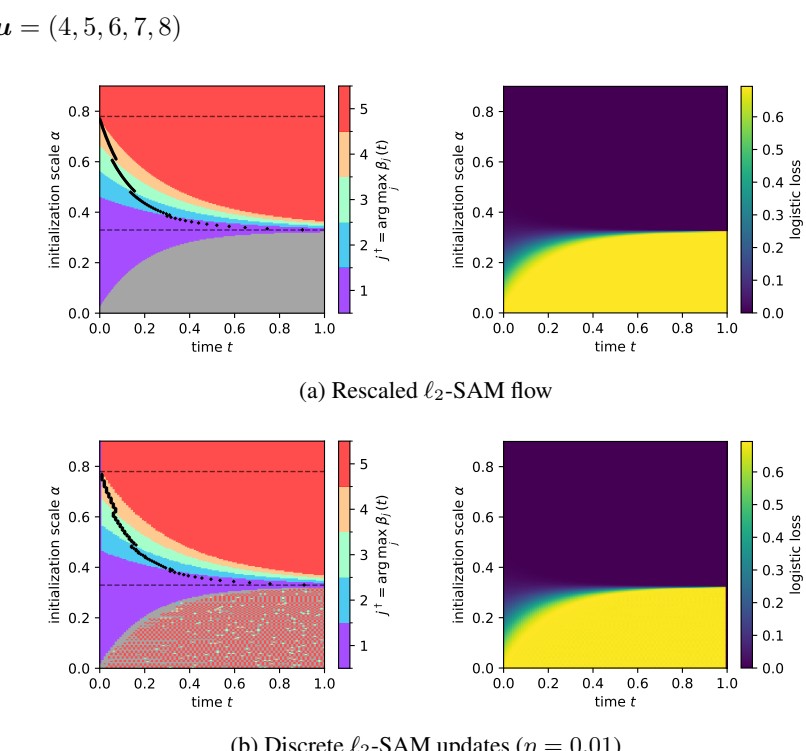

(a) Rescaled $\ell_2$-SAM flow

(b) Discrete $\ell_2$-SAM updates ($\eta = 0.01$)

Figure 16: Dominant index $j^\dagger$ over $\alpha, t$ and logistic loss on $\mathcal{D}_{\boldsymbol{\mu}}$ with $\boldsymbol{\mu} = (4, 5, 6, 7, 8)$ and $\rho = 1$.

2. $\boldsymbol{\mu} = (1, 2, 3, 4, 5)$

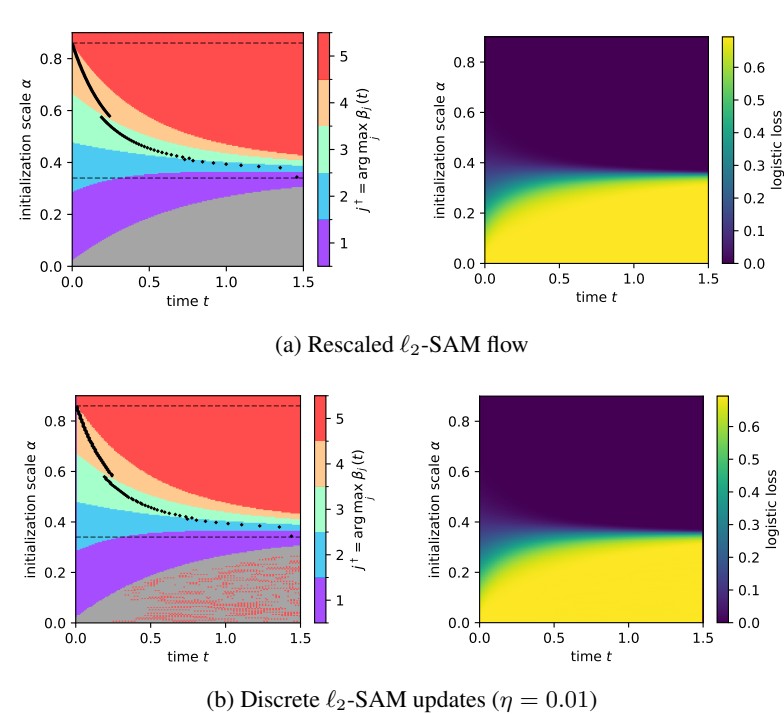

(a) Rescaled $\ell_2$-SAM flow

(b) Discrete $\ell_2$-SAM updates ($\eta = 0.01$)

Figure 17: Dominant index $j^\dagger$ over $\alpha, t$ and logistic loss on $\mathcal{D}_{\boldsymbol{\mu}}$ with $\boldsymbol{\mu} = (1, 2, 3, 4, 5)$ and $\rho = 1$.

3. $\boldsymbol{\mu} = (1, 3, 5, 7, 9)$

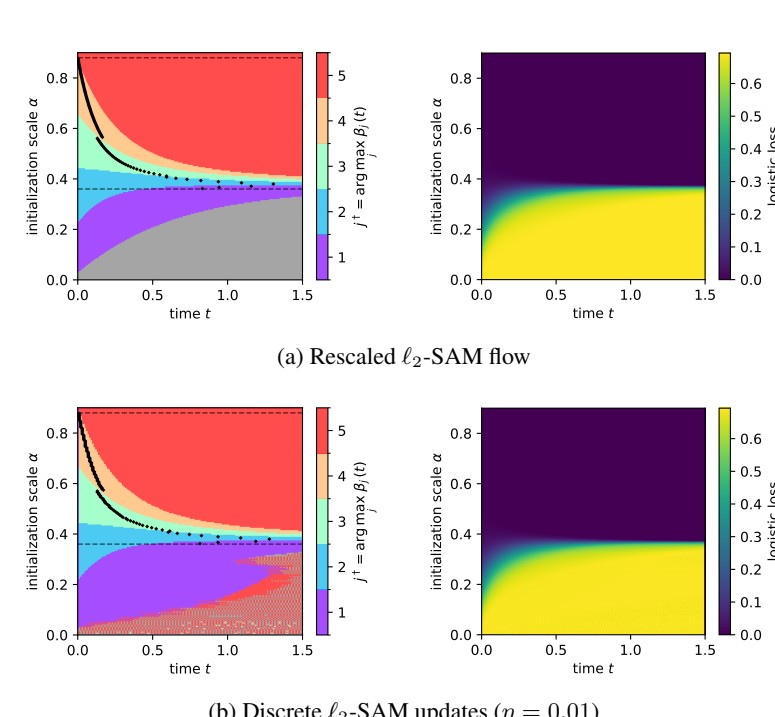

(a) Rescaled $\ell_2$-SAM flow

(b) Discrete $\ell_2$-SAM updates ($\eta = 0.01$)

Figure 18: Dominant index $j^\dagger$ over $\alpha, t$ and logistic loss on $\mathcal{D}_{\boldsymbol{\mu}}$ with $\boldsymbol{\mu} = (1, 3, 5, 7, 9)$ and $\rho = 1$.

4. $\boldsymbol{\mu} = (1, 2, 4, 8, 16)$

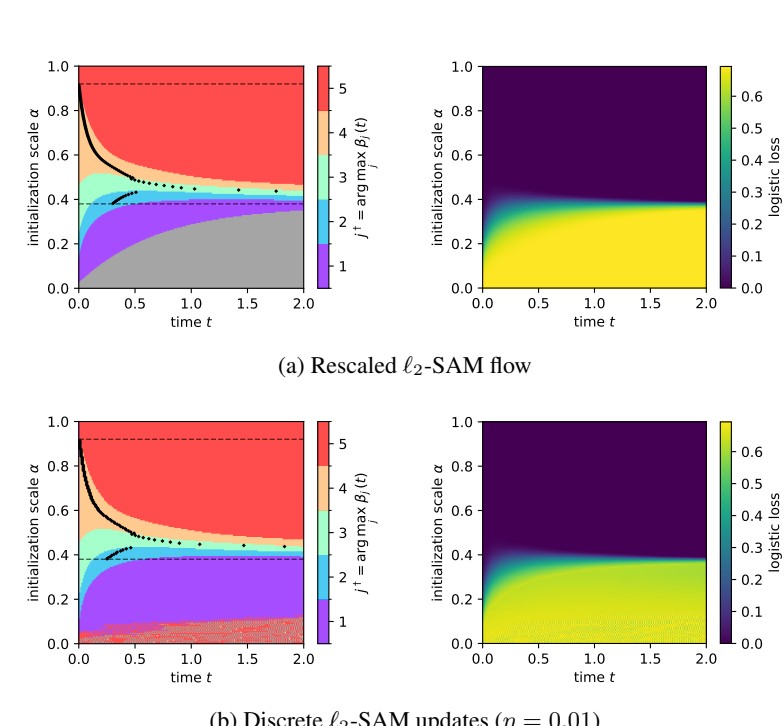

(a) Rescaled $\ell_2$-SAM flow

(b) Discrete $\ell_2$-SAM updates ($\eta = 0.01$)

Figure 19: Dominant index $j^\dagger$ over $\alpha, t$ and logistic loss on $\mathcal{D}_{\boldsymbol{\mu}}$ with $\boldsymbol{\mu} = (1, 2, 4, 8, 16)$ and $\rho = 1$.

### D.8.2 MULTI-POINT CASE: PERSISTENCE OF ONE-POINT BEHAVIOR

To examine whether the sequential feature amplification identified in the one-point analysis persist in more realistic datasets, we construct random linearly separable binary data by sampling two Gaussian clusters centered at $+\mu$ and $-\mu$ for various choices of $\mu$. Specifically, we draw

$$\boldsymbol{x}_n^{(+)} = \boldsymbol{\mu} + \boldsymbol{\varepsilon}_n, \quad y_n = +1, \qquad \boldsymbol{x}_n^{(-)} = -\boldsymbol{\mu} + \boldsymbol{\varepsilon}_n, \quad y_n = -1,$$

with $\boldsymbol{\varepsilon}_n \sim \mathcal{N}(0, \sigma^2 \boldsymbol{I}_d)$ and use $N/2$ samples per class (with $\boldsymbol{\mu} = (1,2), N = 100, \sigma = 0.5$). For visualization, we plot only the first two dimensions of the dataset in the left panels. The middle panels show the results of the rescaled $\ell_2$-SAM flow on this dataset, and the right panels show the discrete $\ell_2$-SAM updates. Across all choices of multi-point datasets, the same sequential feature amplification behavior observed in the one-point setting persists.

For comparison, we present the results of GF and discrete GD with the multi-point dataset generated with mean $\boldsymbol{\mu} = (4, 5, 6, 7, 8)$. The behavior is similar across different choices of $\boldsymbol{\mu}$. As in the one-point setting, both GF and GD consistently recover the major feature, independent of the initialization scale $\alpha$, and they do not exhibit sequential feature amplification.

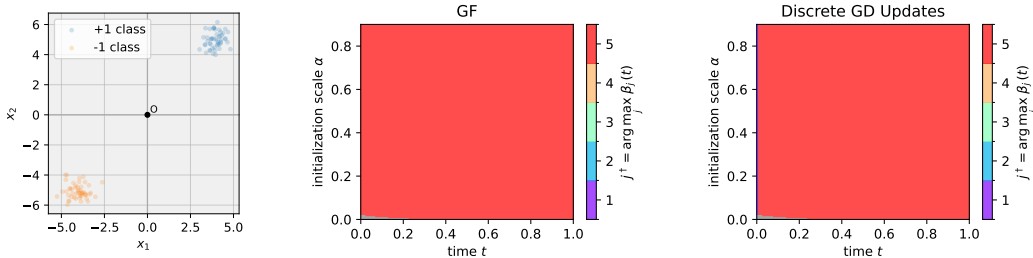

Figure 20: First two dimensions of $\mathcal{D}_{\boldsymbol{\mu}}$ with $\boldsymbol{\mu} = (4, 5, 6, 7, 8)$ and the dominant index $j^\dagger$ over $\alpha, t$ under GF and discrete GD updates.

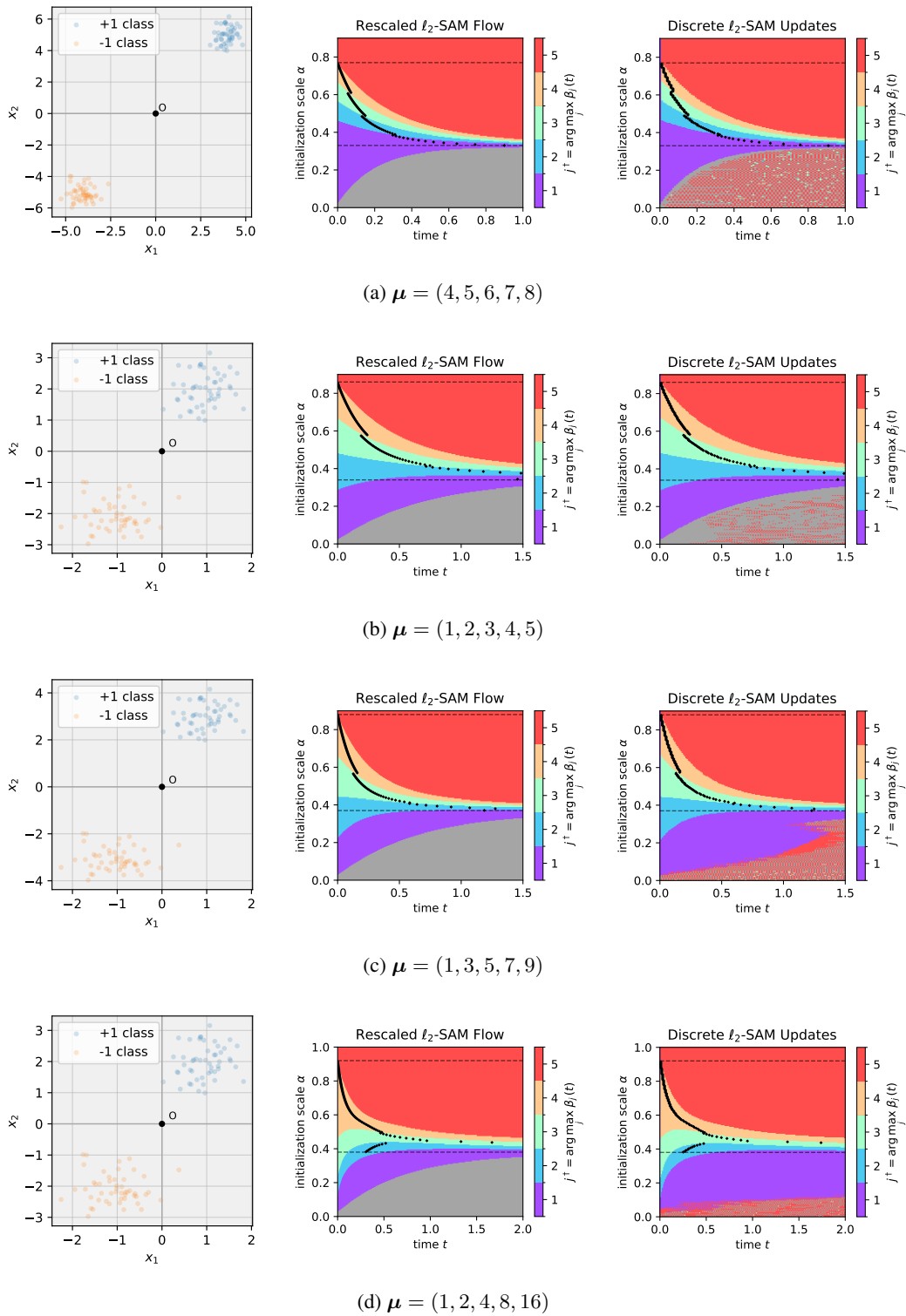

(a) $\boldsymbol{\mu} = (4, 5, 6, 7, 8)$

(b) $\boldsymbol{\mu} = (1, 2, 3, 4, 5)$

(c) $\boldsymbol{\mu} = (1, 3, 5, 7, 9)$

(d) $\boldsymbol{\mu} = (1, 2, 4, 8, 16)$

Figure 21: First two dimensions of $\mathcal{D}_{\boldsymbol{\mu}}$ and the dominant index $j^{\dagger}$ over $\alpha, t$ under the rescaled $\ell_2$-SAM flow and discrete $\ell_2$-SAM updates.

### D.8.3 DEPTH-$L$ CASE: PERSISTENCE OF DEPTH-2 DYNAMICS

We confirm that the sequential feature amplification is not limited to depth $L = 2$; the same coordinate-wise progression arises in deeper diagonal networks (general depth $L$). Specifically, we observe GF and rescaled $\ell_2$-SAM flow on the one-point dataset $\mathcal{D}_{\boldsymbol{\mu}}$ with $\boldsymbol{\mu} = (4, 5, 6, 7, 8)$. The behavior remains similar across different choices of $\boldsymbol{\mu}$, multi-point datasets, and under discrete updates. While GF appears to exhibit Regime 1 (being trapped near the origin), it does not show the sequential feature amplification, even in the deeper models. However, the rescaled $\ell_2$-SAM flow clearly demonstrates the sequential feature amplification for general depth $L$. Even though Regime 1 appears chaotic, Regime 2 and 3 are distintcly observed. Thus, the sequential feature amplification robustly occurs not only at depth $L = 2$ but also in deeper models.

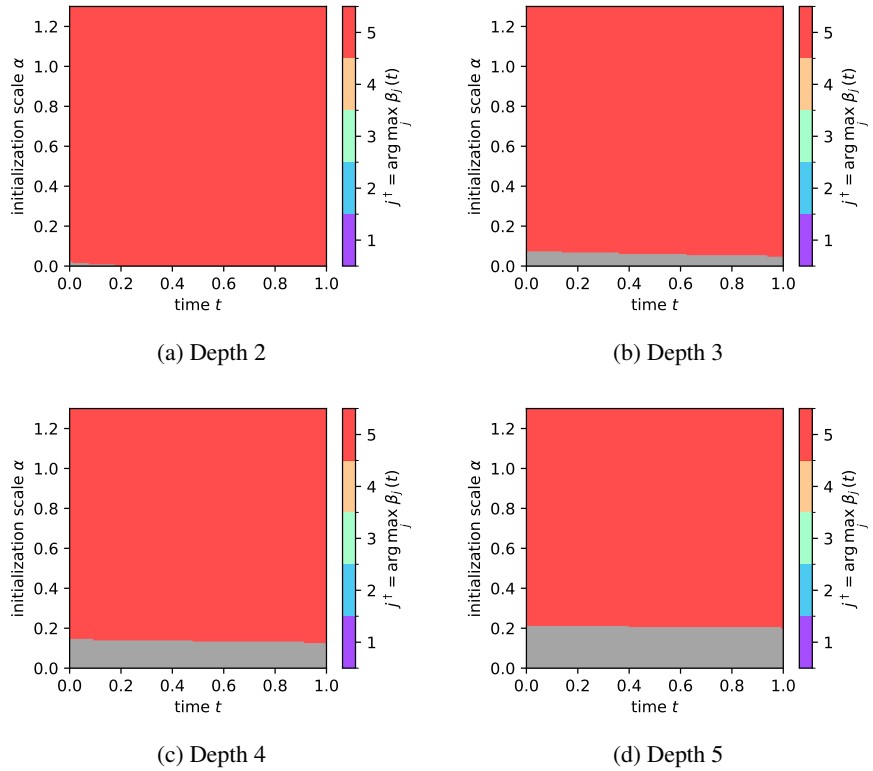

(a) Depth 2                                (b) Depth 3

(c) Depth 4                                (d) Depth 5

Figure 22: Dominant index $j^{\dagger}$ over $\alpha, t$ under the GF on $\mathcal{D}_{\boldsymbol{\mu}}$ with $\boldsymbol{\mu} = (4, 5, 6, 7, 8)$.

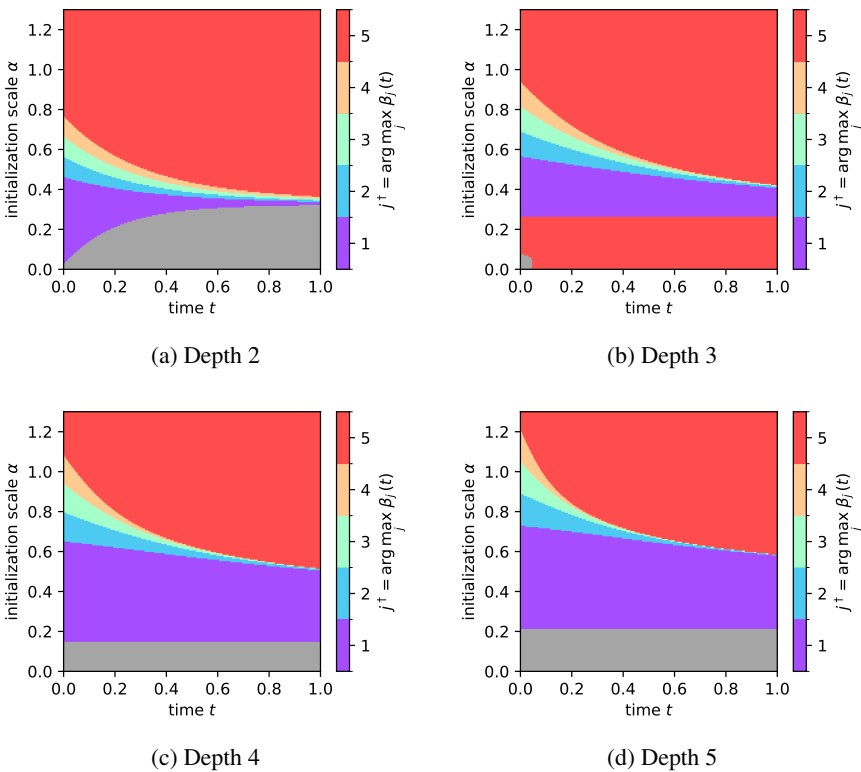

(a) Depth 2

(b) Depth 3

(c) Depth 4

(d) Depth 5

Figure 23: Dominant index $j^\dagger$ over $\alpha, t$ under the rescaled $\ell_2$-SAM flow on $\mathcal{D}_{\boldsymbol{\mu}}$ with $\boldsymbol{\mu} = (4, 5, 6, 7, 8)$ and $\rho = 1$.

# E  EXPERIMENTS

## E.1  LOSS DYNAMICS

For initialization scales in the intermediate regime (Regime 2 in Theorem 4.4), SAM first amplifies minor coordinates and only later focuses on the major ones. This also affects to the training loss curve. As shown in Figure 24, the loss curve of SAM is noticeably flatter than that of GD in the early phase of training. In this experiment, we train the diagonal linear network with full-batch SAM using radius $\rho = 0.5$, learning rate 0.05, and 10000 epochs. We fix the initialization scale to $\alpha = 0.06$ as a representative intermediate value. The data vector is $\mu = (1, 2, 3, 4, 5, 6)$, and all other settings follow the default diagonal-network configuration.

To make this precise, we track the dominant index $\arg\max_j r_j(t)$, where $r_j(t)$ denotes the growth rate of $\beta_j(t)$. In the early phase, this dominant index corresponds to minor features (coordinates with small $\mu_j$), while in the later phase it switches to major features (coordinates with larger $\mu_j$). When SAM is focusing on minor features, the loss decreases slowly, leading to a plateau; once SAM shifts to major features, the loss drops much faster. In contrast, GD does not exhibit this minor-to-major feature focusing behavior, and its loss decreases more rapidly from the beginning, without such plateau.

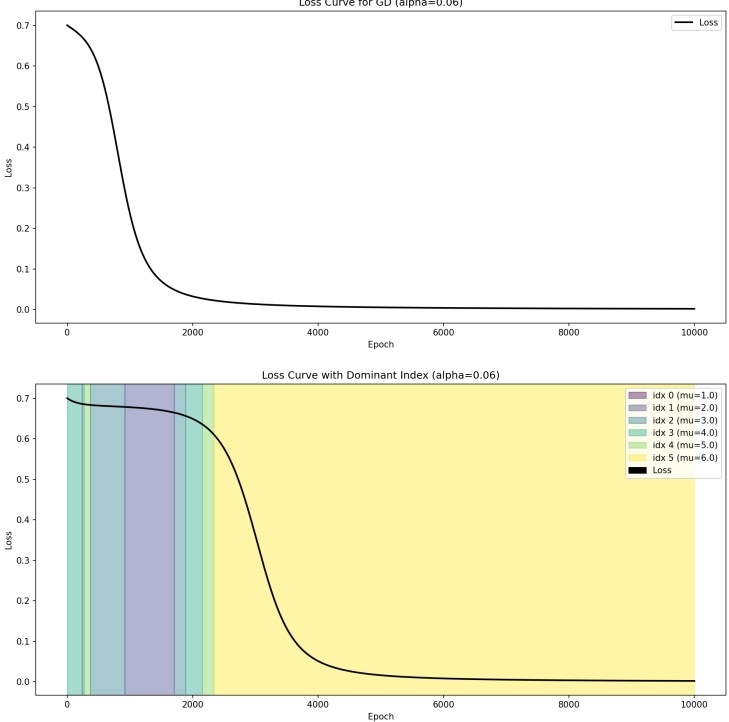

Figure 24: Training loss curves of GD (top) and SAM (bottom) on the 2-layer diagonal network in the intermediate initialization regime ($\alpha = 0.06$). The colored areas correspond to regimes where each feature is mostly amplified. Compared to GD, SAM exhibits an early plateau loss curve: in this phase, SAM primarily amplifies minor coordinates, leading to slow loss decrease. Once SAM shifts its focus to major coordinates, the loss drops rapidly. GD does not display this minor-to-major feature focusing behavior, thereby showing a more steadily decreasing loss without such a plateau.

## E.2  SEQUENTIAL FEATURE AMPLIFICATION UNDER RANDOM INITIALIZATION

In the main analysis, we focused on a symmetric and layer-wise balanced initialization to obtain a clean theoretical characterization. Here, we examine whether the sequential feature amplification phenomenon persists under more general random initialization.

We initialize the two layers independently as

$$\boldsymbol{w}^{(1)}(0), \boldsymbol{w}^{(2)}(0) \sim \mathcal{N}(0, \alpha^2 I),$$

where the parameter $\alpha$ controls the initialization scale as the standard deviation of the Gaussian distribution.

Figure 25a shows the normalized coordinate trajectories $\beta_j(t)/\|\boldsymbol{\beta}(t)\|_2$ under random initialization (Seed 0) for $\alpha = 0.65$, $\boldsymbol{\mu} = (1,2,3,4,5,6)$, and $\rho = 0.1$. In this case, all coordinates except the fourth are sequentially amplified, with activation progressing roughly from the second to the sixth coordinate. Correspondingly, Figure 25b shows that the layer-wise discrepancy $\|\boldsymbol{w}^{(1)}(t) - \boldsymbol{w}^{(2)}(t)\|_2$ rapidly decays to zero, indicating fast balancing of the two layers.

A qualitatively similar but quantitatively different pattern is observed under a different random seed. In Figure 25c (Seed 1), the sequential amplification begins from the third coordinate and proceeds toward the sixth. Despite this seed-dependent variation in the detailed activation order, the overall sequential feature amplification phenomenon persists. Moreover, Figure 25d confirms that the balancedness property is again achieved rapidly in the early stage of training.

These empirical observations are theoretically supported by Lemma D.5, which shows that even when the layers start from imbalanced initializations, the dynamics drive them toward a balanced regime exponentially fast. This explains why the simplified, balanced initialization assumed in the main analysis captures the essential behavior of the training dynamics beyond this restricted setting.

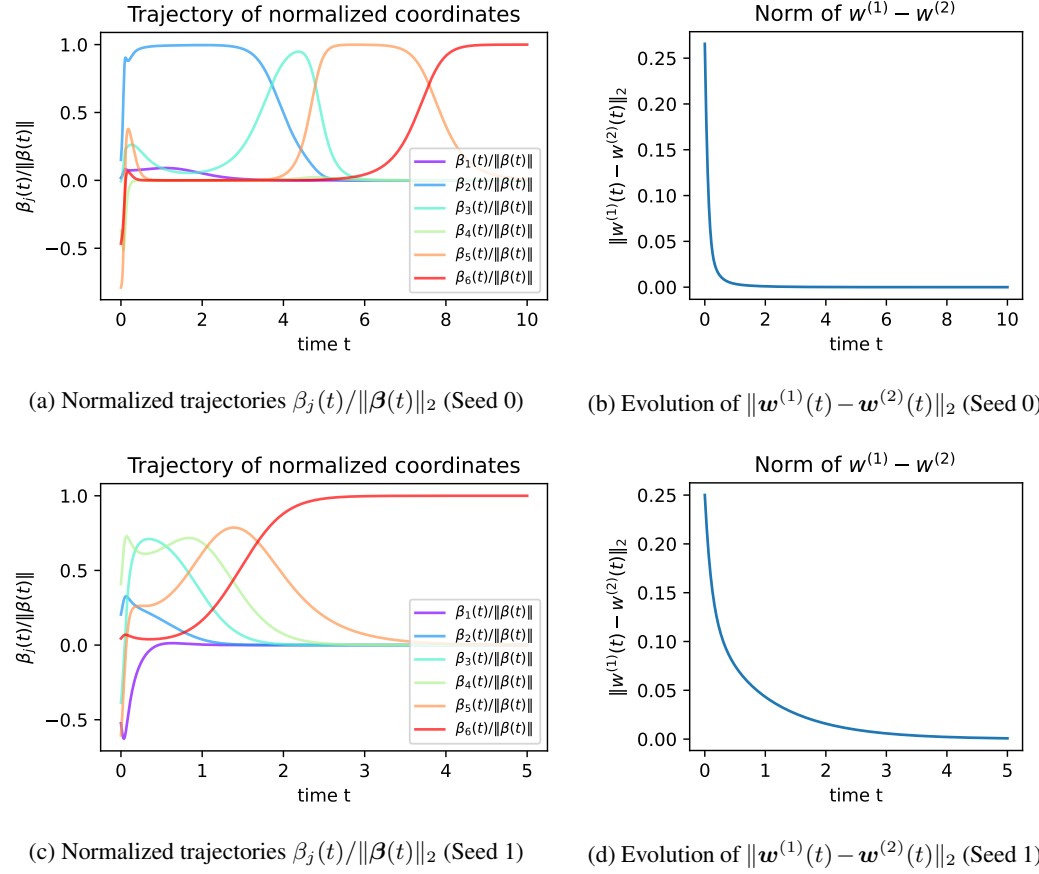

(a) Normalized trajectories $\beta_j(t)/\|\boldsymbol{\beta}(t)\|_2$ (Seed 0)  (b) Evolution of $\|\boldsymbol{w}^{(1)}(t) - \boldsymbol{w}^{(2)}(t)\|_2$ (Seed 0)

(c) Normalized trajectories $\beta_j(t)/\|\boldsymbol{\beta}(t)\|_2$ (Seed 1)  (d) Evolution of $\|\boldsymbol{w}^{(1)}(t) - \boldsymbol{w}^{(2)}(t)\|_2$ (Seed 1)

Figure 25: Sequential feature amplification under random initialization in a two-layer diagonal network. Rows correspond to different random seeds (Seed 0 and Seed 1), and columns correspond to different plot types (left: normalized coordinate trajectories, right: balancedness).

### E.3 ALTERNATIVE 2-LAYER MODELS

To evaluate the generality of our theoretical predictions, we conduct experiments on alternative 2-layer models featuring different parameterizations and metrics. In all cases, the experimental settings and hyperparameters are chosen to closely match those used in our main theoretical simulations with the diagonal network.

#### E.3.1 LINEAR NETWORK

We fix a small matrix dimension $d = 5$. All inputs are $d \times d$ matrices. We first draw a single random "signal" matrix $\mu \in \mathbb{R}^{d \times d}$ with i.i.d. standard normal entries, and then compute its singular value decomposition (SVD)

$$\mu = U_\mu \operatorname{diag}(S_\mu) V_\mu^\top.$$

From this SVD, we construct an orthonormal basis of rank-1 matrices

$$\mu_i = u_i v_i^\top, \quad i = 1, \dots, d,$$

where $u_i$ is the $i$-th column of $U_\mu$ and $v_i^\top$ is the $i$-th row of $V_\mu^\top$. These $\mu_i$ play the role of "feature directions", analogous to the coordinates in the diagonal model.

We use the logistic loss, and the dataset follows the same format as in the diagonal model: we consider the two points $\{+\mu, -\mu\}$ with opposite labels $\{+1, -1\}$. The 2-layer linear network is

$$f_\theta(X) = \langle \beta, X \rangle_F = \langle W^{(1)} W^{(2)}, X \rangle_F,$$

with learnable matrices $W^{(1)}, W^{(2)} \in \mathbb{R}^{d \times d}$ and effective weight $\beta = W^{(1)} W^{(2)}$. Each layer is initially set to the identity matrix, and before training we rescale all layers by a scalar $\alpha$, so that $W^{(1)}(0) = W^{(2)}(0) = \alpha I$ and hence $\beta(0) = \alpha^2 I$.

For training, we use full-batch SAM with radius $\rho = 0.5$, learning rate 0.05, and a finite training epochs of $T = 5000$. We repeat the experiment over a range of initialization scales, $\alpha \in \{0.20, 0.21, \dots, 0.70\}$.

As our tracking metric, we monitor the normalized squared alignment

$$a_i(t) = \frac{\langle \beta(t), \mu_i \rangle_F^2}{\|\beta(t)\|_F^2}, \quad i = 1, \dots, d,$$

where $\beta(t)$ denotes the effective weight at training iteration $t$.

The results are shown in Figure 26. As plotted in the figure, the dynamics of SAM and GD are qualitatively different. For SAM, when the initialization scale is smaller than 0.225, training does not converge to a solution with sufficiently small loss. Beyond this regime, as the initialization scale increases, the dominant singular direction that maximizes the alignment (i.e., $\arg\max_i a_i(T)$) moves from $\sigma_5$ to $\sigma_1$, indicating that SAM sequentially aligns from the minor component to the major component as $\alpha$ grows.

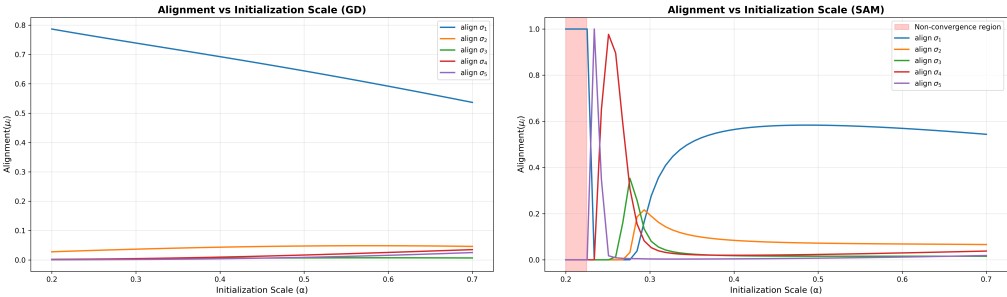

Figure 26: Alignment of the effective weight $\beta(t)$ for GD (left) and SAM (right) across initialization scales.

### E.3.2 CONVOLUTIONAL NEURAL NETWORK

We consider a 2-layer linear convolutional network trained on a synthetic dataset built from a single image matrix $\mu$. This experiment is designed to probe frequency-wise feature selection under SAM.

We fix an image size $d = 32$ and construct a single base image $\mu \in \mathbb{R}^{1 \times d \times d}$ as a sum of cosine plane waves with radial frequencies:

$$\mu(x, y) = \sum_{k=1}^{K} w_k \sum_{l=1}^{L_k} \cos\left(w\pi r_k \frac{x\cos\theta_{k,l} + y\sin\theta_{k,l}}{d} + \phi_{k,l}\right),$$

The experiment uses $K = 5$ different frequency bands, where $r_k$ are target bands, $w_k > 0$ are band weights, and $\theta_{k,l}$, $\phi_{k,l}$ are random orientations and phases for each band. We take $r_k \in \{3, 9, 11, 13, 15\}$ and $w_k = \{1.0, 2.0, 3.0, 4.0, 5.0\}$ for all $k$. We set $L_k = 8$ for all $k$. We then renormalize $\mu$ to have unit euclidean norm, then shift it slightly to be strictly positive. Next, we define the frequency bands by constructing radial masks $M_k \subset \{0, \cdots, d-1\}^2$ in the fourier domain. Let $\hat{\mu}$ denote the 2D FFT of $\mu$. The band energy of $\mu$ at band $k$ is then given by

$$\mu_k = \sum_{m \in M_k} |\hat{\mu}(m)|^2.$$

The bands are sorted by $\mu_k$. As we apply low weights to low frequency bands when constructing $\mu$, in this setting, low frequency bands have smaller $\mu_k$ and treated as minor features, and high frequency bands have larger $\mu_k$ and treated as major features.

The utilized model is a depth-2 convolutional network without nonlinearities. For the first convolutional layer, we use $3 \times 3$ convolution with 32 output channels, stride 1, and padding 1. For the second convolutional layer, we use same size of kernel, channel size, stride, and padding.

We used realistic gaussian initialization for the weights of the convolutional layers. The weights for each layer are independently initialized. Lastly, the final FC layer is a linear layer. the input for fc layer is squeezed 1d vector, and the output is a single logit.

Logistic loss is used, and full-batch training is employed. We use learning rate of $0.03$ and $\rho = 0.1$. We train for 6000 epochs.

**Band-wise effective weights.** To compare with the diagonal model, we require a band-wise decomposition of the effective weight $\beta(\theta)$ in input space. Since the network is linear, $\beta(\theta)$ can be recovered from gradients. At a given parameter vector $\theta$, we consider the empirical margin

$$s(\theta) = \mathbb{E}_{(x,y)} [y f_\theta(x)] = \frac{1}{2}(f_\theta(\mu) - f_\theta(-\mu)).$$

We compute the gradient of $s(\theta)$ with respect to the input and form a "virtual gate" version of $\beta$ in input space:

$$\nabla_x s(\theta)|_{x,y} = y (\nabla_x f_\theta(x)).$$

So,

$$\beta_{\mathrm{map}}(u, v) = \mathbb{E}_{(x,y)}\left[(\nabla_x f_\theta(x) \odot x)_{u,v}\right],$$

which is proportional to $(\beta(\theta) \odot \mu)_{u,v}$ in our linear setting. In practice, this expectation is computed exactly by averaging over $x \in \{\mu, -\mu\}$.

We then take the 2D FFT of $\beta_{\mathrm{map}}$, denoted $\widehat{\beta}_{\mathrm{map}}$, and define the band-wise effective weights by

$$\beta_k(\theta) = \sum_{m \in M_k} \left|\widehat{\beta}_{\mathrm{map}}(m)\right|^2.$$

For each training epoch $t$ we record the vector

$$(\beta_1(\theta_t), \ldots, \beta_K(\theta_t)),$$

and, in particular, the index of the dominant band

$$k_{\mathrm{dom}}(t) = \arg\max_k \beta_k(\theta_t).$$

In our initialization-scale experiments, we repeat this procedure over a range of $\alpha \in [0.13, 0.20]$ and, for each $\alpha$, track both the dominant band $k_{\mathrm{dom}}$ at the end of training. This provides a CNN analogue of the feature-selection behavior observed in the diagonal model, where coordinates are replaced by frequency bands.

Figure 27 displays how the final dominant frequency band selected by the CNN varies with the initialization scale $\alpha$. Consistent with expectations, when trained with SAM, the model emphasizes minor features (i.e., low frequency bands) for small $\alpha$, and shifts its focus to major features (high frequency bands) as $\alpha$ increases. In contrast, under standard GD, the dominant frequency band remains unchanged regardless of the initialization scale.

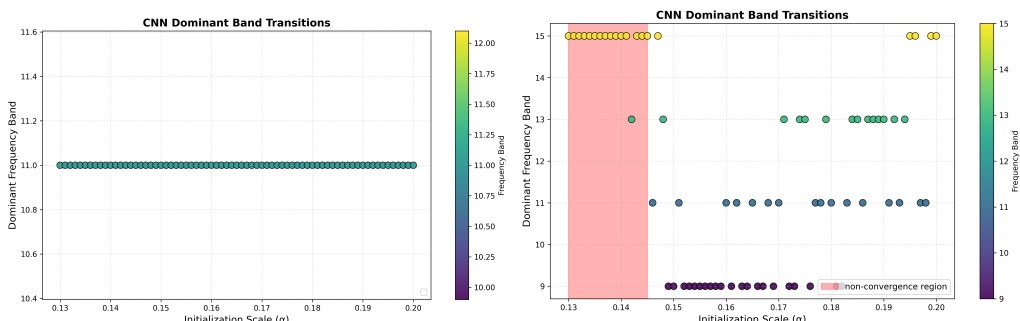

Figure 27: Dominant band for GD (top) and SAM (bottom) across gaussian initialization with different scales. Each point shows the dominant band (the band that model mostly focuses on) at the end of training; SAM systematically shifts from minor (low-frequency) to major(high-frequency) bands as $\alpha$ increases, whereas GD remains insensitive to $\alpha$.

### E.4 GRAD-CAM

As our theoretical analysis rigorously characterizes the dynamics of SAM in linear diagonal networks, we extend our empirical investigation to convolutional neural networks (CNNs) to examine whether the same phenomena persist in more realistic architectures. Combining the results for both $\ell_\infty$-SAM and $\ell_2$-SAM, our theory predicts three practical regimes: for small initialization scale $\alpha$, SAM collapses toward the origin; for large $\alpha$, SAM behaves similarly to GD; and for intermediate $\alpha$, SAM preferentially amplifies minor to intermediate features relative to GD.

To examine these predictions in practice, we train depth-2 CNNs with ReLU activations using both SAM and GD. We then apply Grad-CAM (Selvaraju et al., 2019; Gildenblat & contributors, 2021) to visualize which regions of the input image are emphasized by each model. In addition to qualitative visualizations, we compute the average values of pixels whose Grad-CAM activation exceeds a threshold (0.5) and plot this quantity as a function of the initialization scale $\alpha$. To characterize the sequential feature amplification as a function of the initialization scale, we rescale the default random initialization by multiplying it by $\alpha$ and train the model under this controlled initialization scheme. Unlike the theoretical setting of Theorem 4.5, which assumes a structured initialization, we use randomized initialization with rescaling in practice. In the corresponding figures, we indicate collapse-to-origin behavior in green and blow-up behavior in purple.

We conduct experiments on MNIST (Deng, 2012), SVHN (Netzer et al., 2011), and CIFAR-10 (Krizhevsky et al., 2009). Across all datasets, we consistently observe that GD-trained models concentrate on dominant, high-intensity pixels, whereas SAM-trained models emphasize lower-intensity, minor pixel regions. These results demonstrate that the distinct feature prioritization mechanism predicted by our theory persists in nonlinear CNN architectures.

### E.4.1 MNIST

We first study this phenomenon on MNIST. MNIST has a simple structure, where the black background takes the minimum pixel value (0) and the white digit takes the maximum pixel value (1).

We construct a subset of $1,000$ images whose labels are in $0, 1, 2, 3$ and train models using either GD or $\ell_2$-SAM. After training, we visualize the learned attention patterns using Grad-CAM, as shown in Figure 28. We observe that the GD-trained model primarily bases its predictions on the white digit region, whereas the $\ell_2$-SAM–trained model concentrates more strongly on the black background region. Unless otherwise stated, we use a learning rate of $0.1$, a SAM perturbation radius of $0.5$, and train for $500$ epochs with a batch size of $64$. We use no momentum and no weight decay. For the CNN architecture, we use $3 \times 3$ convolutional kernels and do not apply batch normalization or layer normalization.

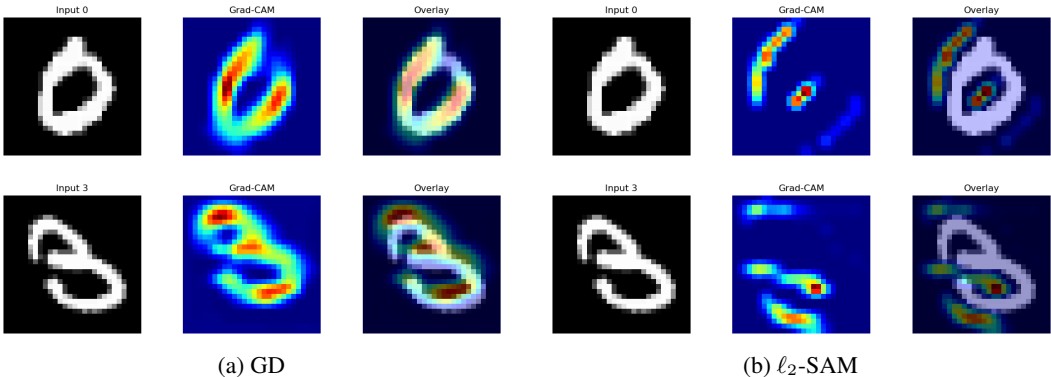

(a) GD

(b) $\ell_2$-SAM

Figure 28: Grad-CAM comparison between GD and $\ell_2$-SAM on MNIST (labels 0–3).

To study the practical behavior of $\ell_\infty$-SAM, we train models using $\ell_\infty$-SAM on a subset of $1,000$ MNIST images with labels in $\{0, 1\}$. We then visualize the Grad-CAM maps, as shown in Figure 29. We observe a bias pattern similar to that of $\ell_2$-SAM, where the model places greater emphasis on background regions corresponding to minor features. We use the same hyperparameters as in the previous experiment: learning rate $0.1$, perturbation radius $0.5$, training for $500$ epochs, and a batch size of $64$.

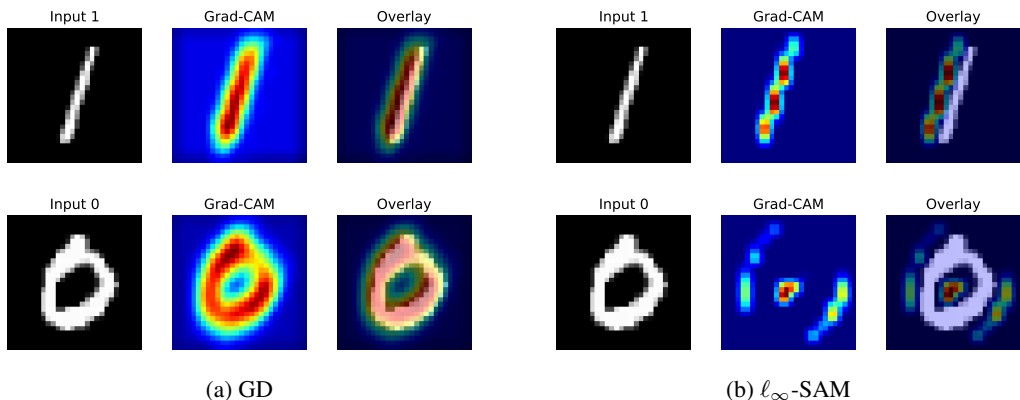

(a) GD

(b) $\ell_\infty$-SAM

Figure 29: Grad-CAM comparison between GD and $\ell_\infty$-SAM on MNIST (labels 0–1).

We now quantify the average values of activated pixels (Grad-CAM > 0.5) as a function of the initialization scale $\alpha$ across different dataset subsets. In this experimental setup (Figure 30), we observe that GD consistently concentrates more on the white digit region, which can be interpreted as the major component in the pixel value manner, unless GD fails to minimize the loss because of too large initialization scale. We denote as purple dots where GD blows up. Moreover, we observe

three regimes of $\alpha$ of SAM. We denote as green dots where too small initialization scale fails to escape near the origin and so the loss is not changed. Here can be seen as Regime 1. After that, SAM concentrates on the pixels whose average is almost 0, so the background region. This implies SAM concentrating on the minor component of the data more than GD, which can be seen as Regime 2. When GD blows up, SAM also goes out of the trend and almost blows up.

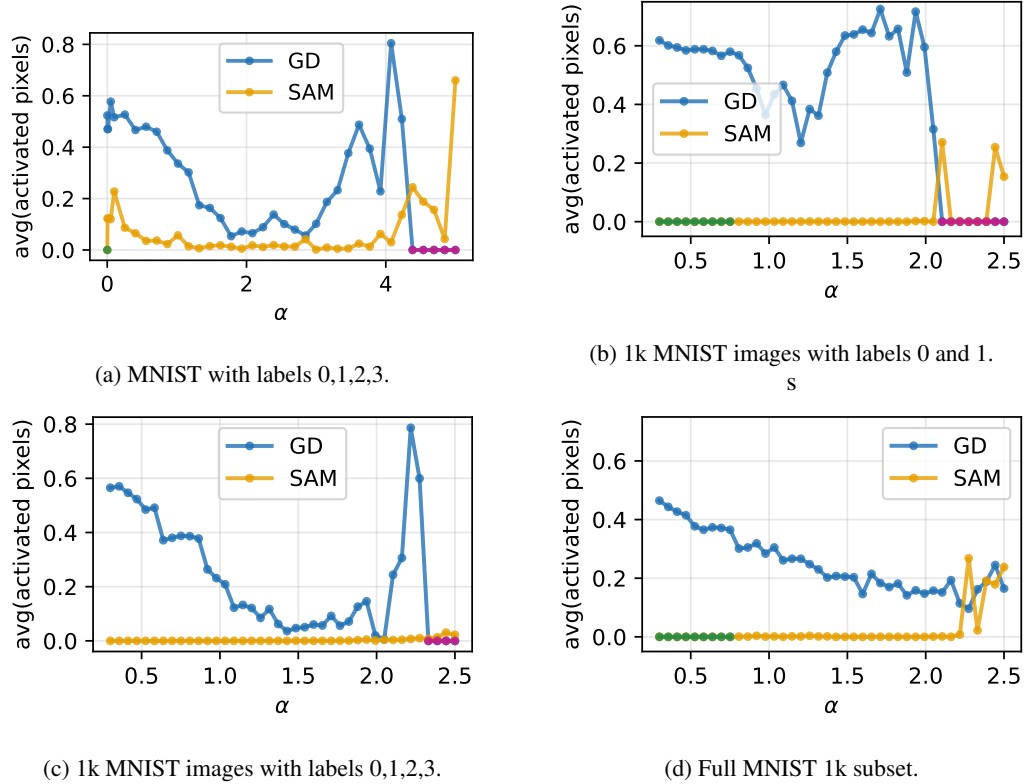

(a) MNIST with labels 0,1,2,3.

(b) 1k MNIST images with labels 0 and 1.
s

(c) 1k MNIST images with labels 0,1,2,3.

(d) Full MNIST 1k subset.

Figure 30: Average number of pixels with Grad-CAM activation exceeding 0.5 as a function of the initialization scale $\alpha$, comparing GD and $\ell_2$-SAM across different MNIST subsets.

$\ell_\infty$-SAM exhibits a similar pattern (Figure 31). When $\alpha$ is small, the dynamics collapse toward the origin. For intermediate values of $\alpha$, $\ell_\infty$-SAM tends to prioritize minor features more strongly than GD. For sufficiently large $\alpha$, however, the behavior of $\ell_\infty$-SAM deviates from this trend.

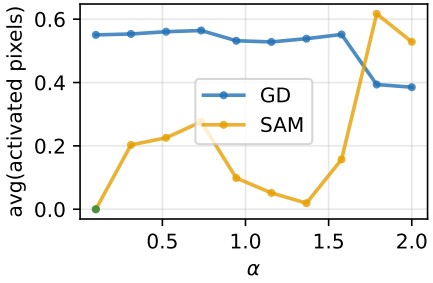

Figure 31: Average number of pixels with Grad-CAM activation exceeding 0.5 as a function of the initialization scale $\alpha$, comparing GD and $\ell_\infty$-SAM on 1k MNIST images with labels 0 and 1.

### E.4.2 SVHN

We next study this phenomenon on SVHN. SVHN is more complex than MNIST, as it contains both images with dark backgrounds and light digits, as well as images with light backgrounds and dark digits. Nevertheless, we observe that $\ell_2$-SAM consistently emphasizes the darker regions of the image.

We construct a subset of 1,000 images with labels in $\{0, 1\}$ and train models using either GD or $\ell_2$-SAM. We use a learning rate of 0.01, a SAM perturbation radius of 0.05, and train for 200 epochs.

The images in Figure 32 contain dark digits on light backgrounds. In this case, we observe that SAM concentrates more strongly on the digit regions than the background, as the digits constitute the minor features in these images. By contrast, the images in Figure 33 contain light digits on dark backgrounds. For these images, SAM concentrates more strongly on the background regions than on the digits, as the background constitutes the minor feature in this setting.

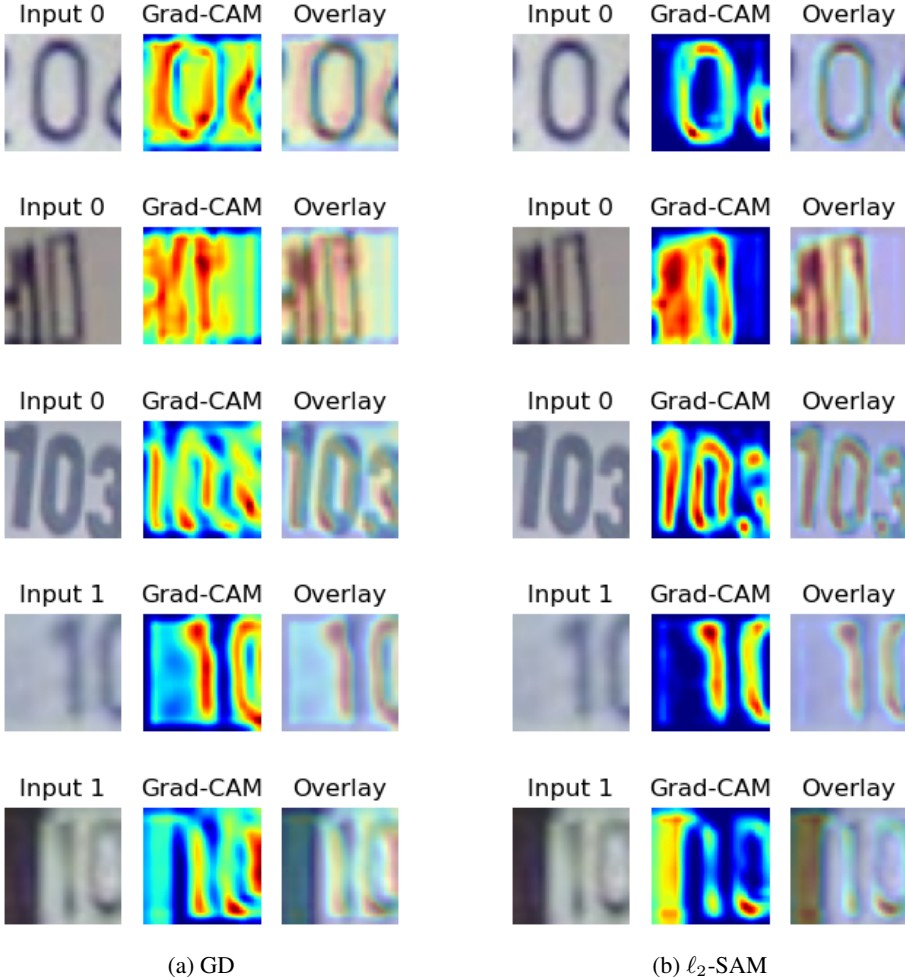

(a) GD        (b) $\ell_2$-SAM

Figure 32: Grad-CAM comparison between GD and $\ell_2$-SAM on SVHN (1k images, labels 0–1) with dark digits and light backgrounds.

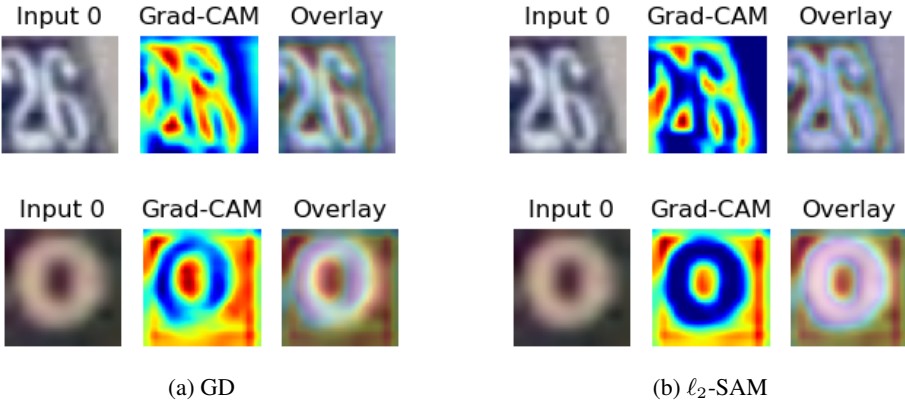

(a) GD            (b) $\ell_2$-SAM

Figure 33: Grad-CAM comparison between GD and $\ell_2$-SAM on SVHN (1k images, labels 0–1) with light digits and dark backgrounds.

Across different values of $\alpha$, we observe that small $\alpha$ causes $\ell_2$-SAM to collapse toward the origin, while intermediate $\alpha$ leads $\ell_2$-SAM to emphasize minor features with lower pixel intensities as shown in Figure 34, where pixel intensity is computed as the average over the three color channels.

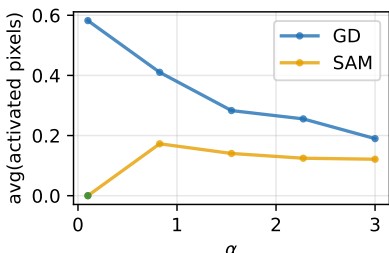

Figure 34: Average number of activated pixels (Grad-CAM > 0.5) as a function of the initialization scale $\alpha$, comparing GD and $\ell_2$-SAM.

### E.4.3   CIFAR-10

We also observe the same phenomenon on the CIFAR-10 dataset. We construct a subset of CIFAR-10 with labels in $\{0, 1\}$ and train models using a learning rate of $0.01$, a SAM perturbation radius of $0.05$, for $500$ epochs. As shown in Figure 35, small values of $\alpha$ lead SAM to emphasize minor features, while larger values of $\alpha$ make the behaviors of GD and SAM increasingly similar.

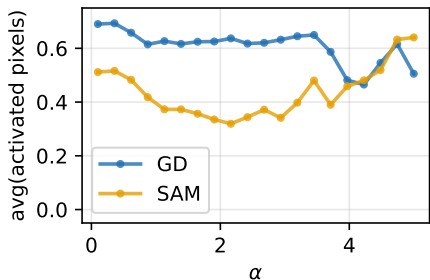

Figure 35: Average number of activated pixels (Grad-CAM > 0.5) as a function of the initialization scale $\alpha$, comparing GD and $\ell_2$-SAM.

