# OpenReview forum: "Minor First, Major Last: A Depth-Induced Implicit Bias of Sharpness-Aware Minimization"
_ICLR.cc/2026/Conference — ICLR 2026 Poster_

### Official Review · Reviewer_f8R2 · 2025-10-23

**Soundness:** 3
**Presentation:** 4
**Contribution:** 2
**Rating:** 4
**Confidence:** 4

**Summary:**

This paper studies the implicit bias of SAM for linear diagonal networks for a single data point example. While previous works have studied the regression problem, this work focuses on classification and identifies some new phenomena:
- for $\ell_\infty$ SAM with depth at least $2$, the model can converge towards directions that do not maximize the margin
- for $\ell_2$ SAM with depth $=2$, the authors identify a novel "sequential feature discovery" phenomenon where the network most significant coordinate can switch during training

**Strengths:**

The paper is well written and highlights some new unknown phenomena for learning with SAM that are quite interesting. In particular, I found the sequential feature discovery quite cool

**Weaknesses:**

The main limitation of this work is the single data point assumption. This work indeed solely focuses on a single point dataset example, which is obviously very toyish. I presume extending this work to multiple points is challenging, but I would like to have at least discussions on the challenges when considering multiple points and also simulations illustrating similar phenomena on multiple training examples would be cool.

I also have a question of how surprising/interesting is this sequential feature discovery phenomenon. First, in practice, SAM is often used with very small $\rho$, so that we can expect the relevant initialisation regime is $\alpha \gtrsim \rho$, so that we would be in the Regime 3. Actually, this sequential feature discovery illustrates how complex and non-monotonic the dynamics can be as soon as we leave the linear parametrization, but does it really say more than that? And in particular, is this property really specific to SAM (and not happening for GD)?

It is not so clear what Section 4.2.4 (and Proposition 4.7) in particular really says about this sequential discovery feature. In particular, it focuses on **lower bounds of the rates** but it seems we cannot say anything about what coordinate is really the dominant one at some fixed time. In particular, I do not see how these results explain what we see in Figure 3.

I also have a concern/remark about the time reparametrization done in Equation (2). At first sight, I would say the trajectories would only match if $$\int_{0}^\infty \ell'(\langle \beta(\hat{\theta}(t)),\mu\rangle)\mathrm{d} t = -\infty.$$
So I guess a justification of that would be needed. Also, this time reparametrization leads to finite time blow ups as discussed line 240. However I am not sure how to interpret the result in case of blow-up: the trajectory can be considered until the first (smallest) blow up time and nothing happens afterwards? If that would be the case, this means that only one coordinate converges to $\infty$ in case of blow up, while all the other ones remain bounded as $t\to\infty$. This seems very weird to me, as in general when overfitting in classification, the coordinates would all converge to infinity (although potentially at different rates). Maybe this is made possible by the $\ell_\infty$-SAM regularization, but I guess it should at least be discussed as it seems surprising to me.

Another hidden assumption is that $L(\theta(0))< L(\mathbf{0})$ (eg assuming $\alpha$ has only positive coordinates) which actually seems to play a critical role in the analysis. I think it would be needed to discuss how necessary this assumption is and what would happen otherwise.

----------------------
# Other remarks:
- Theorem 4.2 assumes three points (a,b,c), but are these needed? Moreover, the final Proposition 4.7 does not totally explain it: it says that any $j$
- sequential feature discovery made me think about *incremental learning*. Although this is different, it might be worth to cite the works [1] and [2] that characterize it for diagonal linear networks
- Figure 3 is very nice. I would have liked a comparison with GD somewhere here.
- Maybe recall before Theorem 4.5 that $0<\mu_1<\ldots<\mu_d$, as I believe this is where it is really needed (and not before) in the paper


----------------
# References

[1] Berthier, Raphaël. "Incremental learning in diagonal linear networks." Journal of Machine Learning Research 24.171 (2023): 1-26.

[2] Pesme, Scott, and Nicolas Flammarion. "Saddle-to-saddle dynamics in diagonal linear networks." Advances in Neural Information Processing Systems 36 (2023): 7475-7505.

**Questions:**

see above

---

> ### Author Response · Authors · 2025-11-23
>
> We sincerely thank the reviewer for the detailed and thoughtful feedback.
>
> ### **W1. Restricted dataset with one point.**
>
> As another reviewer also raised this point, we address it in our global response. We would be grateful if you could refer to that part for more details.
>
> ### **W2. Concern about whether sequential feature discovery still occurs with the small $\rho$ values typically used in practice.**
>
> We agree that, in practice, SAM is often used with a ``small'' perturbation radius $\rho$. As you also point out, however, what actually determines which regime we are in is not the absolute size of $\rho$, but its **scale relative to the initialization** $\alpha$ in $\mathbf{w}(0)=\alpha\mathbf{1}$. To relate our parameterization to standard practice, note that commonly used initializations correspond to
>
> - Xavier: $\alpha \approx \sqrt{\frac{1}{n}}$,
> - Kaiming/He: $\alpha \approx \sqrt{\frac{2}{n}}$,
>
> where $n$ is the layer width. For typical widths, this gives
>
> | **layer width $n$** | **Xavier $\alpha$** | **Kaiming $\alpha$** |
> | --- | --- | --- |
> | 128 | ~0.09 | ~0.125 |
> | 512 | ~0.044 | ~0.062 |
> | 2048 | ~0.022 | ~0.031 |
>
> We obtain $\alpha \approx 0.02–0.13$, which is of the same order as commonly used choices of $\rho$ such as $\rho \approx 0.05$. This means that for realistic widths and $\rho$ values, typical initialization scales naturally place the training near these regime boundaries, and small changes in $\alpha$ or $\rho$ can move the dynamics across our Regimes 1–3, including the sequential feature discovery regime. Thus, the phenomenon we analyze is not confined to an artificially tuned hyperparameter setting, but is expected to be relevant for realistic SAM configurations.
>
> At a high level, Regime 2 is the regime in which we can meaningfully explain why SAM behaves differently from GD and achieves improved performance. In our analysis, Regime 1 corresponds to settings where the loss remains high and optimization is ineffective, while Regime 3 corresponds to settings in which SAM and GD exhibit nearly identical dynamics. This makes Regime 2 the practically relevant regime for understanding SAM’s distinctive behavior.
>
> ### **W3. Sequential feature discovery may add limited new insight beyond showing complex non-linear dynamics.**
>
> First, the sequential feature discovery reveals that **SAM is highly sensitive to the initialization scale**, in sharp contrast to GD. As shown in Figure 5(a), GD consistently recovers the major feature regardless of the initialization scale $\alpha$ or the time $t$, whereas SAM exhibits qualitatively different behaviors under different initialization scales. To the best of our knowledge, our results show, for the first time, that—even in logistic loss, where the weights diverge rather than converge to a finite minimizer—the initialization scale can induce unexpected and qualitatively different behaviors of SAM.
>
> Second, our results highlight that **focusing solely on the infinite-time limit direction—as is common in prior work on logistic loss—is insufficient for understanding SAM**. As illustrated in Figure 1(b), SAM’s finite-time dynamics differ substantially from its asymptotic $\ell_1$ max-margin direction. Motivated by this, we formally show that the normalized ascent step of SAM introduces pre-asymptotic behaviors that do not appear in the infinite-time analysis. This suggests that, for SAM and potentially other optimizers, **finite-time behavior can be as important as the limiting direction**.
>
> Third, this phenomenon highlights **the challenges in analyzing SAM’s dynamics**, which stem from the normalized ascent step. Because the perturbation depends on the entire gradient direction, the resulting dynamics become highly nonlinear and coupled. We believe that carrying out a formal analysis of this complex behavior and identifying sequential feature discovery is therefore valuable in its own right.

---

> ### Author Response · Authors · 2025-11-23
>
> ### **W4. Sequential feature discovery is specific to SAM?**
>
> We indeed find that the observed phenomenon is **specific to SAM and does not occur under standard GD**, and we have added additional empirical evidence to clarify this distinction.
>
> We directly compared SAM and GD under identical settings. As shown in Figure 3(a), SAM exhibits a clear sequential feature discovery pattern depending on initialization scale. In contrast, Figure 5(a) shows that GD consistently converges to the major feature regardless of initialization scale or time, resulting in a uniformly "all-red" heatmap. To further strengthen this point, we added:
>
> - Figure 11 (GF vs. discrete GD under the same metric),
> - Figure 16 (GF and discrete GD under multi-point setting), and
> - Figure 18 (GF under deeper networks),
>
> all of which exhibit trivial "all-red" behavior for GD (with only negligible gray regions under small initialization). These results consistently show that **GD does not exhibit the sequential behavior** observed in SAM.
>
> Our results on alternative two-layer models further indicate that sequential feature discovery is specific to SAM. For two-layer linear models, Figure 22 shows that SAM exhibits sequential feature discovery, whereas GD displays unchanged behavior across different values of $\alpha$. For two-layer convolutional neural networks, Figure 23 similarly shows that, unlike SAM, GD does not change its dominant frequency band.
>
> One might relate our phenomenon to **saddle-to-saddle dynamics or incremental learning** in diagonal linear networks. However, we emphasize that the underlying mechanisms are fundamentally different. We now provide an explicit comparison in Lines 128–137 and Section A.2, to which we respectfully refer the reviewer.
>
> ### **W5. Unclear link between lower-bound analysis in Section 4.2.4 and the sequential feature discovery.**
>
> The sequential feature discovery itself is explained by Theorem 4.5. Section 4.2.4 is not intended to *derive* the phenomenon directly but to **analyze it more deeply through cumulative lower-bound arguments**. We agree that these results cannot by themselves determine which coordinate becomes dominant at a fixed time, since they provide only lower bounds, not a full closed-form characterization.
>
> In fact, we derive an exact expression for $\beta_j(t)$ conditional on the term $I(t):=\int_0^t \frac{1}{n_{\boldsymbol{\theta}}(s)} ds$, but $I(t)$ cannot be solved in closed form; we only obtain upper and lower bounds. Thus, the best one can hope for in this setting is an inequality-based characterization, not an explicit solution.
>
> That said, the purpose of Theorem 4.6 not solely about identifying the dominant index $j^{\dagger}$. **Theorem 4.6 goes further by quantifying how large the coordinates must grow**, thereby strengthening the qualitative picture. We realized that this intention was not clearly explained in the original version, so we added clarifying text (lines 496–499) and a numerical illustration of $\mathrm{LB}_j(\alpha)$ in Appendix D.7.
>
> Across various choices of $\boldsymbol{\mu}$, we observe that $\mathrm{LB}_j(\alpha)$ frequently exceeds $10$, indicating that minor or intermediate coordinates can become **over 10 times larger** than the major coordinate—showing substantial amplification. This lower-bound analysis does not determine the exact maximizer at a specific time, but it matches key trends: (i) **the amplification grows stronger with smaller $\alpha$**, and **larger spectral contrast** $\mu_d/\mu_1$, and (ii) the maximized feature $j^*$ shifts toward major feature as $\alpha$ increases, exhibiting the **initialization-wise sequential feature discovery** described in Proposition 4.7. In this sense, Theorem 4.6 provides **indirect but meaningful theoretical support** for the phenomenon, even though it does not provide a full characterization.

---

> ### Author Response · Authors · 2025-11-23
>
> ### **W6. Unclear justification of the time reparametrization.**
>
> Thank you for the insightful comments. We agree that the way the time reparameterization was originally presented (Equation (2) and line 240 in the original version) could be confusing. In the revision, we clarify this point in lines 194–208.
>
> We denote the original SAM flow by $\mathbf{w}\_{\text{orig}}(t_\text{orig})$ and the rescaled flow by $\mathbf{w}(t)$. There exists a strictly increasing map $t\_\text{orig}=\tau(t)$ such that
>
> $$
> \mathbf{w}_{\text{orig}}(\tau(t))=\mathbf{w}(t).
> $$
>
> Applying the chain rule gives
>
> $$
> \frac{d \mathbf{w}}{dt}
> = \frac{d \mathbf{w}_{\mathrm{orig}}}{d\tau}\frac{d\tau}{dt}
> = -\frac{\nabla\mathcal{L}(\hat{\mathbf{w}}(t))}{\ell'(\boldsymbol{\beta}(\hat{\boldsymbol{\theta}}(t))^{\top}\boldsymbol{\mu})},
> \qquad
> \frac{d\tau}{dt}
> = -\frac{1}{\ell'(\boldsymbol{\beta}(\hat{\boldsymbol{\theta}}(t))^{\top}\boldsymbol{\mu})}.
> $$
>
> and we have
>
> $$
> \tau(t)
> = \int_{0}^{t} -\frac{1}{\ell'(\boldsymbol{\beta}(\hat{\boldsymbol{\theta}}(s))^{\top}\boldsymbol{\mu})}ds.
> $$
>
> We do not require that $\tau(t) \to \infty$ implies  $t\to\infty$ in our analysis. In fact, the two time parameterizations need not preserve divergence of time variables. As illustrated by the finite-time blow-up case in Theorem 3.2, it can happen that the rescaled time remains finite while the original time diverges. For this reason, the integral condition proposed by the reviewer is not necessary in our setting. What our arguments fundamentally rely on is that the original and rescaled flows trace out the same spatial trajectory in parameter space, rather than that they share the same notion of ``infinite time''.
>
> ### **W7. Unclear interpretation of the finite-time blow-up behavior.**
>
> Thanks again for raising this question and we have added Remark 3.3 and Remark 3.4 to clarify the interpretation of the finite-time blow-up and Lines 261-263 to explain the distinctive property of the $\ell_\infty$-SAM analysis. The seemingly unusual finite-time blow-up behavior comes entirely from the **rescaled** $\ell_\infty$-SAM flow, where the dynamics become fully *coordinate-wise decoupled* on the single-example dataset.
>
> For the one-point dataset $\mathcal{D}_{\boldsymbol{\mu}}=\{ (\boldsymbol{\mu}, +1)\}$, the gradient takes the form (Equation (5))
>
> $$
> \nabla_{\mathbf{w}^{(i)}} \mathcal{L}(\boldsymbol{\theta}) = \ell'\big(\langle \boldsymbol{\beta}( \boldsymbol{\theta}),\boldsymbol{\mu}\rangle\big)\boldsymbol {\mu} \odot \Big( \bigodot_{\ell\ne i} \boldsymbol{w}^{(\ell)} \Big),
> $$
>
> and the $\ell_\infty$-SAM perturbation
>
> $$
> \boldsymbol{\varepsilon}_\infty(\boldsymbol{\theta}) := \rho\ \mathrm{sign}(\nabla \mathcal{L}(\boldsymbol{\theta}))
> $$
>
> depends only on the *signs* of the coordinates. Thus, in this special setting, every coordinate evolves independently under the perturbation, and the rescaled $\ell_\infty$-SAM flow in Equation (2) *separates into $d$ one-dimensional ODEs*. This is why Theorem 3.2 characterizes each coordinate $\beta_j(t)$ separately and why the condition $\alpha_j \gtrless \rho$ applies coordinate-wise.
>
> Let $t\_\text{min}^\*:=\min_{j: \alpha_j>\rho} t_j^\*$ be the earliest blow-up time from Theorem 3.2. As $t \to t^*\_\text{min}$, only the corresponding coordinate diverges in the rescaled system, while the others remain finite because the coordinates do not interact. Converting back to the original $\ell_\infty$-SAM flow time scale,
>
> $$
> \tau(t)
> = \int_{0}^{t} -\frac{1}{\ell'(\boldsymbol{\beta}(\hat{\boldsymbol{\theta}}(s))^{\top}\boldsymbol{\mu})}ds,
> $$
>
> the fact that $\boldsymbol{\beta}(\hat{\boldsymbol{\theta}}(t))^{\top}\boldsymbol{\mu} \to \infty$ implies $\ell'(\cdot)\to0^-$, hence
>
> $$
> \tau(t)\to\infty \quad \text{ whenever }t \to t_\text{min}^*.
> $$
>
> Thus, due to the complete coordinate-wise decoupling in the rescaled $\ell_\infty$-SAM flow, only the coordinate achieving the earliest blow-up time diverges as $\tau \to \infty$, while all other coordinates remain bounded.

---

> ### Author Response · Authors · 2025-11-23
>
> ### **W8. Reliance on the assumption $\mathcal{L}(\boldsymbol{\theta}(0))<\mathcal{L}(\boldsymbol{0})$.**
>
> Our theoretical analysis indeed assumes a symmetric and balanced initialization. Specifically, for $\ell_2$-SAM analysis, we initialize $\boldsymbol w^{(1)}(0)=\boldsymbol w^{(2)}(0)=\alpha \boldsymbol 1$ with $\alpha>0$, which implies a positive initialization and ensures that $\mathcal{L}(\boldsymbol \theta(0))<\mathcal{L}(\boldsymbol 0)$.
>
> This assumption plays a technical role: it removes inter-layer imbalances and allows us to focus on the effect of each feature coordinate $\mu_j$ on the training dynamics, while also enabling an unambiguous control of the initialization scale through the single parameter $\alpha$. The coordinate-wise and layer-wise symmetric initialization makes the dynamics more analytically tractable.
>
> To assess how essential this assumption is in practice, we conduct additional numerical experiments with **random initialization**, where
>
> $$
> \boldsymbol w^{(1)}(0), \boldsymbol w^{(2)}(0) \sim \mathcal{N}(0,\alpha^2 \boldsymbol I).
> $$
>
> In this setting, $\alpha$ controls the standard deviation of the initialization rather than the deterministic scale.
>
> Figure 21 shows the results for $\alpha=0.65$, $\boldsymbol\mu=(1,2,3,4,5,6)$, and $\rho=0.1$ under random initialization. Figures 21(a) and 21(c) demonstrate that the sequential feature discovery phenomenon persists, but the detailed activation pattern varies with the random seed. For example, under one seed the coordinates from 2 through 6 (except  the fourth) are sequentially amplified, whereas under another seed the amplification starts from the third coordinate and proceeds toward the sixth. This indicates that while the order of amplification depends on the initialization seed, the overall sequential structure is consistently preserved even without positive, symmetric initialization. Moreover, Figures 21(b) and 21(d) show that the two layers rapidly become balanced in practice across both random seeds, as the quantity $\\|\boldsymbol w^{(1)}(t)-\boldsymbol w^{(2)}(t)\\|$ decays sharply toward zero in the early stage of training.
>
> To theoretically justify this robustness, we prove in Lemma D.5 that when the two layers start from *unbalanced* initializations, they rapidly become balanced: $\\| \boldsymbol w^{(1)}(t)- \boldsymbol w^{(2)}(t)\\|$ decays exponentially fast to zero. This shows that layer-wise balance is not a fragile assumption but an emergent property of the dynamics. Even when $\boldsymbol w^{(1)}(0)\neq \boldsymbol w^{(2)}(0)$ and $\mathcal{L}(\boldsymbol \theta(0))$ is not necessarily below $\mathcal{L}(\boldsymbol 0)$ initially, the dynamics quickly enter a balanced regime.
> Therefore, the assumption of positive, layer-wise equal initialization is not strictly necessary for the empirical phenomenon of sequential feature discovery. Rather, it provides a clean analytical starting point, while in more general random settings, the dynamics self-regularize toward this regime in the early training phase.
>
> ### **R1. Use of the assumptions in Theorem 4.2 and the explanation of Proposition 4.7.**
>
> You are correct that the original presentation of Theorem 4.2 included more assumptions than necessary. In the revised version, we remove the directional-convergence assumption on the gradients and relax the required convergence of the inner and outer layers $\boldsymbol w^{(1)},$ $\boldsymbol w^{(2)}$ to that of the predictor $\boldsymbol \beta := \boldsymbol w^{(1)} \odot \boldsymbol w^{(2)}$. We show that the result holds under only the following two conditions: **(i) vanishing loss** and **(ii) convergence of the predictor**. This simplifies the theorem and the proof is improved. Moreover, we generalize the original single-sample version of Theorem 4.2 to general linearly separable datasets, which makes the statement more broadly applicable.
>
> In addition, Theorem 4.8 provides a complementary result showing that, in the special case of a single-example dataset, the predictor $\boldsymbol{\beta}(t)$ converges in the $\ell_1$ max-margin direction **without any assumption,** as long as the initialization scale $\alpha$ is larger than the threshold.
>
> Regarding your comment on Proposition 4.7, we are not fully certain which specific gap you intended to highlight, as the sentence appears truncated. If you could restate the concern, we would be happy to address it in detail.
>
> ### **R2. Relation to incremental learning.**
>
> Thank you for the comment. We agree that the relation to incremental learning should be discussed and have added a comparison and citations to [1,2] in Lines 128–137 and Section A.2.
>
> Although the two phenomena appear related, they are fundamentally different: prior works study successive coordinate **activation** from the zero initialization, whereas our work studies successive **amplification** from a dense initialization. We further clarify this distinction in Section A.2.

---

> ### Author Response · Authors · 2025-11-23
>
> ### **R3. Comparison with GD in Figure 3.**
>
> Thank you for the positive comment. As noted in line 360, we already included the GF result in the appendix (Figure 5). Its behavior is so simple that we did not place it in the main text. GF always concentrate on the major feature regardless of the initialization scale $\alpha$ or the time $t$, producing an entirely red heatmap.
>
> To clarify the contrast between GD and $\ell_2$-SAM, the revised version now additionally includes Figures 11, 16, and 18, which show GD and GF results for single-sample datasets, multi-sample datasets, and deeper models. Across all settings, GD/GF never show sequential feature discovery and immediately collapse to the major feature, confirming that the behavior in Figure 3 is unique to SAM.
>
> ### **R4. Placement of a key assumption before Theorem 4.5.**
>
> Thank you for the suggestion. We have implemented this change in the revised version.

---

> ### Comment · Reviewer_f8R2 · 2025-11-24
>
> I thank the authors for their detailed answer and updated version of the document. Most of my concerns have been addressed and I will thus raise my score in consequence.
>
> I still have one last question about the finite time blow up phenomenon. The authors claim in their answer:
> > Thus, due to the complete coordinate-wise decoupling in the rescaled -SAM flow, only the coordinate achieving the earliest blow-up time diverges as , while all other coordinates remain bounded.
>
> As explained in my original review, this is somewhat surprising to me (and thus very interesting). My next question is then: do we observe such a phenomenon (of bounded non-zero coordinates while the largest one explodes) in practice? Or said differently: do we have finite time blow up so we actually observe this, or do we actually never observe this (suggesting there is actually no finite time blow up happening)?
>
> I hope my question is clear enough and would be glad to read the authors' answer to it.

---

> > ### Author Response · Authors · 2025-12-03
> >
> > Thank you very much for your thoughtful follow-up question and for your positive assessment of our work. We also appreciate your careful attention to the finite-time blow-up phenomenon. To avoid any possible confusion, we have revised the theorem statement accordingly: what was previously denoted by $t_{\min}^*$ is now renamed as $T$, the rescaled blow-up time.
> >
> > Regarding your main question—whether the behavior “one coordinate explodes while all other non-zero coordinates remain bounded’’ is actually observed—we confirm that this phenomenon indeed appears in our experiments. We provide a concrete example in Section C.4 of the revised manuscript. In this example, under the original $\ell_\infty$-SAM flow, as $t \to \infty$, only the dominant coordinate (aligned with $\boldsymbol e_d$) grows without bound, while every other coordinate becomes more and more negligible in the normalized trajectory.
> >
> > In the rescaled $\ell_\infty$-SAM flow, the same phenomenon is visible but occurs within a finite rescaled time $T$: the trajectory is simply reparameterized so that the blow-up happens at a finite time. The remaining coordinates stay bounded up to $T$, which follows from the coordinate-wise decoupling of the rescaled system.
> >
> > We hope this clarifies your question.

---

### Official Review · Reviewer_uC1X · 2025-10-24

**Soundness:** 2
**Presentation:** 2
**Contribution:** 3
**Rating:** 4
**Confidence:** 4

**Summary:**

This work investigates the implicit bias of two versions of SAM related to the $L_2$ and $L_{\infty}$ norm in a classification setting whereas most previous work focusses on the finite mean squared loss regression setting. The analysis is based flow arguments for a data set of size one. They find that the SAM can learn features in different sequences whereas gradient descent would always prefer the major feature first. This provides a potential mechanism to explain the success of SAM. In the appendix their finding is substantiated by an experiment with Gradcam on a vision task.

**Strengths:**

Fully distinguishing the implicit bias induced by SAM and GF is definitely an important open problem. This work reveals a new phenomenon for the SAM optimizer: learning minor features first then major features first. The use of a single sample and the classification setting is original compared to previous works. The work gives good theoretical characterization what happens before reaching the margin. The theory is illustrated for small example in the appendix, indicating that the result might transfer to more complex settings.

**Weaknesses:**

1) Sequential feature discovery is a known phenomenon for gradient flow at least also known as saddle to saddle dynamics [1]. There are more works showing this for the gradient flow setting. It would be good to reposition the claim in this light as sequential feature discovery for SAM.

2) There is recent work on margin classification for mirror flow and steepest descent flow which cover more ground, how is the analysis for SAM more difficult? Can the authors contrast their work with [2,3]? What would we need to do for an extension or what prevents it?

3) The work is dense making it hard to value each part of the analysis. I would like to propose to put some parts in the appendix completely and provide more details of the current proofs in the main text and make clear where the text starts again and details of the proof are given. For instance highlight how Theorem 4.5 is proven and put the helper lemmas and propositions to the appendix while still highlighting the dynamical description as this important for intuition as well.

4) The experimental validation should be put in the main text as it helps substantiate the claims made. In addition the experimental validation is limited and could be improved to larger scale settings. Additional data analysis of the experiments is also needed. For example rescaling will most likely have changed the final performance of the model. Moreover, hyperparameters play an important role in these type of experiments and the architecture used these would need be mentioned in a table.

6) If SAM learns minor features first can we not see this in the loss curve trajectories in training i.e. GF should learn faster than SAM? Please show a loss plot illustrating this.

Minor points:
In appendix D the titles could be improved by using $L_{\infty}$, also in the text line 2194 it is said L2 sam is used but in the figures it is Linfty, which is it?

Typo line 364 coordinate

Formulation: line 451 "this is consistent with figure" should it not be an illustration of or this is substantiated by

[1] Pesme, Scott and Nicolas Flammarion. “Saddle-to-saddle dynamics in diagonal linear networks.” Journal of Statistical Mechanics: Theory and Experiment 2024 (2023): n. pag.

[2] Pesme, Scott et al. “Implicit Bias of Mirror Flow on Separable Data.” ArXiv abs/2406.12763 (2024): n. pag.

[3] Tsilivis, Nikolaos et al. “Flavors of Margin: Implicit Bias of Steepest Descent in Homogeneous Neural Networks.” ArXiv abs/2410.22069 (2024): n. pag.

While the work makes an interesting finding the current manuscript may need a substantial rewrite and more experimental illustrations given the limited theory setup.

**Questions:**

See weaknesses.

---

> ### Author Response · Authors · 2025-11-23
>
> We thank the reviewer for the helpful comments.
>
> ### **W1. Relation to saddle-to-saddle dynamics.**
>
> As the reviewer pointed out, at a high level, the sequential feature dynamics we study for SAM resemble the saddle-to-saddle behavior of gradient descent. However, the underlying mechanism, the dynamics (continuous minor-to-major feature amplification vs. discrete coordinate activations), and the setting (SAM with logistic loss vs. GF with squared loss) are fundamentally different. We have clarified this distinction in the revised Related Work section (Section 1.2) and have provided a detailed comparison in Appendix A.2.
>
> ### **W2. Comparison with Mirror-Flow and Steepest-Descent Margin Analyses.**
>
> In mirror flow and steepest descent flow, the dynamics are governed by first order ODEs where the geometry is fixed by a norm or mirror map. However, SAM uses a two-step update with an inner maximization at each step (controlled by the perturbation radius $\rho$), so the effective gradient depends on second-order information and is not clearly representable as a steepest descent flow / mirror flow. Moreover, unlike the learning rate, we cannot send $\rho \to 0$ in our regime.
>
> Therefore, extending the tools of [2,3] to SAM would require new techniques for this "bi-level" dynamics. In fact, the inner step destroys the simple homogeneity structure exploited in those works.
>
> In detail, the $\ell_2$-SAM update has the form
>
> $$
> \boldsymbol \theta (t+1) := \boldsymbol \theta (t) -\eta \nabla \mathcal{L}(\boldsymbol\theta (t) + \boldsymbol \varepsilon_2(\boldsymbol\theta (t))), \quad \boldsymbol \varepsilon_2(\boldsymbol \theta) := \rho\ \frac{\nabla \mathcal{L}(\boldsymbol \theta)}{\\|\nabla \mathcal{L}(\boldsymbol \theta)\\|_2}
> $$
>
> If we taylor expand around $\boldsymbol \theta$,
>
> $$
> \nabla \mathcal{L} (\boldsymbol \theta + \boldsymbol \varepsilon_2(\boldsymbol \theta)) \approx \nabla \mathcal{L}(\boldsymbol \theta) + \nabla^2 \mathcal{L}(\boldsymbol \theta) \boldsymbol \varepsilon_2(\boldsymbol \theta) + \text{higher order terms}.
> $$
>
> So the SAM update direction is approximately $-\nabla \mathcal{L}(\boldsymbol \theta) - \nabla^2 \mathcal{L}(\boldsymbol \theta) \boldsymbol \varepsilon_2(\boldsymbol \theta)$. Even if the model is homogeneous so that $\nabla \mathcal{L}(\boldsymbol \theta)$ scales nicely with $\boldsymbol \theta$, the second order term does not scale in the same way, because
>
> - $\boldsymbol \varepsilon(\boldsymbol \theta)$ is normalized by $\\|\nabla \mathcal{L}(\boldsymbol \theta)\\|_2$, and has fixed magnitude $\rho$ independent of $\\| \boldsymbol \theta\\|$.
> - The Hessian term includes curvature information that depends on the absolute location in parameter space, not just the direction of $\boldsymbol \theta$.
>
> Thus the SAM direction is not simply proportional to $\nabla \mathcal{L}(\boldsymbol \theta)$. This breaks the homogeneity-based rescaling invariance used in [2,3], so their margin analyses do not directly extend to SAM dynamics.
>
> ### **W3. Improving Readability and Proof Structure.**
>
> We thank the reviewer for the concrete suggestions. We agree that the arguments around Theorem 4.5 were dense, and we carefully explored the reviewer’s proposal of moving the helper results to the appendix and keeping only the theorem statement in the main text. However, we found that removing Lemma 4.3 and Proposition 4.4 made the logical flow harder to follow, since one needs to define the quantity $m_c(t)$, which is naturally derived from Lemma 4.3, in order to state and understand Theorem 4.5. For this reason, we chose to retain these two components in the main text, while also summarizing the proof idea at a high level right after Theorem 4.5.
>
> We also note that the proofs of all theorems other than Theorem 4.5 are deferred to the appendix, already aligning with the reviewer’s suggestion. In our revised version, we enhanced readability by adding clarifying comments and discussions, addressing the issues and questions raised by the reviewers.

---

> ### Author Response · Authors · 2025-11-23
>
> ### **W4. Experimental validation and extensions.**
>
> We thank the reviewer for the suggestion. We have revised the experiment section as follows.
>
> **(i) Experiments in the main text:** We now summarize the key experimental findings directly in the main text and refer to Appendix E for full details.
>
> **(ii) Additional experiments larger-scale settings, and further data analysis:** In Section E.3, we added more experiments on a 2-layer CNN and a 2-layer linear network, where we systematically vary the dataset construction and the corresponding metric as we move to different architectures. In Section C.4 and D.8, we added experiments on multi-point and deeper-depth settings, and showed that these scenarios also exhibit similar sequential feature discovery phenomena. For depth larger than 2, we additionally extend our theoretical analysis (Lemma 4.3 and Proposition 4.4) to match these observations.
>
> **(iii) Practical scenarios:** In Section E.4, we include experiments on practical CNN models trained on MNIST, SVHN, and CIFAR-10, where we track which image pixels the model focuses on. These experiments demonstrate SAM’s bias toward minor features in more realistic vision settings.
>
> **(iv) Architecture details:** We added precise descriptions of the model architectures and all hyperparameters used in the experiments (including learning rates, SAM radius, number of epochs, and initialization scales) to improve clarity and reproducibility.
>
> ### **W5. Illustration of loss curve.**
>
> We have added a loss comparison between SAM and GD in the 2-layer diagonal setting (see Section E.1 and its figure). In the intermediate initialization regime, the SAM loss curve is indeed flatter in the early phase, while GD’s loss decreases faster from the start, which is consistent with our interpretation that SAM initially prioritizes minor features before focusing on the major ones.
>
> ### **Minor points.**
>
> Thank you for pointing out the typos. We have revised Lines 364 and 451, as suggested by the reviewer. Regarding Line 2194 in our original submission, we conducted the experiment on both $\ell_2$- and $\ell_\infty$-SAM; the text mentioning only $\ell_2$-SAM was a mistake and the captions were correct. In our current revised version, we present Grad-CAM experiments with both $\ell_2$- and $\ell_\infty$-SAM.
>
> For the titles in Appendix D, we are afraid we did not fully understand your suggestion about using $L_{\infty}$. Could you clarify what specific change you have in mind?

---

> > ### Comment · Reviewer_uC1X · 2025-11-27
> >
> > I would like the authors for their extensive and thorough response.
> >
> > **Clarifications W1 and W2**
> > These clarifications are very helpful.
> >
> > **Experiments W4 and W5**
> > The additional experimental validation is welcome such as loss curves and more realistic settings. The experiments are in line with the minor feature learning finding.
> >
> > **Further theory additions and W3**
> > I believe the manuscript is still quite dense making it most likely less accessible to a general audience.
> > Some changes that can help are:
> > 1) Loss curve in main text instead of the appendix. For example as fig c in the first plot.
> > 2) Experiments plot as well of the gradcam.
> >
> > Furthermore the manuscript has changed extensively especially the appendix this may require an additional round of reviews and is not suitable for the reviewer-author stage. The authors have clarified the reason for the changes in the general response can they also substantiate why this not warrants a new round of reviews.
> >
> > **Relevant literature on diagonal linear networks**
> > Besides noise and learning rate recent literature on the diagonal linear network and implicit bias other effects of optimization have been studied:
> >
> > 1. Jacobs, Tom et al. “Mirror, Mirror of the Flow: How Does Regularization Shape Implicit Bias?” ArXiv abs/2504.12883 (2025): n. pag.
> > 2. Wang, Shuyang and Diego Klabjan. “A Mirror Descent Perspective of Smoothed Sign Descent.” Conference on Uncertainty in Artificial Intelligence (2024).
> > 3. Papazov, Hristo et al. “Leveraging Continuous Time to Understand Momentum When Training Diagonal Linear Networks.” International Conference on Artificial Intelligence and Statistics (2024).
> > 4. Jacobs, Tom and Rebekka Burkholz. “Mask in the Mirror: Implicit Sparsification.” ArXiv abs/2408.09966 (2024): n. pag.
> >
> > For now I keep my score.

---

> > > ### Author Response · Authors · 2025-12-03
> > >
> > > ### **Improving Readability**
> > >
> > > Thank you for your concrete suggestions regarding readability. Following your first review, we moved the helper proposition for Theorem 4.5 to the appendix and revised the corresponding explanation in the main text, which substantially reduced density and improved clarity. We also incorporated your suggestions by adding both the loss curves and the GradCAM visualizations directly into the main paper.
> > >
> > > We hope that the revised version addresses the reviewer’s concern about readability.
> > >
> > > ### **Relevant Literature**
> > >
> > > Thank you for recommending relevant papers. We reviewed the suggested references and added them to the related works section.
> > >
> > > ### **Regarding the Scope of Changes and the Need for Additional Review**
> > >
> > > The revisions to the main body are limited in scope: they focus on improving readability, clarifying statements, and relaxing certain assumptions for broader applicability. As the reviewer noted, the more substantial additions appear in the appendix, where we provide clarifications, expanded proofs, and supplementary validation.
> > >
> > > Importantly, none of these updates alter the core theoretical or empirical results presented in the original main body.  While a few theorem statements have been refined, their main claims remain unchanged, and the new results primarily relax assumptions or extend existing ones without modifying the original conclusions. These updates therefore strengthen and supplement our original results rather than change them.
> > >
> > > For these reasons, although the revision improves clarity and completeness, it does not alter the core content that was already evaluated during the initial review. We therefore believe that the scope and nature of the changes do not warrant an additional round of reviews.

---

### Official Review · Reviewer_bXyt · 2025-10-31

**Soundness:** 3
**Presentation:** 3
**Contribution:** 3
**Rating:** 6
**Confidence:** 3

**Summary:**

The paper analyzes the implicit bias of sharpness-aware minimization (SAM) in linearly separable binary classification using (L)-layer linear diagonal networks. For (L=1), both $\ell_\infty$-SAM and $\ell_2$-SAM recover the $\ell_2$ max-margin solution, matching gradient descent (GD). For depth (L=2), the behavior diverges sharply: (\ell_\infty)-SAM’s limit direction is highly initialization-dependent—converging to zero or to any standard basis vector—whereas GD aligns with the data’s dominant coordinate. In contrast, $\ell_2$-SAM exhibits **sequential feature discovery**, where the predictor first leans on minor coordinates and gradually shifts to larger ones as training progresses or initialization increases. The authors trace this to $\ell_2$-SAM’s perturbation normalization, which initially amplifies small coordinates before allowing major ones to dominate. Synthetic and real-data experiments corroborate these theoretical findings, revealing depth-sensitive and optimizer-specific implicit biases distinct from GD.

**Strengths:**

1. Exceptionally clear and easy to follow.
2. The theoretical analysis is solid and convincingly demonstrates how network depth affects the implicit bias of SAM.

**Weaknesses:**

It is unclear whether these theoretical findings can inform practical algorithmic improvements—for example, proposing a better SAM-style method.

**Questions:**

See weakness.

---

> ### Author Response · Authors · 2025-11-23
>
> We appreciate the reviewer’s careful evaluation and helpful suggestions.
>
> ###  **W1.Unclear practical algorithmic improvements.**
>
> From a dynamical viewpoint, varying the initialization scale while fixing $\rho$, and varying $\rho$ while fixing the initialization, play essentially the same role, as both control the **effective strength of the SAM perturbation**. Together with the sequential feature discovery phenomenon, this observation suggests a natural **scheduling strategy** for $\rho$: one may start training with a small perturbation radius $\rho$ to encourage the learning of dominant (major) features, and then gradually increase $\rho$ over time to promote the discovery of more subtle (intermediate and minor) features. This proposal follows directly from the mechanism identified in our theoretical analysis.
>
> Importantly, Proposition 4.4 implies that even when $\rho$ varies over time, the characterization of the most amplified coordinate remains valid instantaneously. In particular, at each time point, the index that is maximally amplified is still governed by the same principle, which suggests that controlling $\rho$ over time provides a principled way to steer which features are preferentially amplified during training.
>
> At a higher level, combining our results for both $\ell_\infty$-SAM and $\ell_2$-SAM, we show that SAM is able to prioritize **subdominant features** that GD tends to systematically ignore, as GD is heavily biased toward the most dominant features, as shown in the experimental results in E.4. This has practical implications in settings where **important but weak signals** are critical. For instance, in visual recognition tasks where an object of interest (e.g., a bicycle) appears in low-contrast or shadowed regions, GD-style training may overemphasize dominant background patterns, whereas SAM-style training, by amplifying sensitivity to minor features, can better capture such subtle structures. This suggests that SAM-style methods may be particularly advantageous in tasks where weak but semantically meaningful features are crucial.
>
> Exploring principled strategies to leverage this behavior in practical algorithms is an interesting direction for future research, but lies outside the scope of the present paper.

---

### Official Review · Reviewer_gsYm · 2025-11-01

**Soundness:** 4
**Presentation:** 3
**Contribution:** 3
**Rating:** 6
**Confidence:** 3

**Summary:**

The paper studies the bias (and training trajectory) sharpness-aware minimization in diagonal linear networks for a dataset consisting of one data point and logistic loss. In linear models, no difference between GD and SAM is found. For a larger number of layers, infinity-norm based SAM can exhibit different behavior depending on initialization and the perturbation radius of SAM. $\ell_2$ based SAM converges to the direction of the $\ell_1$ max-margin solution, but does so by first having the minor coordinates as the dominant direction and then sequentially moving, from coordinate to coordinate, to the major one.

**Strengths:**

The highlighted rich behaviors that SAM can exhibit is interesting and it is good to have more work studying logistic loss in this setting.  The paper is well written and the experiments on MNIST to some extend backup the theoretical insights.

**Weaknesses:**

Clearly, the model and data are very specialized. Maybe my main question would the impact of a non-linearity like ReLU would be. This is not covered by the experiments.

No code was provided, making it more difficult to reproduce results or checking up implementational details of the experiments. For example, I do not think the paper details how exactly the data for Figure 7 is generated (although from the picture one might guess that it could be points drawn from a Gaussian distribution with means (1,2) and (-1,-2)).

**Questions:**

What about more practical experiments for $\ell_2$ SAM and $L>2$? Do you still observe the sequential feature discovery? Or is this something that only happens in a very narrow regime?

---

> ### Author Response · Authors · 2025-11-23
>
> We thank the reviewer for their careful reading and insightful comments.
>
> ### **W1. Missing analysis of nonlinear models.**
>
> We agree that our theoretical model considers a specialized setting. The use of linearly separable data with linear diagonal networks is, however, a standard and widely adopted framework in prior work on implicit bias, as it allows precise analysis of optimization-induced effects.
>
> To address the role of nonlinearities such as ReLU, we provide empirical results in Section E.4 on CNNs with ReLU, evaluated on MNIST, SVHN, and CIFAR-10. These experiments exhibit the same qualitative behavior: compared to GD, SAM preferentially amplifies minor features, suggesting that our conclusions could extend beyond purely linear models.
>
> ### **W2. Limited reproducibility due to incomplete experimental details.**
>
> We appreciate the reviewer for pointing this out. To improve reproducibility, we have added complete experimental details as well as corresponding results in the revised version. The implementation setup, data-generation procedures, and reproduced figures are now documented in Sections C.4, D.8 and E.
>
> Regarding Figure 7 of the original paper, the reviewer’s guess is correct. The data were sampled from two Gaussian clusters centered at $(1,2)$ and $(-1,-2)$. In the revised version, we generate the dataset in exactly the same way and present it in Figure 8. For more direct comparison with the main text, we include the corresponding results in Figures 9 and 17.
>
> ### **Q1. Unclear whether sequential feature discovery persists in more practical settings.**
>
> We provide additional practical experiments for both $\ell_2$-SAM and deeper models ($L > 2$) in the revised manuscript. In particular, Figure 19 shows that the sequential feature discovery phenomenon persists in deeper networks, indicating that it is not restricted to shallow architectures.
>
> Beyond the linear diagonal setting, we extend our study to 2-layer linear networks and CNNs. In Section E.3, we demonstrate that the sequential feature discovery is also observed in 2-layer linear networks and CNNs trained on synthetic banded datasets, where we systematically vary both the dataset construction and evaluation metrics across architectures.
> Beyond these controlled linear setting, we further evaluate $\ell_2$-SAM and GD on ReLU-based CNNs trained on MNIST, SVHN, and CIFAR-10 (Section E.4). Across these datasets, we consistently observe the same qualitative behavior: $\ell_2$-SAM tends to emphasize minor or background-related features compared to GD. For example, on MNIST, $\ell_2$-SAM allocates more attention to background (black) regions, whereas GD primarily focuses on the foreground digit pixels. On SVHN, $\ell_2$-SAM similarly emphasizes dark regions corresponding to either background or fine-grained digit structure. On CIFAR-10, SAM prioritizes finer-scale, small-pixel patterns compared to GD.
> Taken together, these results suggest that sequential feature discovery is not confined to a narrow or artificial regime, but robustly persists across deeper networks ($L > 2$), and more realistic linear and nonlinear architectures.

---

> > ### Comment · Reviewer_gsYm · 2025-11-28
> >
> > Thank you for the additional experiments and details on the experimental setup. I will keep my positive score.

---

### Official Review · Reviewer_SBDC · 2025-11-11

**Soundness:** 3
**Presentation:** 3
**Contribution:** 1
**Rating:** 2
**Confidence:** 3

**Summary:**

The paper analyzes the implicit bias of sharpness aware minimization (SAM) in the setting of binary classification, linear separable data, and $L$-layer linear diagonal networks. The work shows the implicit bias changes with depth. In addition, unlike GD / GF, the limiting direction of $\ell_\infty$ SAM is dependent on initialization.

**Strengths:**

The paper is overall easy to read. In addition, the fact that the implicit bias of SAM changes with depth is interesting.
A nice result is that $\ell_\infty$-SAM's limiting direction is dependent on initialization unlike GD / GF. (That said, for non linear architectures, GD / GF is also dependent on initialization).

**Weaknesses:**

One weakness is the restrictive setting as the paper deals with L-layer diagonal linear networks and assumes the data is linear separable.

Theorem 4.2 assumes directional convergence of the inner and outer layer as well as their flows which is an extremely strong assumption and something that is highly nontrivial to prove in general.

In addition, the analysis is restricted to a dataset consisting of one point (that has a monotonic structure with respect to its entries).

Finally, the analysis heavily uses the fact that the depth is 1 or 2 and it seems hard to generalize the analysis to the multi-layer case (L > 2).

**Questions:**

1. For Theorem 4.1 and Theorem 4.2, what are the conditions on $\rho$ (the radius of SAM)?

2. In Theorem 4.2, could the authors explain what the main obstructions are for proving directional convergence in the special case of one data point?

3. In theorem 3.2, what how do the coordinates of the coefficient $\beta(t)$ grow in the original flow? In a related vein, what is the rate of convergence of the loss using SAM in the original flow?

4. How would the theorems in section 4 change for $L > 2$?

5. Are the figures generated using rescaled flow or the original flow?

---

> ### Author Response · Authors · 2025-11-23
>
> Thank you for the thoughtful and constructive feedback.
>
> ### **W1. Restrictive setting such as $L$-layer linear diagonal networks and linearly separable data.**
>
> Linear diagonal networks under linear separability can indeed be viewed as a simplified setting. However, this setup has been widely used as a **theoretical testbed** for studying implicit bias, because it is a simplified model with a homogeneous structure that exhibits depth-induced implicit bias while being relatively mathematically tractable. Despite their simplicity, such models are known to exhibit nontrivial phenomena, including distinct lazy and rich regimes depending on initialization, which are central to understanding implicit bias.
>
> As noted in the related works (lines 98–107 and 120–126), a large number of prior papers study implicit bias specifically in linear diagonal networks under linear separability. In this context, our setting follows a standard and theoretically meaningful choice, and we believe it is a reasonable model for developing a precise theory of SAM.
>
> |  | network | data (or task) | implicit bias |
> | --- | --- | --- | --- |
> | Soudry et al., 2018 [1]  | depth 1 linear | linearly separable | GD |
> | Gunasekar et al., 2018 [2]  | $L$-layer linear diagonal | linearly separable | GD |
> | Yun et al., 2020 [3], Moroshko et al., 2020 [4] | $L$-layer linear diagonal | separable | GD |
> | Woodworth et al., 2020 [5], Nacson et al., 2022 [6] | 2-layer linear diagonal | (regression) | GD |
> | Pesme et al., 2021 [7] | 2-layer linear diagonal | (regression) | SGD |
> | Even et al., 2023 [8] | 2-layer linear diagonal | (regression) | GD, SGD |
> | Andriushchenko & Flammarion, 2022 [9]  | 2-layer linear diagonal | (regression) | SAM |
> | Clara et al., 2025 [10] | $L$-layer linear diagonal | (regression) | SAM |
> | Lyu et al., 2025 [11] | 2-layer linear diagonal | (regression) | Momentum based methods |
>
> Moreover, our experiments complement the theoretical results by extending them to 2-layer linear networks and CNNs. In Section E.3, we show that the sequential feature discovery phenomenon is also observed in 2-layer CNNs and linear networks trained on synthetic banded datasets, where we systematically vary both the dataset construction and evaluation metrics across architectures. In Section E.4, we further extend our analysis to a practical setting by studying CNNs trained on MNIST/SVHN/CIFAR-10 datasets.
>
> ### **W2. Strong directional-convergence assumptions in Theorem 4.2.**
>
> Firstly, we would like to note that Theorem 4.2 is not the central contribution of our work. Its purpose is solely to relate the implicit bias of $\ell_2$-SAM to that of GD via the limit direction. The main contribution of our work appears in the later parts of the paper, where we uncover a sequential feature discovery phenomenon, the surprising and novel main result of our paper. For this reason, we did not initially attempt to optimize Theorem 4.2 under the weakest possible assumptions.
>
> Nevertheless, trying to address the reviewer’s comment, we have significantly improved Theorem 4.2 in the revised version. We remove the directional-convergence assumption on the parameter flow and relax the required convergence of the inner and outer layers $\boldsymbol w^{(1)},$ $\boldsymbol w^{(2)}$ to that of the predictor $\boldsymbol \beta := \boldsymbol w^{(1)} \odot \boldsymbol w^{(2)}$. The result now holds for almost all linearly separable dataset under only two mild conditions: (i) vanishing loss and (ii) convergence of the predictor.
>
> Moreover, in the original version, Theorem 4.8 already provides a complementary guarantee showing that, in the special case of a single-example dataset, $\boldsymbol{\beta}(t)$ automatically converges in the $\ell_1$ max-margin direction **without any assumption,** whenever the initialization scale $\alpha$ exceeds a threshold (line 521-526).
>
> Finally, as explained in Q2 of the rebuttal, establishing directional convergence of the predictor without any assumptions is a highly nontrivial open problem. Thus, with the current analytical techniques available, the remaining assumptions in the improved Theorem 4.2 are likely close to minimal.

---

> ### Author Response · Authors · 2025-11-23
>
> ### **W3. Restricted dataset with one point.**
>
> First, we agree that restricting the analysis to a single-point dataset is a limitation. As another reviewer also raised this point and requested a discussion of the challenges in extending to multiple points as well as simulations on multi-sample datasets, we address this issue in our global response, where we (i) extend our asymptotic results to (almost) all general linearly separable datasets and (ii) provide empirical results on multi-point datasets. We would be grateful if you could refer to that part for the detailed discussion.
>
> Second, the assumption that the entries of the data point have a monotonic structure is made purely for notational convenience and can be imposed **without loss of generality** (by appropriately reordering the coordinates). Our arguments only use the order of the coordinate magnitudes, not any special structure beyond that.
>
> ### **W4 & Q4. Limited depth generality (analysis restricted to $L = 1$ and $2$).**
>
> First, our results for $\ell_\infty$-SAM are already stated for arbitrary depth $L$, as presented in Section 3, and therefore are not limited to the case $L=1$ or $2.$
>
> Second, for $\ell_2$-SAM, we have extended some of the main components of our depth-2 analysis to deeper networks. In Appendix D, we provide Lemma D.13, which generalizes Lemma 4.3 to the case $L>2$, and Proposition D.14, which extends Proposition 4.4 to deeper networks. Using these results, we show that the feature amplification dynamics persist in deeper models. Concretely, for $L=2$, we compare $m_c(t)$ with $\mu_j$ to identify the most amplified feature. For $L>2$, we introduce new quantities $z_c(t)$ and $z_j$, and prove that the ordering of $z_j$ matches that of $\mu_j$. This establishes that the core mechanism underlying sequential feature discovery extends beyond depth 2.
>
> Third, we provide empirical evidence in Section D.8.3 showing that the main phenomenon we analyze in the depth-2 model—namely, sequential feature discovery—also consistently appears in deeper models with $L>2$.
>
> Finally, regarding asymptotic implicit bias of Theorem 4.2, we note that for GD, the implicit bias is known to be of $\ell_{2/L}$-type for depth-$L$ linear diagonal networks. It is therefore natural to conjecture that $\ell_2$-SAM exhibits a similar $\ell_{2/L}$-type implicit bias. However, since the asymptotic limit direction is not the main focus of our work, we leave a rigorous treatment of this question for future work.
>
> ### **Q1. Conditions on $\rho$ in Theorem 4.1 and 4.2.**
>
> Theorem 4.1 and 4.2 hold for any fixed perturbation radius $\rho$, including $\rho=0$ (GD). In the revised version, we state this explicitly. The reason is that both theorems characterize the asymptotic regime where the predictor norm diverges. In this regime, the $\rho$-scaled perturbation becomes negligible, so the results are independent of the specific value of $\rho.$
>
> ### **Q2. Obstructions to proving directional convergence in Theorem 4.2.**
>
> First, we would like to note that we have improved Theorem 4.2 in the revised version by removing the assumption of directional convergence of the gradient. The only remaining assumption is directional convergence of the predictor, and the result now applies to almost all linearly separable datasets. Moreover, Theorem 4.8 shows that, in the special case of a single-example dataset, for large initialization scale $\alpha$, the predictor indeed converges in direction without any such assumption.
>
> Next, we discuss why showing the directional convergence for $\ell_2$-SAM flow trajectories is difficult in general. One natural idea is to adapt the proof on GF from Ji & Telgarsky, 2020 [12]. However, their proof relies essentially on the fact that GF satisfies
>
> $$
> \dot{\boldsymbol{\theta}}=-\nabla \mathcal{L}(\boldsymbol{\theta}),
> $$
>
> i.e., it is the gradient flow of a scalar potential. This structure enables the geometric tools in their analysis.
>
> In contrast, the $\ell_2$-SAM flow
>
> $$
> \dot{\boldsymbol{\theta}}=-\nabla \mathcal{L}\left(\boldsymbol{\theta}+\rho\frac{\nabla \mathcal{L}(\boldsymbol{\theta})}{\\|\nabla \mathcal{L}(\boldsymbol{\theta})\\|_2}\right),
> $$
>
> is **not** the gradient flow of any scalar function. The perturbation and normalization break the structure, so the Ji & Telgarsky framework does not extend to $\ell_2$-SAM. This structural obstacle is precisely why establishing directional convergence is highly nontrivial and remains open outside the range of $\alpha$ covered by Theorem 4.8.
>
> We have also added a brief discussion of the Ji & Telgarsky, 2020 [12] result to the Related Works section (lines 101–102) to clarify its connection to our setting.

---

> ### Author Response · Authors · 2025-11-23
>
> ### **Q3. Growth of $\beta_j(t)$ and loss convergence rate in the original flow in Theorem 3.2.**
>
> Regarding the growth of the coefficients $\beta_j(t)$ in the original flow, we note that the rescaled flow and the original flow trace the **same spatial trajectory** in parameter space. As a consequence, the qualitative behavior of each coordinate is identical: when the initialization satisfies $\alpha_j \le \rho$, the corresponding coordinate remains bounded, whereas when $\alpha_j > \rho$, the coordinate diverges to $+\infty$. The convergence rate of the loss in the original flow is governed by the growth of $\boldsymbol{\beta}(t)$, since the loss takes the form $\ell(\langle \boldsymbol{\beta}, \boldsymbol{\mu}\rangle)$.
>
> In the revised manuscript, we added lines 194-208, Remark 3.3 and 3.4, where we describe the correspondence between the rescaled and original flows. Denoting the original SAM flow by $\boldsymbol{w}\_\text{orig}(\tau)$ and the rescaled flow by $\boldsymbol{w}(t)$, there exists a time-reparameterization map $t_\text{orig}=\tau(t)$ such that
>
> $$
> \boldsymbol w_\text{orig}(\tau(t))=\boldsymbol w(t),
> $$
>
> with
>
> $$
> \tau(t)= \int_{0}^{t} -\frac{1}{\ell'(\boldsymbol\beta(\hat {\boldsymbol \theta}(s))^{\top}\boldsymbol\mu)}ds.
> $$
>
> Since $\ell'(u)\to 0$ as $u \to \infty$, the rescaled time variable accelerates relative to the original time scale in the large-margin regime. In particular, when $\beta_j(t)$ diverges, the system enters this regime and the rescaled flow proceeds faster than the original one. As a result, the growth of $\beta_j$ as a function of the original time variable $t_\text{orig}$ is strictly **slower** than its growth in the rescaled time.
>
> Obtaining an explicit growth rate for $\beta_j(\tau)$, or for the corresponding loss decay, is intractable as far as we are aware. Even in the single-example setting, the original $\ell_\infty$-SAM flow contains a shared scalar factor
>
> $$
> -\ell'(\langle \boldsymbol\beta(\hat{\boldsymbol\theta}(t)), \boldsymbol\mu \rangle),
> $$
>
> that couples all coordinates. This global coupling prevents coordinate-wise decoupling and makes closed-form characterization of the coordinate growth or loss convergence rate infeasible.
>
> Our analysis therefore focuses on the **spatial trajectory** of $\boldsymbol{\beta}$ rather than its precise time parametrization. Importantly, this spatial trajectory is unchanged when the common scalar factor above is omitted, which enables coordinate-wise analysis and leads to the characterization in Theorem 3.2.
>
> For further discussion on the interpretation of the finite-time blow-up in the rescaled flow, please see our response to comment W7 of reviewer f8R2.
>
> ### **Q5. Whether the figures use the rescaled SAM flow or the original flow**
>
> Figure 1 is generated using discrete updates. Figures 2 and 3 are generated using the rescaled $\ell_\infty$-SAM flow. Importantly, the trajectory of the rescaled $\ell_\infty$-SAM flow coincides with that of the original $\ell_\infty$-SAM flow, so the behavior in Figure 2 is identical under both flows.
>
> For the figures in the appendix, we explicitly distinguish the cases. We use the terms "$\ell_\infty$-SAM flow" and "$\ell_2$-SAM flow" for the original flows, and "rescaled $\ell_\infty$-SAM flow", "rescaled $\ell_2$-SAM flow", or "discrete" updates when those versions are used.

---

> ### Author Response · Authors · 2025-11-23
>
> **References**
>
> [1] Daniel Soudry, Elad Hoffer, Mor Shpigel Nacson, Suriya Gunasekar, and Nathan Srebro. The implicit bias of gradient descent on separable data. Journal of Machine Learning Research, 19(70): 1–57, 2018.
>
> [2]  Suriya Gunasekar, Jason D Lee, Daniel Soudry, and Nati Srebro. Implicit bias of gradient descent on linear convolutional networks. Advances in neural information processing systems, 31, 2018.
>
> [3]  Chulhee Yun, Shankar Krishnan, and Hossein Mobahi. A unifying view on implicit bias in training linear neural networks. arXiv preprint arXiv:2010.02501, 2020.
>
> [4]  Edward Moroshko, Blake E Woodworth, Suriya Gunasekar, Jason D Lee, Nati Srebro, and Daniel Soudry. Implicit bias in deep linear classification: Initialization scale vs training accuracy. Advances in neural information processing systems, 33:22182–22193, 2020.
>
> [5] Blake Woodworth, Suriya Gunasekar, Jason D Lee, Edward Moroshko, Pedro Savarese, Itay Golan, Daniel Soudry, and Nathan Srebro. Kernel and rich regimes in overparametrized models. In Conference on Learning Theory, pp. 3635–3673. PMLR, 2020.
>
> [6]  Mor Shpigel Nacson, Kavya Ravichandran, Nathan Srebro, and Daniel Soudry. Implicit bias of the step size in linear diagonal neural networks. In International Conference on Machine Learning, pp. 16270–16295. PMLR, 2022.
>
> [7]  Scott Pesme, Loucas Pillaud-Vivien, and Nicolas Flammarion. Implicit bias of sgd for diagonal linear networks: a provable benefit of stochasticity. Advances in Neural Information Processing Systems, 34:29218–29230, 2021.
>
> [8]  Mathieu Even, Scott Pesme, Suriya Gunasekar, and Nicolas Flammarion. (s) gd over diagonal linear networks: Implicit bias, large stepsizes and edge of stability. Advances in Neural Information Processing Systems, 36:29406–29448, 2023.
>
> [9] Maksym Andriushchenko and Nicolas Flammarion. Towards understanding sharpness-aware minimization. In International conference on machine learning, pp. 639–668. PMLR, 2022.
>
> [10]  Gabriel Clara, Sophie Langer, and Johannes Schmidt-Hieber. Training diagonal linear networks with stochastic sharpness-aware minimization. arXiv preprint arXiv:2503.11891, 2025.
>
> [11] Bochen Lyu, He Wang, Zheng Wang, Zhanxing Zhu. Effects of Momentum in Implicit Bias of Gradient Flow for Diagonal Linear Networks. Proceedings of the AAAI Conference on Artificial Intelligence. 2025. p. 19242-19250.
>
> [12] Ziwei Ji, Matus Telgarsky. Directional convergence and alignment in deep learning. *Advances in Neural Information Processing Systems*, 2020, 33: 17176-17186.

---

### Author Response · Authors · 2025-11-23
**Summary of Revision**

We sincerely thank the reviewers for their thoughtful comments and constructive feedback. These insights helped us substantially improve the clarity and strength of the paper. In our revision, we have color-coded the modifications in green.

Below, we outline some noteworthy updates:

- **Expanded empirical evaluation and practical experiments**

    Added results for discrete SAM updates, multi-point datasets, and deeper network settings (Sections C.4 and D.8), as well as experiments with random Gaussian initialization, alternative two-layer architectures(lienar network and convolutional neural network), and CNNs across practical datasets (Section E).

- **Strengthened asymptotic theory**

    Extended Theorems 3.1, 4.1, and 4.2 from the single-example regime to general linearly separable datasets.

- **Extended theory to deeper networks**

    Generalized Lemma 4.3 and Proposition 4.4 to depth-$L$ settings (Lemma D.13 and Proposition D.14).

- **Clarified relation to prior saddle-to-saddle work**

    Added an explicit comparison with saddle-to-saddle dynamics and clarified the conceptual distinctions (Sections 1.2 and A.2).

- **Added clarification on flow reparameterization**

    Included additional explanations regarding time reparameterization in the flow dynamics (Section 2, Remarks 3.3 and 3.4).

- **Expanded discussion of the multi-point setting**

    Added a discussion of the challenges and limitations of the multi-point case at the beginning of Section 3.2.

- **Added numerical illustrations for Theorem 4.6**

    Included new numerical experiments and clarifying interpretations for Theorem 4.6 (Section D.7).

- **Improved overall clarity and readability**

    Refined presentation and exposition, particularly in Section 4.

---

### Author Response · Authors · 2025-11-23
**Global Response to Concerns about the Single-point Dataset Assumption**

We thank Reviewers f8R2 and SBDC for pointing out the limitation of our initial focus on a single-point dataset. This feedback prompted us to both strengthen our theoretical results and to add empirical results on multi-point datasets, to demonstrate that the phenomena observed in the single-point setting persist in more realistic scenarios.

First, Thms. 3.1, 4.1, and 4.2 were originally stated only for a single-point dataset. In the revised version, we strengthen all three results to **general linearly separable datasets:**

- $\ell_\infty$-SAM on a linear model converges in the $\ell_2$ max-margin direction (Thm. 3.1);
- $\ell_2$-SAM on a linear model converges in the $\ell_2$ max-margin direction (Thm. 4.1);
- $\ell_2$-SAM on a depth-2 linear diagonal network converges in the $\ell_1$ max-margin direction (Thm. 4.2).

Thus, the max-margin biases we originally identified in the single-point setting are not artifacts of that toy example, but hold for general linearly separable datasets.

Secondly, we complement our single-example theory with **empirical results on** **multi-point datasets**, which closely mirror the behavior observed in the single-point setting. In App. C.4.2, we construct random linearly separable datasets by sampling two Gaussian clusters (Fig. 8) and, under the same experimental setup as in Fig. 2, we plot the resulting $\ell_\infty$-SAM trajectory (Fig. 9). The resulting curves are nearly indistinguishable from the single-point case, supporting that the qualitative conclusions of Thm. 3.2 and Cor. 3.4 also hold in the multi-point setting. Furthermore, in App. D.7.2, we generate several random linearly separable datasets and observe that the sequential feature discovery phenomenon persists in the multi-point case as well (Fig. 17).

Lastly, as we explain in lines 233–246 of the revised manuscript, there are substantial theoretical obstacles to extending our **pre-asymptotic** results from the single-point dataset to general multi-point datasets, which crucially differs from its asymptotic counterpart. For the asymptotic behavior (Thms. 3.1, 4.1, and 4.2), we analyze the limit direction when the parameter trajectory $\mathbf{w}(t)$ diverges and the $\rho$-scaled SAM perturbation becomes asymptotically negligible. In this regime, SAM and GD share the same limiting direction, which makes it possible to establish asymptotic equivalence even in the multi-point setting, once margin growth are ensured.

By contrast, in regimes where the perturbation is **not** negligible, SAM behaves fundamentally differently from GD, and its dynamics are governed by the gradient evaluated at a perturbed point dependent on the radius $\rho$. In the multi-point setting, the gradient direction is no longer aligned with a single vector (as in the one-point case), but becomes a highly nonlinear mixture of all training examples, whose relative influence keeps changing as margins evolve. The SAM perturbation then rescales this mixed gradient at every step, so that both its magnitude and direction depend on the entire margin distribution and change as influential examples shift. As a result, in the multi-point regime where SAM and GD genuinely diverge, the SAM flow ODE does not admit any simple reduction or tractable dynamical representation that we are aware of. This is precisely what motivates us to analyze the minimalist single-example dataset $\mathcal{D}_{{\mu}} = \\{(\boldsymbol{\mu}, +1)\\}$: in this setting the SAM dynamics admit a tractable dynamical characterization, while still revealing the depth-dependent phenomena that are unique to SAM (Thms. 3.2, 4.5, and 4.6).

Even though our analysis is conducted under a single-point setting, a characterization of **finite-time behavior** that exhibits an implicit bias substantially different from its infinite-time counterpart is still highly meaningful. A closely related example that underscores the importance of finite-time analysis arises in the study of the implicit bias of Adam. Due to the presence of the stability term ε in the denominator, Adam induces an $\ell_2$-type implicit bias in its asymptotic behavior [1]. However, empirical observations in realistic finite-time regimes reveal dynamics that deviate significantly from this prediction. To reconcile this discrepancy, a subsequent work has analyzed modified variants of Adam without the ε term, ultimately demonstrating that the implicit bias aligns instead with an $\ell_\infty$-type behavior [2].

Our work follows a similar line of reasoning. We demonstrate that the finite-time dynamics of SAM give rise to an implicit bias that is **substantially different** from the bias predicted by infinite-time analyses. By explicitly focusing on the practically relevant finite-time regime, we show that restricting attention solely to asymptotic behavior can be misleading, and that a systematic understanding of **pre-asymptotic implicit bias** is essential for accurately describing real-world optimization dynamics.

---

> ### Author Response · Authors · 2025-11-23
> **References**
>
> [1] Bohan Wang, Qi Meng, Wei Chen, Tie-Yan Liu. The implicit bias for adaptive optimization algorithms on homogeneous neural networks. In: *International Conference on Machine Learning*. PMLR, 2021. p. 10849-10858.
>
> [2] Chenyang Zhang, Difan Zou, Yuan Cao. The implicit bias of adam on separable data. *Advances in Neural Information Processing Systems*, 2024, 37: 23988-24021.

---

### Meta-Review · Area_Chair_cP8a · 2026-01-03

**Summary:**

The paper studies the implicit bias of sharpness-aware minimization (SAM) under continuous time flows for $L$-layer linear diagonal networks trained with logistic loss, comparing $\ell_\infty$ and $\ell_2$ variants to gradient descent flow (GD). For $L=1$, all methods exhibit the same implicit bias, converging to the $\ell_2$ max margin direction on linearly separable data. For $L=2$, the behavior differs: $\ell_\infty$ SAM can converge to a limiting direction distinct from GD, while $\ell_2$ SAM shares the same asymptotic bias but may exhibit different finite time trajectories, temporarily aligning with other directions before ultimately converging to the max margin solution. In the revision, the asymptotic results for the $L=2$ case, which originally relied on a single point dataset assumption, are strengthened to hold for general linearly separable datasets.

Reviewers generally find the analysis technically sound and the paper clearly written, while also noting the stylized nature of the setting (diagonal networks and an initial focus on a single example dataset) and raising questions about assumptions, generality, and empirical scope. The authors’ revision addresses many of these concerns, including extending key asymptotic results, expanding the empirical evaluation, and clarifying technical aspects such as time reparameterization. A single review assigns a low score, emphasizing restrictiveness and strong assumptions. This perspective is noted, and the authors have made a clear effort in the revision to address or mitigate many of the issues raised. Otherwise, based on the explicit content of the discussion as a whole, particularly indications of revised, more favorable assessments and the sustained positivity of above threshold reviews, the discussion trajectory appears to favor acceptance. Overall, while the work has clear limitations and is somewhat stylized, it offers a solid treatment of SAM’s behavior in the logistic loss setting, fits well within the implicit bias literature, and the balance of reviewer feedback supports acceptance.

The authors are encouraged to further revise the paper for clarity and organization, and to ensure that the sequential feature learning behavior is more clearly situated within the existing literature on the body of work on incremental learning.

**Reviewer Concerns:**

The rebuttal and revision address many of the substantive reviewer concerns. In particular, concerns about the reliance on a single point dataset were addressed by extending the main asymptotic results to general linearly separable datasets and by adding additional empirical results. Requests for clarification on time reparameterization, finite time behavior, and related technical details were addressed through additional explanations and experiments. At the same time, it should be noted that several of these additions, including new theoretical results and expanded analyses, were introduced during the rebuttal stage and have not undergone a full additional round of peer review, so their epistemological status should be interpreted with appropriate caution.

Some concerns remain partially outstanding, such as further streamlining and clearer positioning relative to the broader implicit bias literature and the relationship between sequential feature discovery and prior work on saddle to saddle dynamics and incremental learning. Additionally, questions about how directly the theory translates into practical algorithmic guidance are acknowledged by the authors but largely left as directions for future work.

**Reviewer Scores:**

Reviewer f8R2 explicitly stated that most concerns were addressed and indicated they would raise their score after the revision.

Reviewer gsYm stated that the additional experiments and clarifications resolved their concerns and that they would keep their positive score.

Reviewer bXyt did not indicate a change in score but expressed generally positive views, and their main questions were addressed in the rebuttal; their score would likely remain the same or increase slightly.

Reviewer uC1X acknowledged substantial improvements and stated they would keep their score for now, while noting remaining concerns. The phrasing suggests openness to reconsideration pending further streamlining (but no explicit intention to revise the score was stated).

Reviewer SBDC did not explicitly update their score. While several of their technical concerns were addressed in the revision, it remains unclear whether this would lead to a change in their assessment.

---

### Decision · Program_Chairs · 2026-01-26

Accept (Poster)